

# A statistical global burned area model for seamless integration into Dynamic Global Vegetation Models

Blessing Kavhu[1,3], Matthew Forrest[1], Thomas Hickler[1,2]

[1]Senckenberg Biodiversity and Climate Research Centre (SBiK-F), Frankfurt am Main, Germany.
[2]Institute of Physical Geography, Goethe University Frankfurt am Main, Frankfurt am Main, Germany
[3]Environmental Studies, University of California, Santa Cruz, 1156 High 5th, Santa Cruz, 95064, California, United States.

*Correspondence to*: Blessing Kavhu(kavhublessing@gmail.com)

**Abstract.** Fire-enabled Dynamic Global Vegetation Models (DGVMs) play an essential role in predicting vegetation dynamics and biogeochemical cycles amid climate change. Modeling wildfires has been challenging in process-based biophysics-oriented DGVMs, as human behaviour plays a crucial role. This study aims to reveal a global statistical model for the relationships between biophysical and socioeconomic drivers of wildfire dynamics and monthly burned area (BA) that can be integrated into DGVMs. We developed Generalised Linear Models (GLMs) to capture the relationships between potential predictor variables that are simulated by DGVMs and/or available in future scenarios and the latest global burned area product from GFED5. Predictor variables were chosen to represent aspects of fire weather, vegetation structure and activity, human land use and behaviour and topography. The final model was chosen by minimizing collinearity and by maximizing model performance in terms of reproducing observations. The final model included eight predictor variables encompassing the Fire Weather Index (FWI), a novel Gross Primary Productivity Index (GPPI), Human Development Index (HDI), Population Density (PPN), Percentage Tree Cover (PTC), Percentage Non-Tree Cover (PNTC), Number of Dry Days (NDD), and Topographic Positioning Index (TPI). Given its simplicity, our model demonstrated a remarkable capability, explaining 56.8% of the burnt area variability, comparable to other state-of-the-art global fire models. FWI, PTC, TPI and PNTC were positively related to BA, while GPPI, HDI, PPN, and NDD were negatively related to wildfire. While the model effectively predicted the spatial distribution of burned areas (NME = 0.72), its standout performance lay in capturing the seasonal variability, especially in regions often characterized by distinct wet and dry seasons, notably southern Africa, Australia and parts of South America ($R^2 > 0.50$). Our model reveals the robust predictive power of fire weather and vegetation dynamics emerging as reliable predictors of seasonal global fire patterns. The presented model should be compatible with most, if not all, DGVMs used to develop future scenarios.



## 1 Introduction

Globally, the impacts of climate change continue to manifest through extreme weather events and changes in weather patterns (Clarke et al., 2019). Notably, climate change has led to more severe fire weather in large parts of the world and record fires have recently occurred in Australia and Canada, each burning more than 15 million ha (Barnes et al., 2023; Jain et al., 2024). Even though the effects of fires may be positive through contributing to selected natural ecosystem processes, large and frequent fires are often destructive and have far-reaching effects through loss of life, biodiversity, landscape aesthetic value, and increase in forest fragmentation and soil erosion (Bowman et al., 2017; Knorr et al., 2016; Nolan et al., 2022). The negative role of climate change in driving large and frequent burning has been well documented (Brown et al., 2023). However, climate change by itself does not fully account for the recent changes in global wildfire patterns as human activities are crucial drivers as well (Pausas and Keeley, 2021). For instance, recent empirical investigations have highlighted a notable 25% reduction in burnt area extent over the past two decades, explicitly attributing this decline to human activities (Andela et al., 2017). Wu et al. (2021) argue that future demographic and climate patterns will cause an increase in burned areas, particularly in high latitude warming and tropical regions. However, Knorr et al. (2016) concludes that under a moderate emissions scenario, global burned areas will continue to decline, but they will begin to rise again around mid-century with high greenhouse gas emissions. Cunningham et al. (2024), on the other hand reported that although total burned area is declining globally, extreme fire events are increasing as consequence of climate change especially in boreal and temperate conifer biomes. Future global fire dynamics are clearly driven by the overarching interaction between human activities (altered ignition patterns, surveillance and management) and climate (Krawchuk et al., 2009). Accurately evaluating these factors through modeling could guide prescribing solutions that will ensure reliable fire management and attainment of Sustainable Development Goals (SDGs) (Koubi, 2019; Robinne et al., 2018).

Modeling continues to be an essential tool for comprehending and forecasting wildfire dynamics, founded on the intricate interplay among fire weather, vegetation, and human activities (Bistinas et al., 2014; Hantson et al., 2016). Models for wildfire can be process-based or statistical. While process-based models delve into the physics and dynamics of wildfires and vegetation, statistical models, on the other hand, tend to focus on analyzing historical data and identifying correlations to predict future wildfire events (Morvan, 2011; Xi et al., 2019). Process-based models such as fire-enabled DGVMs stand out in understanding interactions between climate, vegetation, and human activities in a mechanistic manner (Hantson et al., 2016; Rabin et al., 2017). However, their predictive skill is often not yet satisfactory (Hantson et al., 2020). One of their greatest limitations lies in representing the often-dominating effects of humans on fire ignitions, fire spread, and fire suppression in a mechanistic process-based way as this might be elusive (Archibald, 2016; DeWilde and Chapin, 2006; Hantson et al., 2022). Hence statistical approaches have often been used to evaluate human impacts on wildfires, in combination with weather and vegetation drivers (Haas et al., 2022; Kuhn-Régnier et al., 2021). Besides, some authors reported that the application of statistical models for ecosystems other than the ones used in their derivation is often not reliable (e.g Perry, 1998). Integration



of mechanistic process-based techniques and statistical methods remains a lasting solution to advance our understanding of
fire dynamics.

Global wildfire modeling offers a macroscopic perspective, allowing researchers to analyze large-scale patterns across diverse
ecosystems (Doerr and Santín, 2016; Flannigan et al., 2009). The strength of modeling fires at a global scale lies in its ability
to capture overarching patterns (spatial, seasonal and inter-annual) that might provide valuable insights for strategic wildfire
control. While one can argue about the potential oversimplification of local factors and the challenges in representing fine-
scale heterogeneity, global models do, on the other hand, excel in capturing and understanding the effect of climate change,
partly because they capture large climatic gradients (Robinne et al., 2018). The ability to capture the interconnectedness of
ecosystems and fire regimes on a planetary scale contributes to a more holistic approach to understand global vegetation
dynamics and carbon cycling (Bowman et al., 2013; Kelly et al., 2023). As such, studies on evaluating drivers of burnt areas
at a global scale in the face of ongoing climatic shifts are crucial in ensuring sustainable management of vulnerable ecosystems.

There is a growing recognition of the significance of exploring both inter-annual and seasonal variations to comprehensively
understand the dynamics of fire across diverse ecosystems (Dwyer et al., 2000), partly because of the strong seasonal dynamics
of vegetation. Also, understanding seasonal cycles of fires helps to identify peak fire seasons, regions prone to seasonal
outbreaks, potential shifts in fire regimes over time and facilitating adaptive management strategies (Carmona-Moreno et al.,
2005). Incorporating monthly data in global fire modeling helps researchers to accurately capture seasonal variations in fire
activity. Hence, global models developed using monthly data are necessary.

Recent efforts have seen global burned area models based on diverse datasets and statistical approaches such as Convolution
Neural Network (CNN)(Bergado et al., 2021), Random Forest (RF) and generalized additive models (GAM) (Chuvieco et al.,
2021). However, the integration of these techniques into DGVM is yet to be realized.  Haas et al. (2022)) developed a statistical
global model for burned area using a GLM, however, their model does only simulate annual dynamics, not seasonal patterns.
Generally, most earlier fire modules in DGVMs were informally parameterised models and do not consider the fuller range of
predictors available in a more rigorous statistical framework (Fosberg et al., 1999; Pfeiffer et al., 2013). This left an opportunity
to improve burned area models in DGVMs to accurately represent the detailed seasonal dynamics. To our knowledge, there
haven't been any reports on a simpler and more efficient statistical model specifically crafted to capture the seasonal cycles of
global burned areas, while also being easily integrated into DGVMs. Closing this gap can best be facilitated by using up-to-
date remote sensing datasets pertinent to fire modeling. This integration can efficiently enhance our comprehension of
inadequately understood factors while leveraging the potential of finely detailed temporal resolution burnt area datasets using
a DVGM-integrable statistical model.





The main aim of this research is to build a parsimonious statistical model for global seasonal burned areas that can be integrated
into a DGVM. The specific objectives are to 1) to improve our understanding of major drivers of global burned area dynamics,
2) to leverage a GLM for predicting global burnt areas using DGVM-integrable predictors and 3) to evaluate the interannual
and seasonal cycles of burnt area extent, both globally and regionally.

## 2 Data and Methods

In this study, we used a GLM to assess the drivers and distribution of global wildfires based on a combination of vegetation,
weather, anthropogenic and topographic predictors. The spatial and temporal variability (interannual and seasonality) was also
evaluated. Fig. 1 provides an overview of the steps that were followed during modeling.

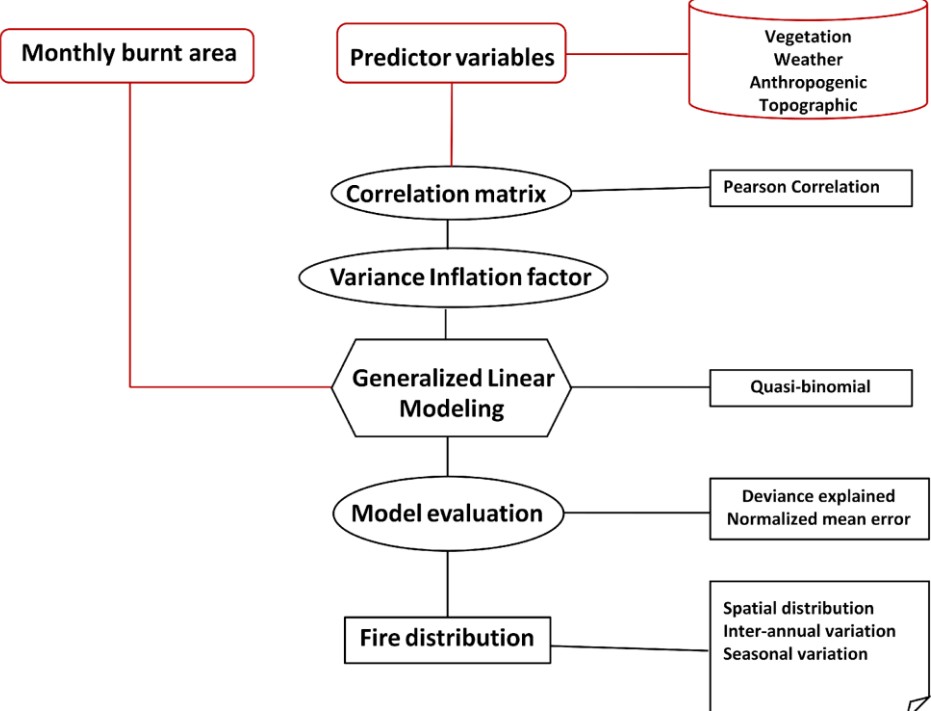


**Figure 1: Study workflow showing an overview of steps followed in model calibration and evaluation together with the outputs.**






**2.1 Fire data**

Monthly BA data for the period 2002 and 2018 were derived from monthly mean fractional BA from the GFEDv5. GFED 5 data are selected because of their improved ability to detect burnt area scars (Chen et al., 2023). GFED5 BA data are classified according to 17 major land cover types using the MODIS classification scheme. We used this land cover information to remove burnt area in cropland land cover type (type croplands and croplands/natural vegetation mosaic), to exclude the effect of cropland residue burning which we suppose likely has different drivers from burning in non-arable lands. BA data comes at a resolution of 0.25° × 0.25°, therefore we aggregated it by a factor of 2 to a resolution of 0.5°. This was done for ease of processing at a global scale and at the same time to ensure that our outputs are DGVM integrable since they are commonly applied at 0.5° globally.

**2.2 Predictor variables**

Whilst there are many possible variables that could be tried as predictors of fire, especially in terms of socioeconomics predictors, we here only use variables which don't prohibit the use of the model for future projections. These variables are: climate and vegetation variables typically available in a DGVM framework; socioeconomic variables with future scenario projections; and time-invariant topographic variables. Previous studies used several variables that we couldn't include due to lack of future scenario projections such as nighttime lights, cattle density (Forkel et al., 2019a), Vegetation optical depth (Forkel et al., 2019b), Lightning (Rabin et al., 2017), Soil moisture (Mukunga et al., 2023), soil fertility (Aldersley et al., 2011). Consequently, we considered predictor variables that are compatible with DGVM integration to calibrate the model effectively. The chosen predictor variables were categorized based on their representational nature and their roles in fire modeling. Table 1 provides a comprehensive overview of each variable category, including their sources and spatio-temporal resolutions.

| Predictor | Abbreviations | Classification category *(Climate, vegetation, landcover, landscape fragmentation, ignition, suppression topographic effect)* | Original spatial resolution | Temporal resolution | Source |
|---|---|---|---|---|---|
| Percentage Grass cover | PGC | Vegetation | 300m | Annual | ESA CCI landcover |
| Percentage non-tree vegetation cover | PNTC | Vegetation | 250m | Annual | MODIS |
| Topographic positioning index | TPI | Topography | 90m | Static | digital elevation model products of global 250 m GMTED2010 and |




| | | | | | near-global 90 m SRTM4.1dev. |
|---|---|---|---|---|---|
| Human Development Index | HDI | Ignition/suppression/ fragmentation | subnational | Annual | Global data lab |
| Road density | RD | Ignition/suppression/ fragmentation | 0.5° × 0.5° | Static | Global Roads Inventory Project (GRIP) database |
| Population density | PPN | Ignition/suppression/ fragmentation | 2.5 arc minutes | 5-year intervals | Socioeconomic data and applications centre (SEDAC) |
| Percentage crop cover | PCC | Fragmentation | 5 arc minutes | Annual | HistorY Database of the Global Environment (HYDE 3.3) |
| Percentage pasture cover | PPS | Vegetation | 5 arc minutes | Annual | HistorY Database of the Global Environment (HYDE 3.3) |
| Precipitation seasonality | PS | Climate | 0.5° × 0.5° | Annual | Copernicus climate data store |
| Fire weather index | FWI | Climate | 0.5° × 0.5° | Monthly | Copernicus climate data store |
| Precipitation of the driest quarter | PPNQ | Climate | 0.5° × 0.5° | Annual | Copernicus climate data store |
| Number of dry days | NDD | Climate | 0.5° × 0.5° | Annual | Copernicus climate data store |
| Percentage grazeland cover | PGZC | Vegetation | 5 arc minutes | Annual | HistorY Database of the Global Environment (HYDE 3.3) |
| Percentage rangeland cover | PRC | Vegetation | 5 arc minutes | Annual | HistorY Database of the Global Environment (HYDE 3.3) |
| Annual average precipitation | AAP | Climate | 5 arc minutes | Annual | Copernicus climate data store |
| Gross primary productivity | GPP | Vegetation | 0.5° × 0.5° | Monthly | MOD17A1 |
| Aboveground biomass | AGB | Vegetation | 0.5° × 0.5° | Longterm average | |
| Percentage Tree cover | PTC | Vegetation | 250m | Annual | MODIS |
| Fraction of Absorbed Photosynthetically Active Radiation | FAPAR | Vegetation | 500m | Monthly (originally 8 days) | MODIS |


**Table 1: List of predictor variables that were considered in this study including their classifications, resolution (spatial**
**& temporal) and the respective data sources.**






## 2.3 Vegetation-related predictors

Thonicke et al. (2010), for example, discussed the crucial role of vegetation structure in shaping fire occurrence, spread and intensity. Consequently, our study considered eight vegetation predictor variables to comprehensively evaluate their role on global fire distribution. These variables encompass Percentage Grass Cover (PGC), Percentage Non-Tree Cover (PNTC), Percentage Crop Cover (PCC), Percentage Graze Cover (PGZC), Percentage Rangeland Cover (PRC), and Percentage Tree Cover (PTC).

PGC defines the land covered by grass, influencing fuel availability, while PNTC considers non-tree vegetation such as grass and shrubs, contributing to overall fuel dynamics. PCC reflects the presence of cultivated crops which have been found to suppress fire occurrence as they fragment the landscape acting and so act as a barrier to fire spread (Haas et al., 2022). Previous studies reported that landcover change has a significant contribution to wildfire distribution (Gallardo et al., 2016; Vilar et al., 2021). To understand the relationship between landcover and burnt area distribution, we incorporate PGZC, PRC, PTNC and PTC.

Numerous studies discussed the varying effects of vegetation parameters on fire events (Bowman et al., 2020; Kuhn-Régnier et al., 2021). Accordingly, Gross Primary Productivity (GPP), Aboveground Biomass (AGB), and Fraction of Absorbed Photosynthetically Active Radiation (FAPAR) were considered in this study as proxies for vegetation health and productivity.

## 2.4 Vegetation-related predictors

Topography can influence the occurrence and spread of fires especially in regions with complex terrain and microclimatic conditions (Blouin et al., 2016; Fang et al., 2015; Oliveira et al., 2014). To capture the impact of topography, some studies used slope (Cary et al., 2006) and surface area ratio (Parisien et al., 2011) in their models and reported topography to marginally contribute to wildfire dynamics. However, recent studies reported some significant contributions of topography to global burnt area distribution when using the topographic positioning index (TPI) (Haas et al., 2022). TPI is designed to encompass and evaluate the complex influence of terrain features, such as elevation and slope, on the distribution of burned areas. Thus, TPI goes beyond simplistic representations of landscapes and offers a more nuanced perspective on how terrain characteristics contribute to the occurrence and extent of wildfires. Given the role of terrain on fire behavior and propagation patterns, the inclusion of TPI in this study allows for a comprehensive examination of wildfire distribution.

## 2.5 Anthropogenic Influence Predictors

To comprehensively capture the impact of anthropogenic factors on both fire ignition and suppression, our study integrates three key predictors: Human Development Index (HDI), Population Density (PPN), and Road Density (RD). The inclusion of HDI aims to encapsulate human influence on ecological landscapes, thereby affecting the dynamics of both ignition and suppression processes. Although HDI itself may not directly relate to fire occurrence, it stands as a valuable socio-economic





indicator that significantly influences overall fire dynamics and management, like how Gross Domestic Product (GDP) has
been used in other fire models (Perkins et al., 2022). To address the limitations of using GDP as a proxy for human development
in predicting global fires, we opted for HDI. Previous research has utilized GDP for this purpose (Zhang et al., 2023), however,
GDP is an indicator of a country's economic performance (Callen, 2008). In contrast, HDI data is much broader as it captures
the country's social and economic development levels, making it a more suitable and consistent measure for our analysis. HDI
evaluates a country or other administrative region's development status based on the critical factors of life expectancy,
education, and income, providing a nuanced understanding of the socio-economic context shaping fire behavior (Teixeira et
al., 2023).

## 2.6 Weather-Related Predictors

To capture the impact of fire weather on the distribution of wildfires, we employed the Canadian Fire Weather Index (FWI),
renowned for its comprehensive framework integrating diverse meteorological parameters to evaluate potential fire behavior
and danger (de Jong et al., 2016). The FWI is widely adopted by fire management agencies facilitating informed decisions on
fire prevention, preparedness, and suppression strategies, and global context has been shown to correlate well with burned area
across the globe (Jones et al., 2022). We used the number of dry days (NDD) as a proxy for biomass production limitations.
While it falls in the category of weather-related fire predictors, in this study it's an indirect indicator of how moisture
availability can affect available combustible vegetation. We incorporated additional covariates capturing seasonal and annual
weather dynamics that influence fires, including Precipitation Seasonality (PS), and Annual Average Precipitation (AAP). The
selection of these predictors was informed by their significance in previous global fire modeling studies (Chuvieco et al., 2021;
Joshi and Sukumar, 2021; Le Page et al., 2008; Mukunga et al., 2023; Saha et al., 2019), as well as insights from seminal
works such as that by (Pechony and Shindell, 2010).

## 2.7 Data Processsing

We harmonized the spatial and temporal resolution of the predictor dataset to conform to our analytical framework, which had
a spatial resolution of 0.5° and a temporal resolution of one month. This involved employing techniques such as aggregation,
resampling, and consolidation. For instance, while the native temporal resolution of FAPAR was 8 days, we transformed it
into a monthly temporal resolution to align with our primary variable. Most predictors originally possessed an annual temporal
resolution, except for FWI, GPP, and FAPAR, which were also available every month. For annual predictors, we replicated
the same data for each month. Similarly, long-term variables like AGB, RD, and TPI were utilized every month to synchronize
with the shorter-resolution predictors. PPN, which was available at a 5-year interval, was used monthly over the represented
5-year span. Kuhn-Régnier et al. (2021) highlighted the important role of antecedent vegetation as key driver for global fires.
To evaluate the role of fuel accumulation from the previous year on the burnt area, we derived the Gross Primary Productivity
Index (GPPI) using monthly Gross Primary Productivity (GPP) data following Eq. (1). GPPI was originally defined as Monthly
Ecosystem Productivity Index (MEPI) in the work by Forrest et al. (2024). This index allowed us to quantify the relationship





between vegetation growth, fuel accumulation and subsequent fire activity, providing a more nuanced understanding of the factors influencing fire dynamics.

$$\text{GPP index} = \frac{GPP_m}{max(GPP_m, GPP_{m-1}, \ldots, GPP_{m-13,})} \tag{1}$$

Where $GPP_m$ is the month's GPP, and the denominator is the maximum GPP of the past 13 months. Furthermore, we calculated additional metrics including GPP12 (the mean gross primary productivity over the previous 12 months), (FAPAR12) (the mean fraction of absorbed photosynthetically active radiation over the past 12 months), and FAPAR6 (the mean FAPAR over the last 6 months). These metrics serve to capture average vegetation productivity, serving as refined indicators of fuel accumulation.

**2.8 Statistical modeling and final predictor choice**

To address variable collinearity, we conducted pairwise correlation analyses among predictor variables using the R statistical package (CoreTeam, 2014). Following established guidelines (Dormann et al., 2013), we applied the conventional threshold of R > 0.5 to enhance the model's efficiency. This helped us identify and exclude correlated variables from the analysis. Specifically, variables such as AGB, FAPAR12, FAPAR6, AAP, and RD were excluded due to their strong correlations with other variables (see Fig. 2). There were some correlated variables that were however returned in the model due to their significant contribution to fire modelling and model performance. For example NDD was strongly correlated to PTC ( ~ - 0.68), but keeping both increased the variance explained by the full model.





**Figure 2: Correlation matrix of all the variables that were considered for modelling in this investigation**

Moreover, we employed the Variance Inflation Factor (VIF) to evaluate collinearity among predictor variables, removing those with VIF values surpassing 5, as recommended by O'brien. (2007). Post collinearity tests, an additional 3 parameters were adopted to progressively select the best model, namely: 1) a model which comprise of a full suite of categories of covariate combinations (i.e vegetation, climate, topography, ignitions), 2) the deviance explained value and 3) the normalised mean



square error value as illustrated in the making of Burnt Area Simulator For Europe (BASE) (Forrest et al., 2024). Fig A3 shows
ten scatter plots of the final variable selection based on the optimum model, each depicting the relationship between the burnt
area fraction and different environmental or socio-economic variables. The variables include the GPPI, FWI, PNTC, HDI,
PTC, TPI, NDD and PPN.

A quasi-binomial GLM was selected for modeling BA due to its capability to handle non-Gaussian error distributions, seamless
integration into DGVMs and ability to generate partial residual plots (Bistinas et al., 2014; Haas et al., 2022; Lehsten et al.,
2016). Calibration of the model utilized data from 2002 to 2010 while testing utilized data from 2011 to 2018. Residual plots
were utilized to examine the magnitude and nature of each predictor's relationship with wildfire burnt area distribution.

Model performance was assessed using the Normalized Mean Error (NME) following Kelley et al., (2013). NME serves as a
standardized metric for evaluating global fire model performance, facilitating direct comparison between predictions and
observations. The NME was calculated following Eq. (2).

$$NME = \frac{\sum \ Ai \ I \ obs_i - sim_i \ I}{\sum \ Ai \ I \ obs_i - obs \ I} \tag{2}$$


The NME score was computed by summing the discrepancies between observations (obs) and simulations (sim) across all
cells (i), weighted by the respective cell areas (Ai), and then normalized by the average distance from the mean of the
observations. A lower NME value reflects superior model performance, with a value of 0 indicating a perfect alignment
between observed and simulated values. BA fractions were treated as a probability ranging from 0 to 1, following a quasi-
binomial distribution. We applied the logit link function based on the methodology outlined by Haas et al. (2022). After
conducting a collinearity test, the models were systematically evaluated using various combinations of predictor variables. A
total of 25 model runs were conducted, each incorporating different sets of variables while iteratively excluding some, to
discern the extent to which each predictor explained variance when others were not included (see Table 2). To evaluate the
reliability of the predicted interannual variability and seasonal cycles, we applied a regression function to determine the
relationship ($R^2$) between the observed and predicted trends. An $R^2$ of 1 shows good performance in our predictions and an $R^2$
of 0 shows poor performance in our predictions. To assess the trend in predicted interannual variability, we used the Mann-
Kendall test (Kendall, 1975; Mann, 1945). This widely used method detects monotonic trends in environmental data. Being
non-parametric, it works for all distributions, does not require normality, but assumes no serial correlation. We extracted the
trend test results and plotted a map of trend distributions across different GFED regions to identify areas with significant
predicted trends (P<0.05) from those with non-significant trends.





## 3 Results

### 3.1 Optimal model selection and GLM results

Table 2 provides a list of results from our progressive inclusion of model covariates as we aimed to identify the optimum model. The initial models (model 1 to model 3) progressively include more variables, however, a noticeable jump in deviance explained when PNTC is added (Model 3: 0,5298). Models 4 to 8 involve adding vegetation (FAPAR) and various land use types (PCC, PPS, PRC, PGC). This is accompanied by marginal improvement in deviance explained, indicating these factors provide some additional predictive power but are not as impactful as existing vegetation covariates (such as GPP). Models 10 to 12 introduce polynomial terms for PTC. This results in an increase in deviance explained, peaking at around 0.558836 in Model 12. Models 13 to 16 incorporate interactions between HDI and land use types (e.g., PCC and PRC), resulting in marginal increase in deviance explained with the highest recorded in Model 15($\sim$ 0.5664789). Models 19 to 25 fine-tune the overall performance by incorporating various variables and their interactions. Model 24, which includes a comprehensive set of climatic, vegetation, human, and topographic variables along with their interactions, achieves the highest deviance explained ($\sim$0.5720048). The marginal improvements observed in subsequent models indicate that while additional variables contribute to the model, the primary influencing factors were already identified by Model 19, however it was not the simplest model, and comprised of other variables that we don't have future projections for (e.g RD). Therefore, Model 25, which offers a balance of parsimony, simplicity, high deviance explained, and low NME, was selected as the best model in this analysis.

| Model | Formulae | Deviance explained | NME |
|---|---|---|---|
| model 1 | glm(burnt ~ FWI + GPP + HDI + PTC + RD) | 0.3548030 | 0.7472088 |
| model 2 | glm(burnt ~ FWI + GPP + HDI + PTC + RD + PGC) | 0.3699393 | 0.7495652 |
| model 3 | glm(burnt ~ FWI + GPP + HDI + PTC + RD + PNTC) | 0.5298061 | 0.7208771 |
| model 4 | glm(burnt ~ FWI + GPP + HDI + PTC + RD + PNTC + FAPAR) | 0.5312036 | 0.7188448 |
| model 5 | glm(burnt ~ FWI + GPP + HDI + PTC + RD + PNTC + FAPAR + PCC) | 0.5312697 | 0.7191269 |
| model 6 | glm(burnt ~ FWI + GPP + HDI + PTC + RD + PNTC + FAPAR + PCC + PPS) | 0.5328183 | 0.7195616 |
| model 7 | glm(burnt ~ FWI + GPP + HDI + PTC + RD + PNTC + FAPAR + PCC + PRC) | 0.5313813 | 0.7193946 |





| model 8 | glm(burnt ~ FWI + GPP + HDI + PTC + RD + PNTC + FAPAR + PCC + PGC) | 0.5349288 | 0.7190611 |
|---|---|---|---|
| model 9 | glm(burnt ~ FWI + GPP + HDI + PTC + RD + PNTC + FAPAR + PCC + FAPAR12 + PGC) | 0.5359802 | 0.7181930 |
| model 10 | glm(burnt ~ FWI + GPP12 + HDI + poly(PTC, 2) + PNTC + FAPAR + PCC + FAPAR12 + PGC + PPN ) | 0.5295939 | 0.7172668 |
| model 11 | glm(burnt ~ FWI + GPPI + HDI + poly(PTC, 2) + RD + PNTC + FAPAR12 + PGC + PS) | 0.5579946 | 0.7193546 |
| model 12 | glm(burnt ~ FWI + GPPI + HDI + poly(PTC, 2) + RD + PNTC + FAPAR12 + PS) | 0.5571164 | 0.7192122 |
| model 13 | glm(burnt ~ FWI + GPPI + HDI*PCC + PGC + RD + poly(PTC, 2) + PNTC + PS) | 0.5569187 | 0.7214560 |
| model 14 | glm(burnt ~ FWI + GPPI + HDI*PGC + RD + poly(PTC, 2) + PNTC+ PS) | 0.5570586 | 0.7222061 |
| model 15 | glm(burnt ~ FWI + GPPI + HDI*PRC + RD + poly(PTC, 2) + PNTC + PS) | 0.5664789 | 0.7154708 |
| model 16 | glm(burnt ~ FWI + GPPI*PNTC + HDI + RD + poly(PTC, 2) + PS ) | 0.5563012 | 0.7215202 |
| model 17 | glm(burnt ~ FWI + GPPI + HDI + RD + poly(PTC, 2) + PNTC + PS + NDD) | 0.5681926 | 0.7191069 |
| model 18 | glm(burnt ~ FWI + GPPI + HDI + RD + poly(PTC, 2) + PNTC* PS + NDD + TPI) | 0.5711503 | 0.7167015 |
| model 19 | glm(burnt ~ FWI + GPPI + HDI + RD + poly(PTC, 2) + PNTC* PS + NDD + PGC + FAPAR12) | 0.5709692 | 0.7175149 |
| Model 20 | glm(burnt ~ FWI + GPPI + HDI + poly(PTC, 2) + PNTC* PS + NDD + FAPAR12) | 0.5677209 | 0.7182814 |
| Model 21 | glm(burnt ~ FWI + GPPI + HDI + poly(PTC, 2) + RD + PNTC*PS + NDD + TPI + PPN | 0.5714474 | 0.7170576 |
| Model 22 | glm(burnt ~ FWI + GPPI + HDI + poly(PTC, 2)+ PNTC*PS + NDD + TPI + PPN | 0.5705348 | 0.7177887 |
| Model 23 | glm(burnt ~ FWI + GPPI+ HDI + poly(PTC, 2) + RD + PNTC*PS + NDD+ TPI+ PPN | 0.5714474 | 0.7170576 |
| Model 24 | glm(burnt ~ FWI + GPPI + HDI + poly(PTC, 2) + RD + PNTC*PS + NDD + TPI + PPN + AAP | 0.5720048 | 0.7173093 |
| **Model 25** | **glm(burnt ~ FWI + GPPI + HDI + poly(PTC, 2) + PNTC + PPN + NDD + TPI** | **0.5682776** | **0.7186160** |





| Model 26 | glm(burnt ~ FWI + GPPI + HDI + poly(PTC, 2)* NDD + NTC + PPN + NTC + TPI | 0.5687439 | 0.7194855 |

**Table 2: Results of modeling attempts using different combinations of predictor variables using a progressive inclusion of covariates approach.**

Our results reveal that each predictor variable incorporated in the final analysis significantly predicted the distribution of wildfires ($p < 0.05$), as outlined in Table 3.

Table 3. Summary of GLM coefficients for the final model, presenting t-values and p-values for predictors. The results indicate that all predictors in the final model were statistically significant about wildfire distribution ($p < 0.05$).

|  | Estimate | Std.Error | T value | Pr(>\|t\|) |
| --- | --- | --- | --- | --- |
| (Intercept) | - 6.159e+00 | 2.349e-02 | -262.17 | <0.00001 |
| FWI | 9.296e-01 | 1.948e-03 | 477.28 | <0.00001 |
| GPPI | -2.270e+00 | 8.974e-03 | -252.96 | <0.00001 |
| HDI | -1.680e+00 | 1.235e-02 | -135.99 | <0.00001 |
| PNTC | 5.170e-02 | 2.270e-04 | 227.78 | <0.00001 |
| poly(PTC,2)1 | 2.135e+03 | 1.114e+01 | 191.55 | <0.00001 |
| poly(PTC,2)2 | -9.783e+02 | 6.975e+00 | -140.27 | <0.00001 |
| TPI | 2.225e-01 | 3.946e-03 | 56.39 | <0.00001 |
| NDD | -9.550e-03 | 4.757e-05 | -200.78 | <0.00001 |
| PPN | -1.075e-03 | 1.808e-05 | -59.48 | <0.00001 |

Our analysis results revealed the relationship between various predictors and BA distribution, as depicted in Fig. 3. Among the predictors studied, FWI, PNTC, PTC and TPI showed a positive relationship with BA distribution. Notably, FWI and PNTC showed particularly strong relationships, underscoring the substantial role of fire weather, fuel availability on the expansion of BA extent. Conversely, several predictors showed a negative relationship with BA distribution, including the GPPI, HDI, PPN and NDD. A polynomial of PTC shows a slightly bell-shaped relationship with burnt area fraction.

Overall, our observations highlight the critical role of factors such as fire weather, fuel availability, vegetation cover, climate conditions, and landscape characteristics in shaping BA distribution patterns. Fig.3 visually represents the differential





relationship of these predictors on BA distribution, offering a comprehensive overview of the underlying mechanisms driving
wildfire dynamics.

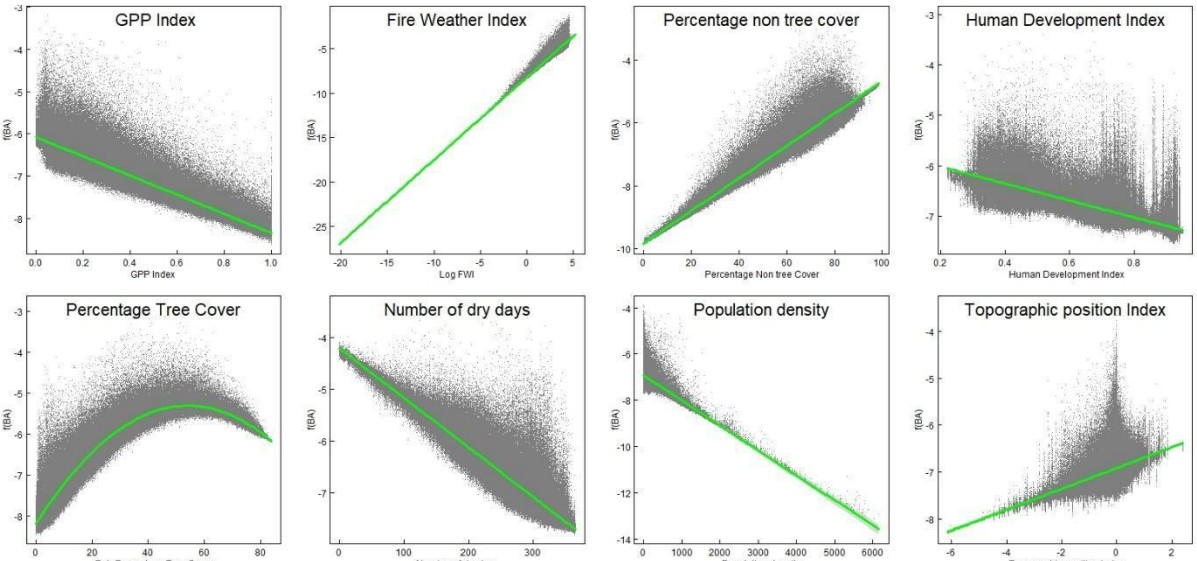


**Figure 3: Partial Residual Plots illustrating the relationship between Burned Areas (BA) and the eight final predictor variables. These plots show the effect of each predictor while the others are held constant (Larson and McCleary 1972) Predictor variables were Gross Primary Production Index (GPP), Fire Weather Index (FWI), Percentage Non-Tree Cover (PNTC), Human Development Index (HDI), Percentage Tree Cover (PTC), Topographic Position Index (TPI), Population Density (PPN) and Number of Dry Days (NDD).**

306

The model demonstrated strong performance in predicting BA, accounting for over 50% of the variability in burnt areas (Deviance explained = 0.568). While our results slightly lagged those of a global fire distribution model by Haas et al. (2022), who found a deviance explained of 0.69, it's noteworthy that their model incorporated a broader array of variables (16 predictors) and operated at a coarser temporal resolution (annual). Our model's performance, based on eight predictors and operating at a finer temporal resolution (monthly), is considered satisfactory and parsimonious.

Assessment of model accuracy yielded an NME of 0.718, indicating a generally close correspondence between observed and predicted burnt area patterns (see Fig. 4). This level of accuracy is comparable to that reported by previous global fire models, such as (Haas et al., 2022) and (Hantson et al., 2016), which reported NMEs ranging from 0.60 to 1.10.

Spatially, our model effectively captured the distribution of BA in the tropics and the southern hemisphere, demonstrating notable similarities between observed and predicted burnt area fractions on an annual basis (see Fig. 4). However, in



extratropical regions, particularly in the northern hemisphere, instances of over-prediction were observed. This discrepancy is
evident in the inconsistencies between observed annual distribution patterns and those predicted by the model.

**Observed burnt area (GFED5)**

**Predicted burnt area**


**Figure 4: Annual burnt area fraction distribution map with the observed burnt area (top) and predicted burnt area**
**(bottom).**





## 3.2 Interannual variability

Our analysis results revealed a substantial global decrease in burnt areas exceeding 1 million square kilometers from 2002 to 2018, with the peak decline observed in 2004 (see Fig. 5). This downtrend was consistently observed in both the actual and predicted extent. Notably, the projected trend exhibited a steeper decline compared to the observed trend, indicating a potential underestimation of inter-annual variability by the model. However, it aligns with the decreasing patterns reported in earlier studies (Andela et al., 2017; Jones et al., 2022). Excluding and holding HDI constant in the model made the projected trend remain steady, suggesting the role of anthropogenic developments driving a downward trend in wildfire distribution.

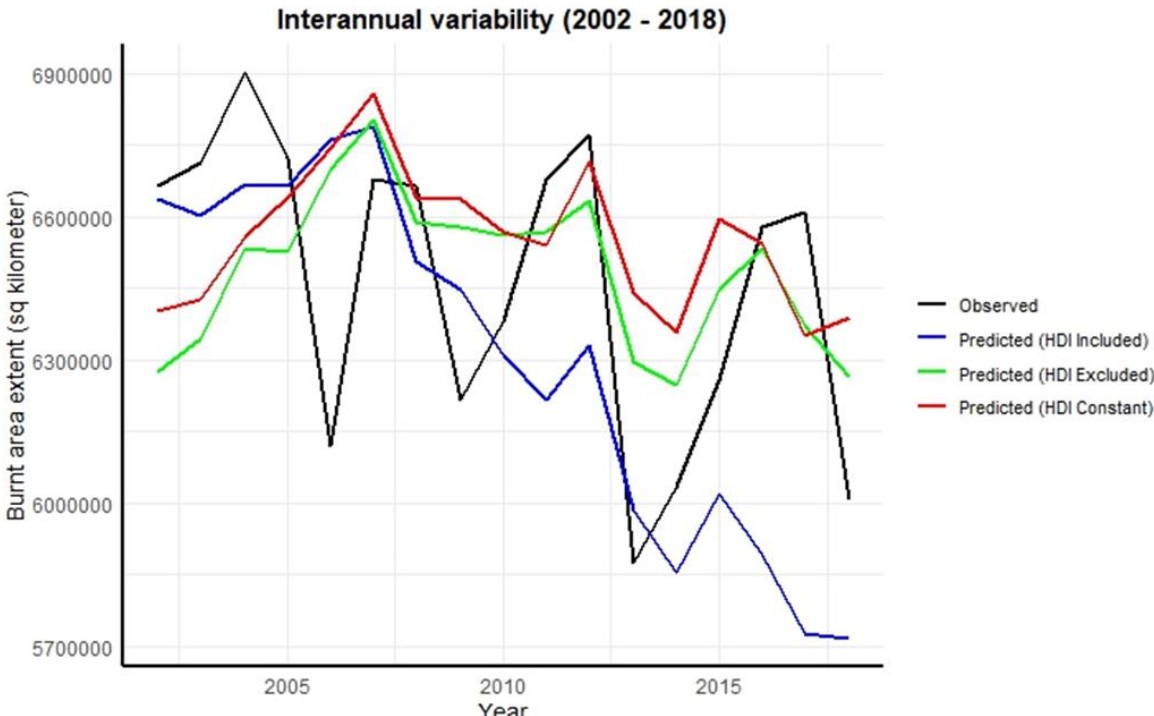

**Figure 5: Interannual variability in burnt area extent showing the observed trend (based on GFED detections) and predicted trends. Included are interannual trends when HDI was excluded and held constant from the value of the first year in the model.**

The Mann Kendall trend analysis further shows significant variation in the magnitude and direction of predicted burnt area extent across the 14 GFED regions (refer to Fig. 6 and Table A1). Five regions (SHAF, SHSA, NHAF, CEAS) predicted a significantly positive trend ($p < 0.05$) in burnt area extent, while the other regions predicted no significant trends (NHSA, SHSA, MIDE, TENA, AUST, EURO, EQAS, CEAM, BONA, BOAS). Overall, the projected positive trend predominated in GFED regions situated in central and southern Africa, and central and southern Asia. In contrast, the Americas, Australia, and Europe demonstrated no significant trend, as illustrated in Fig. 6.



343

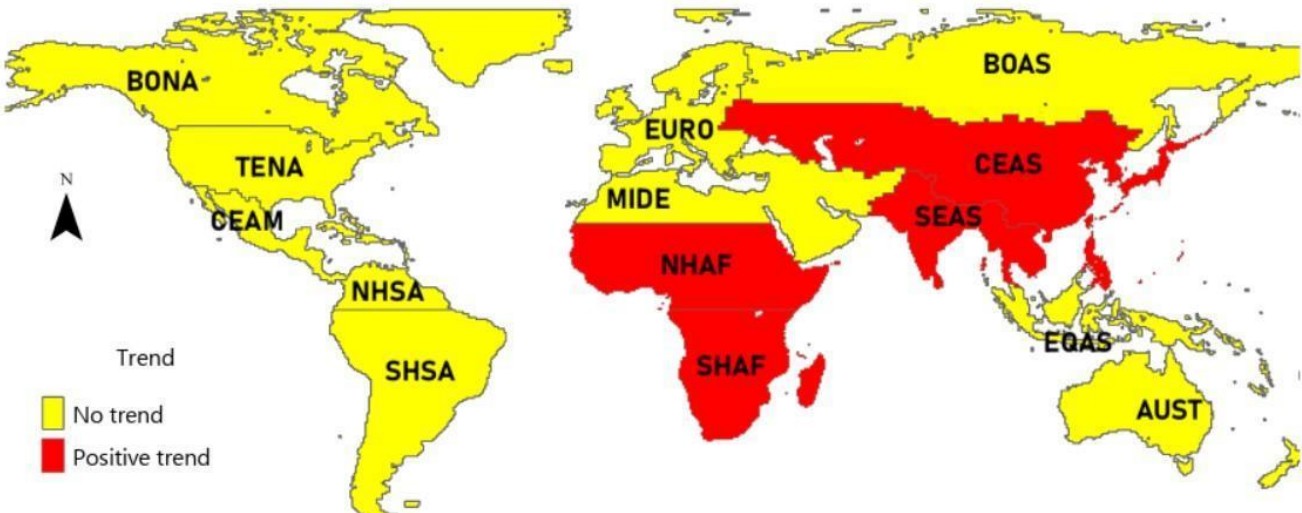

344

**Figure 6: Variation in the trend of interannual variability for burnt areas across different GFED regions.**

346

Our predictive model performed poorly in predicting interannual variability as exhibited by a poor strength of relationship between the predicted trend when compared to the observed ($R^2= 0.24$) (See Fig 7 and Fig A1). This poor relationship was exhibited across most of the GFED regions ($R^2 < 0.50$), except for the NHSA which showed strong similarities between the predicted trend and observed trend ($R^2 = 0.55$). This observation suggests that the combination of covariates that we incorporated in this model has limited strength in capturing global interannual variability in burned area. However, the predicted global trend is in sync with previously reported global trends (Jones et al., 2022).

353



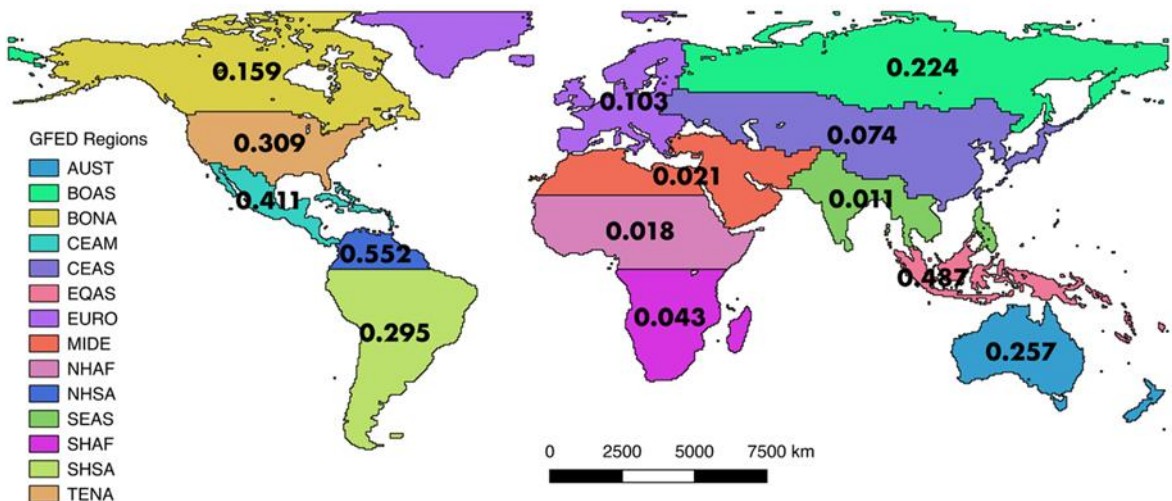

354

**Figure 7: Spatial distribution of r-square values for the relationship between observed and predicted interannual variability per GFED region.**

**3.3 Seasonal Cycle**

Our analysis results show that a global extent of BA shows an alternating seasonal cycle with strong peaks in February and August (see Fig. 8). The predicted pattern slightly underestimates the burnt area, however, appears to be closely knit with the observed trend ($R^2 = 0.54$). Like the global interannual trends, the strength of similarity between observed and predicted seasonal cycles varies according to the GFED region with $R^2$ ranging between 0.06 to 0.99 (refer to Fig. 9). The model predicted better in GFED regions that are situated in Southern Africa, South America, Australia and Asia ($R^2 > 0.50$) (see Fig. 9 and Fig A2). In contrast, poor seasonal predictions were recorded in GFED regions situated in North America, North Africa and Europe as indicated by a poor relationship between observed burnt area and predicted burnt area ($R^2 < 0.50$).

365



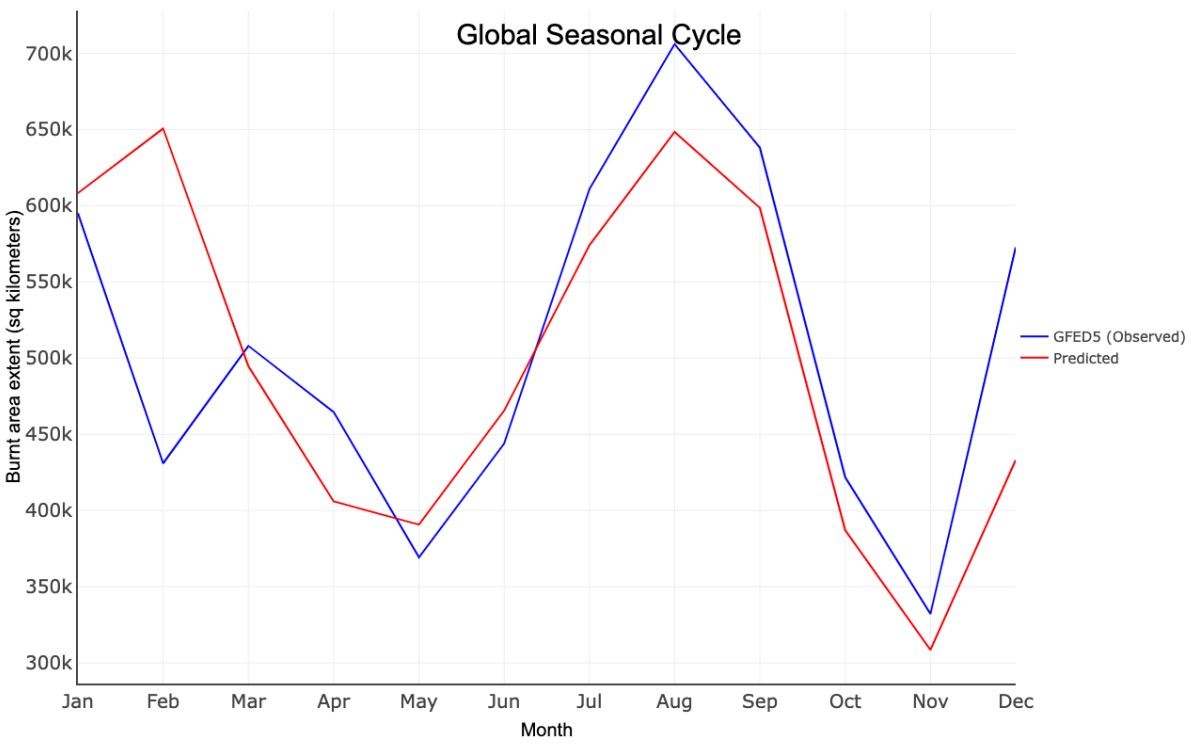

**Figure 8: Global seasonal burnt area patterns showing the observed (GFED 5) and predicted burnt area extent.**

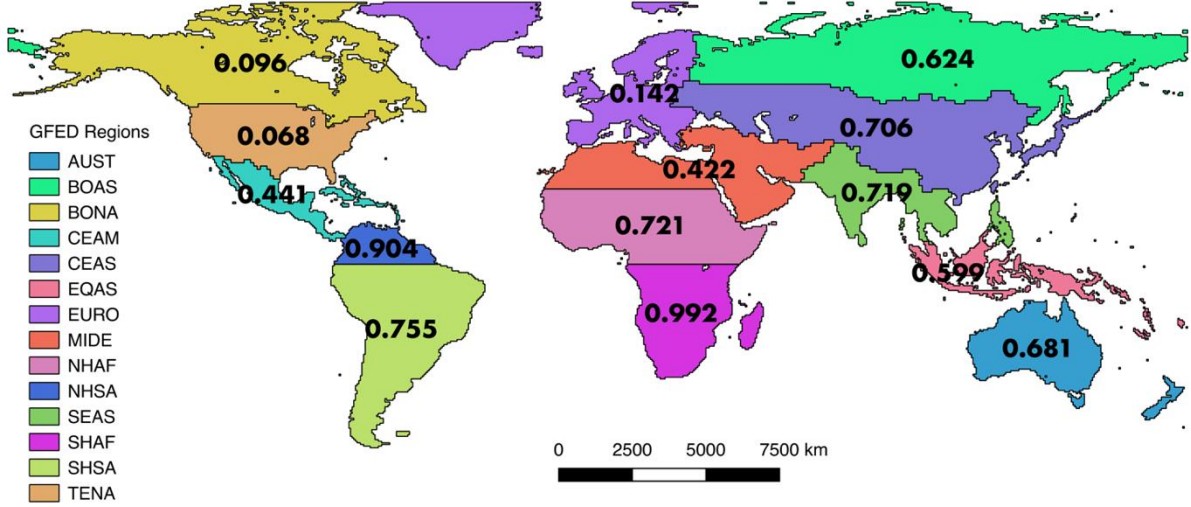

**Figure 9: Spatial distribution of r-square values for the relationship between observed and predicted seasonal variability per GFED region.**





## 4 Discussion

Wildfires are common phenomena whose dynamics may pose relevant impacts on the ecology of species and humans across different biomes. The evolving dynamics of wildfires are anticipated to undergo significant changes in the future due to global environmental shifts. In this study, we sought to tease apart statistical relationships between biophysical and socioeconomic drivers of wildfire dynamics and burned areas to facilitate DGVM integration and reliable prediction of future wildfire dynamics.

### 4.1 Main drivers of global burned area

We found a DGVM compatible parsimonious global statistical model made of FWI, PNTC, PTC, TPI, GPPI, HDI, PPN and NDD. Of all the key variables, FWI and PNTC exhibited a strong positive relationship with fire occurrence, underscoring the importance of conducive fire-weather conditions and combustible fuel in driving wildfire occurrence and spread. High PNTC is most likely related to high amounts of flammable vegetation, such as grasses and shrubs. Our findings that fire weather (~FWI) and fuel availability (~PNTC) influence burned area extent align with previous studies (Andela et al., 2017; Bistinas et al., 2014; Forkel et al., 2019b; Kuhn-Régnier et al., 2021). The other studies, however, did focus on annual burned area, not the seasonal cycle, which is also crucial to adapt to changes in fire risk.

Our results show that higher PNTC leads to higher burnt area fractions. In contrast, areas with lower non-tree cover show lower burnt area fractions. Areas with high PNTC typically consist of grasses and shrubs (~ height < 2 m), while areas with low PNTC are often characterised by trees. Grasses and shrubs often encourage frequent burning much more than trees (Juli et al., 2017; Wragg et al., 2018). Conversely, low PNTC indicates high tree cover, which is often less flammable, leading to fewer fires. Though our findings support previous literature indicating that regions with abundant combustible vegetation and favorable fire-weather conditions are prone to frequent burning (Kraaij et al., 2018; Thonicke et al., 2010), we observed a surprising negative relationship between NDD and burnt area. Previous studies found a positive relationship between NDD and burnt area fractions (Haas et al., 2022), similar to our single factor plots of NDD and burnt area in Fig A3. This result most probably shows that relationships derived with annual data, as in the other studies mentioned here, cannot simply be transferred to seasonal fire predictions. Studies have shown that the effect of dryness on fire varies depending on vegetation communities in Mediterranean ecosystems (Cardil et al., 2019). Stott (2000) echoed similar sentiments for tropical environments, indicating the complex relationship between vegetation, dryness and fire. Our efforts to investigate this complex relationship through an interaction term did not significantly improve our model accuracy (~ model 26). Hence, future studies may benefit from further exploring the complex relationship between dryness, vegetation at a global scale, particularly the effect of incorporating polynomial terms on correlated predictors in a linear model.



Our findings revealed that HDI, GPPI and PPN are negatively associated with trends in global fire extent. Specifically, the
negative relationship between HDI and burnt area implies that technological advancements, improved surveillance systems,
and effective mitigation efforts play a significant role in limiting the extent of burned areas. Contrary to expectations based on
Haas et al. (2022), PPN, which should correlate with more ignitions, does not appear to increase the burnt area extent (see Fig.
3). In fact, we observed that lower PPN corresponded to larger burnt areas, likely due to the impact of human activities on
landscape fragmentation through road construction, and measures to suppress fires in human inhabited spaces to protect
properties (Kloster et al., 2010). Saunders et al. (1991) observed that the response of fire to changes in PPN is governed by
two opposing processes, an increase in population leads to more ignition sources, while simultaneously prompting greater fire
management efforts to suppress fires. They further highlighted that fire suppression rates are highest in densely populated
areas. This suggests that the scale (both spatial and temporal) of analysis may influence the nature and extent to which PPN
affects burnt area extent. Our results for the effect of PPN have important implications for DGVMs and land surface models.
These models differ widely in the assumed effect of PPN, often using a unimodal response (Teckentrup et al., 2019). However,
many DGVMs simulate BA annually, in some cases distributing the wildfires across seasons in a second step, using rather
simplified assumptions (Teckentrup et al., 2019). Similarly, we anticipated a positive relationship between GPPI and burnt
areas, as GPPI is indicative of ecosystem dryness and flammability. However, our findings revealed a negative relationship,
indicating that other factors may be influencing the connection between GPPI and the extent of burnt areas. Our findings are
inline with that of Forrest et al. (2024) who initially investigated the effect of this index on burnt areas in Europe. Unlike
previous global studies that utilized annual GPP, our research employed a more refined measure, GPPI. Future research could
benefit from evaluating the relationships between GPPI and burnt areas in other GFED regions and temporal scales.

### 4.2 Spatial variation in model performance

Our model exhibits stronger performance in predicting the spatial distribution of fires in southern Africa, Australia, and South
America than in other world regions, potentially due to the seasonal patterns of key predictors in these regions, which can be
effectively captured using linear functions. Conversely, our model tends to overpredict fires in the northern hemisphere,
particularly in North and Central America, as well as Asia. Annual burned area variability is relatively high in Asia, Europe,
and Central America, which might make it more difficult to predict it (Chuvieco et al., 2021). Thus, our findings build upon
existing models on global burned area distribution. What sets our model apart from previous models is its ability to reliably
identify global seasonal fire distribution patterns. This simplicity offers a notable advantage, as it facilitates more nuanced
interpretation and implementation of DGVMs compared to annual models.

### 4.3 Attribution of global trends

Our model has contributed novel insights to the existing understanding of the factors influencing global fire trends (Joshi and
Sukumar, 2021; Kraaij et al., 2018; Mukunga et al., 2023). Previous research, such as the work by Andela et al. (2017),





primarily attributed the decline in global burnt areas to agricultural expansion and intensification. Earl and Simmonds (2018)
supported this view, adding that increased net primary productivity in Northern Africa also played a significant role. However,
our results suggest that human development is a more important driver than agricultural expansion alone. Despite the
conventional emphasis on agricultural factors, our attempt to incorporate cropland and rangeland fractions as predictor
variables did not substantially enhance our understanding of this trend (model 5 -10, Table 2). Interestingly, our analysis
revealed that excluding the HDI from our model and holding it constant to the value of the first year predicted a steady trend
that deviates from the observed negative trend in global fire extent and including HDI follows a decreasing trend that aligns
with the observed trend (Fig. 5). This highlights the significant influence of HDI in projecting the purported negative global
fire trend. The HDI is related to factors like advancements in fire control methods, surveillance, technology, and outreach
strategies increasing awareness, particularly in response to the growing human technological developments. Although these
strategies are often implemented independently and on a smaller scale, their cumulative impact on global fire trends is
substantial. Therefore, our model underscores the necessity for global initiatives aimed at enhancing fire control measures
through comprehensive awareness campaigns, capacity-building efforts, resource mobilization, and the development and
deployment of reliable surveillance technologies. By addressing these factors collectively, we can effectively mitigate the
extent and severity of global wildfires, thereby safeguarding ecosystems and human livelihoods.

**4.4 Interannual variability**
Despite demonstrating the significant role of the HDI in predicting global fire trends, our model struggled to achieve high
precision in forecasting interannual variability both globally (see Fig. 5) and within specific GFED regions (see Fig. S1).
Recognizing that this limitation might stem from an inadequate representation of vegetation (fuel) dynamics, we incorporated
FAPAR12 in models 9 to 12 (Table 2) and GPPI in models 11 to 26 (Table 2). Unfortunately, these adjustments did not enhance
our ability to predict the interannual variability of wildfires. Studies have found a relationship between increased precipitation
in the years preceding the fire season and fire activity in the drier savanna regions of Southern Africa (Shekede et al., 2024).
Hence, we also explored the role of previous fuel accumulation on subsequent fire seasons using GPP12 in model 10,
respectively. While this approach did not improve global interannual predictions, it showed a slight enhancement in deviance
explained (from 0.5357 to 0.5461). This improvement might have been confounded by the effects of the fire-aerosol positive
feedback mechanism in Africa (Zhang et al., 2023) and periodic El Niño conditions, which can affect rainfall patterns and lead
to drier vegetation conditions, reducing the predictability of fire occurrence, especially with linear models (Shikwambana et
al., 2022). Other attempts at simulating global annual burned area changes, for example with fire-enabled DGVMs (Fire Model
Intercomparison Project (FireMIP)), LPJ-GUESS-GlobFIRM yielded similarly poor model performance concerning
interannual variability (Hantson et al., 2020). Our modeling efforts highlight the complexity of accurately predicting wildfire
trends and underscore the need for future research to identify covariates that more effectively capture the interannual variability
of fires at a global scale.



### 4.5 Fire seasonality

The findings of this study exhibit robustness in capturing seasonal cycles ($R^2 = 0.536$), facilitated by the inclusion of monthly variables such as the GPPI and the logarithm of FWI, which are pivotal in delineating seasonal fire patterns. While the seasonal predictions demonstrated reliability across most GFED regions globally, notable exceptions were observed in North America, North Africa, and Europe ($R^2 < 0.50$). This discrepancy could be attributed to the intricate climatic conditions inherent to these regions, which influence fire weather in a manner that eludes simple linear modeling. Given the parsimonious design of our model, with only ~eight predictors, we think that the model performance is acceptable. For certain regions, it might be possible to increase model performance by implementing further region-specific predictors and relationships. Accurate predictions regarding the seasonal dynamics of diverse GFED regions can facilitate the identification of temporal windows when fires are prevalent, thereby furnishing valuable insights for simulating carbon emissions in DGVMs.

Globally, our model predicts a notable peak in burned areas during February and August. The February peak corresponds to dry conditions and fuel accumulation in regions such as NHSA, NHAF, and MIDE. In contrast, the August peak primarily emanates from tropical regions characterized by distinct seasonal patterns, particularly in SHSA, SHAF, and AUST. Here, the dry season augments the combustibility of accumulated fuel from the preceding wet season, facilitating fire spread. This observation corroborates earlier studies in the southern hemisphere, which underscore the prevalence of wildfires during prolonged dry spells (Magadzire, 2013; Shekede et al., 2024; Strydom and Savage, 2017). Increased temperatures and desiccated vegetation substantially enhance the likelihood and severity of wildfires during the dry season. Conversely, the onset of the rainy season precipitates a marked reduction in the occurrence of wildfires in these regions. This underscores the enduring influence of fire weather and vegetation dynamics as principal drivers of seasonal burnt area cycles, with factors such as moisture content in vegetation and soil, as well as humidity, playing pivotal roles in modulating ignition and fire extent within ecosystems. The seasonal global forecasts generated by our model hold significant implications for guiding adaptive strategies, fire management and prevention.

### 4.6 Excluded drivers of burned area

Several covariates initially considered, such as landcover variables (~PCC, PPS, PRC, PGC), vegetation (~FAPAR) and socioeconomic (~RD), did not make it to the final model (See Table 2) despite their potential relevance identified in previous studies(Forkel et al., 2019b; Hantson et al., 2015; Knorr et al., 2014; Pausas and Keeley, 2021; Perkins et al., 2022). The differences to our findings are related to differences in the statistical or modelling approach and the fact that most of these studies addressed annual BA patterns, not seasonal variations. Nevertheless, these other factors can clearly also be important for understanding fire dynamics, e.g. influencing fuel availability, landscape structure, and ignition sources. For instance, grazing lands can significantly impact fire behavior by altering fuel types and continuity, with areas used for grazing potentially reducing fuel loads (Davies et al., 2010; Strand et al., 2014). Similarly, FAPAR indicates vegetation health and productivity,




affecting fuel moisture content and thus fire risk (Pausas and Ribeiro, 2013). However, these factors are apparently indirectly
represented by the final model, as they are correlated to the driver variables in the final model. FAPAR, for example, is highly
correlated with GPP. Furthermore, RD is associated with human-caused ignitions and fire suppression capabilities (Forkel et
al., 2019b). However, it was excluded here because its contributions were already effectively represented by HDI and PPN,
which capture broader socioeconomic conditions and infrastructure impacts. Apart from that, Haas et al. (2022) observed a
shift in the direction of contribution for covariates when PPN and RD are used together. Considering that we may not have
future projections for RD unlike PPN, including the issue of collinearity, we decided to retain only PPN in our model.
Furthermore, our attempt to include RD in our models 21, 23 and 24 (Table 2) yielded marginal improvements which were
not different from when we excluded it in model 25. Overall, the decision to exclude most of these covariates was aimed at
reducing redundancy and multicollinearity, ensuring a balance between model complexity and predictive power. By focusing
on more comprehensive variables with high explanatory power, the final model achieves robust explanatory power. However,
the often-small differences in the deviance explained and the NME between different models imply that vegetation-fire
modellers might also pick a slightly different set of variables for DGVM integration without using much predictive power.

**4.7 Shortcomings and Recommendations**

The findings of this study offer valuable insights into the underlying drivers and patterns shaping global fire dynamics. While
our research represents relevant efforts in developing a streamlined model capable of accurately capturing seasonal variations
in global fire distribution, it's important to acknowledge certain limitations. The selection of covariates and the statistical model
was constrained by the necessity for integration within DGVMs applied to predict future dynamics, potentially omitting some
previously identified key predictors (~lightning frequency, gridded livestock densities) and modeling techniques (~ Random
Forest, Neural networks, XgBoost, CatBoost) for global fires(Forkel et al., 2019b; Joshi and Sukumar, 2021; Mukunga et al.,
2023; Zhang et al., 2023). This might contribute to observed shortcomings in our model's ability to predict spatial fire
distribution in certain regions and to capture interannual variability across many parts of the world. Future investigations
should aim to explore the inclusion of other established predictors and methodologies in global fire modeling once they become
easily compatible with DGVM integration. Despite these challenges, our study possesses intrinsic value, and the developed
model stands as a relatively simple tool for informing global seasonal fire predictions.

**5. Conclusions**

Global fire patterns undergo constant changes influenced by fluctuations in vegetation, weather conditions, topography, and
anthropogenic factors. Despite numerous attempts in previous studies to describe global fire distribution, the development of
a concise model capable of explaining and predicting global fire patterns, particularly one that seamlessly integrates with
DGVMs, is essential for a reliable assessment of the impacts of global change. In this study, we present a parsimonious
statistical model specifically tailored for global seasonal burned areas, with the goal of integration into DGVMs.




Our findings highlight the significance of socio-economic advancements, particularly those improving fire management strategies, as evidenced by the negative trend in global fire extent predicted through the inclusion of the HDI as a crucial socio-economic predictor in our model. Additionally, fire weather and vegetation dynamics, specifically the FWI and the novel GPP index, emerged as robust predictors of seasonal global fire patterns. While our parsimonious model exhibited limitations in predicting the interannual variability of global fires, it demonstrated commendable accuracy in forecasting the spatial and seasonal distribution of wildfires. We hope that our research outcomes will stimulate a more rigorous implementation of global fire models within DGVM frameworks. This, in turn, will contribute to a deeper understanding of the intricate dynamics of global fire patterns and enhance our capacity to effectively manage and mitigate the consequences of evolving fire regimes in the face of ongoing global changes.

**Code and data availability**

The code used in this analysis model fitting, and plotting is available at https://doi.org/10.5281/zenodo.14177016. Data used for model fitting are available at https://doi.org/10.5281/zenodo.14110150.

**Author contribution**

BK contributed to conceptualisation of the model and data analysis and model fitting. MF and TH supported in developing the statistical framework and interpreting the results. BK drafted the manuscript, with input from MF and TH.

**Acknowledgements**

This project has received funding from German Research Foundation (DFG) grant "Fire in the Future: Interactions with Ecosystems and Society (FURNACES) project" under grant number HI 1538/14-1. BK received a salary from FURNACES.

**Competing interests**

The authors declare that they have no known competing financial interests or personal relationships that could have influenced the work reported in this paper.




## Appendices


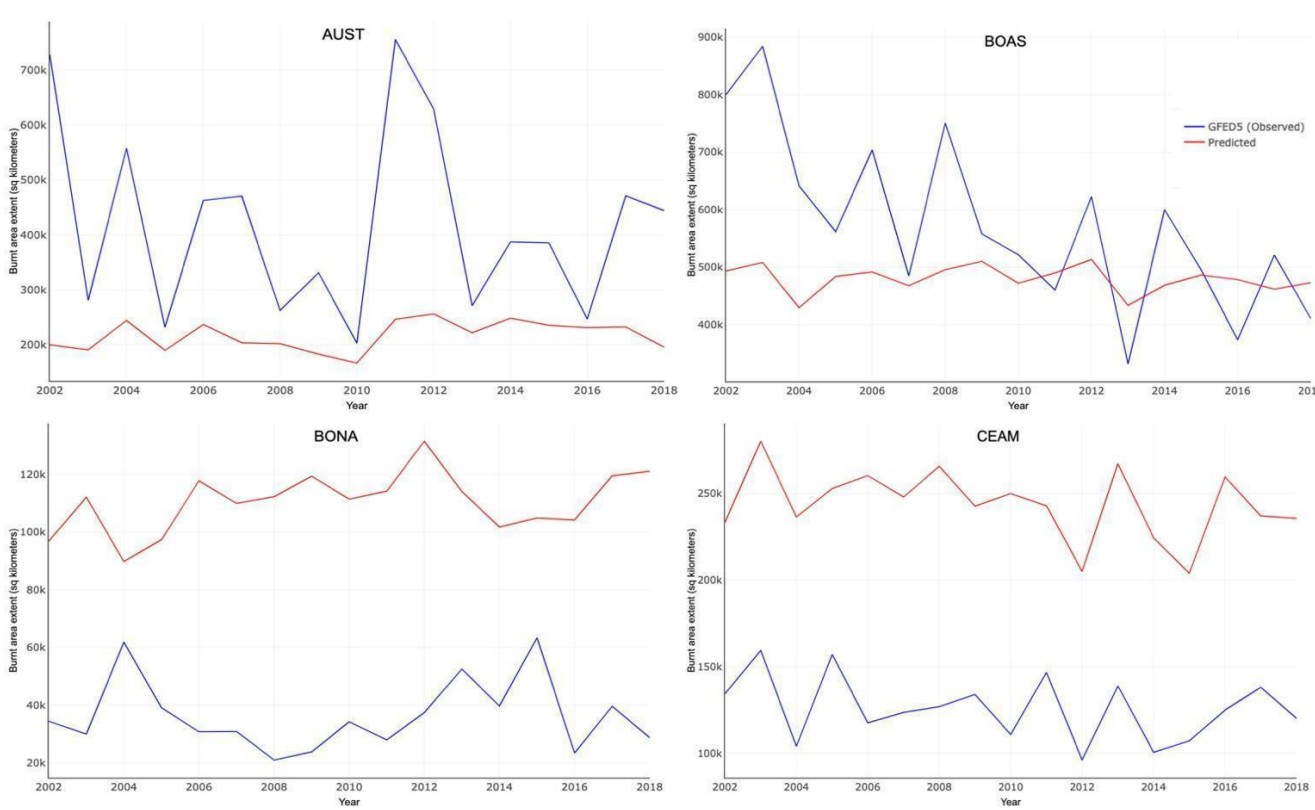










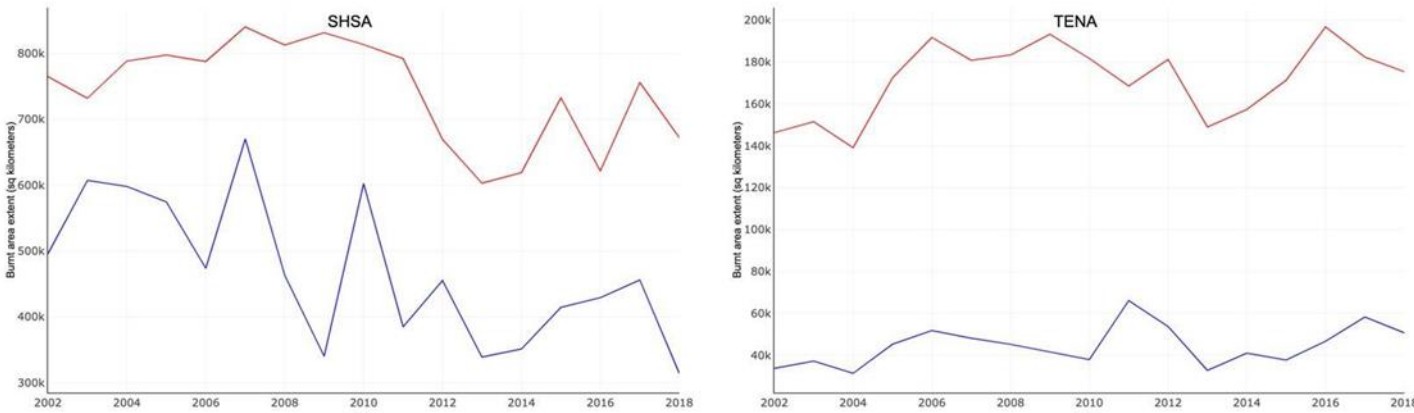

**Figure A1: Shows the observed (in red) and predicted (in blue) interannual variability in burnt area fractions across different GFED regions.**

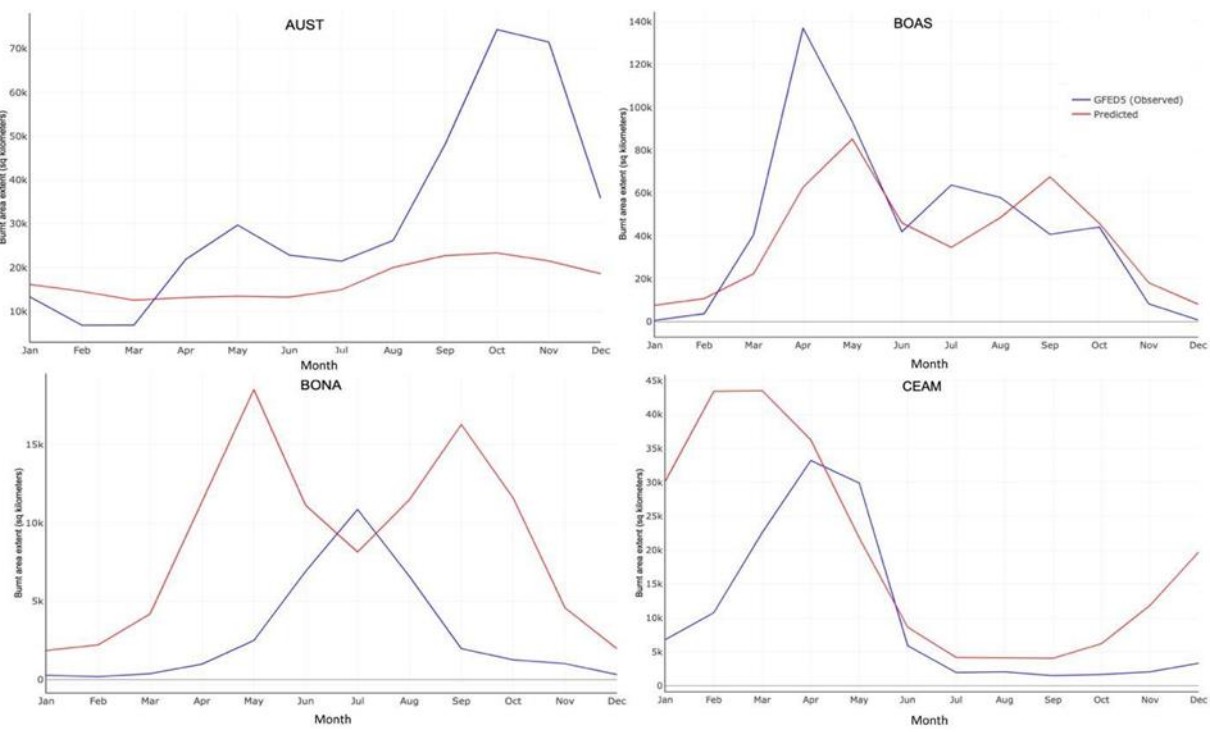



572

573



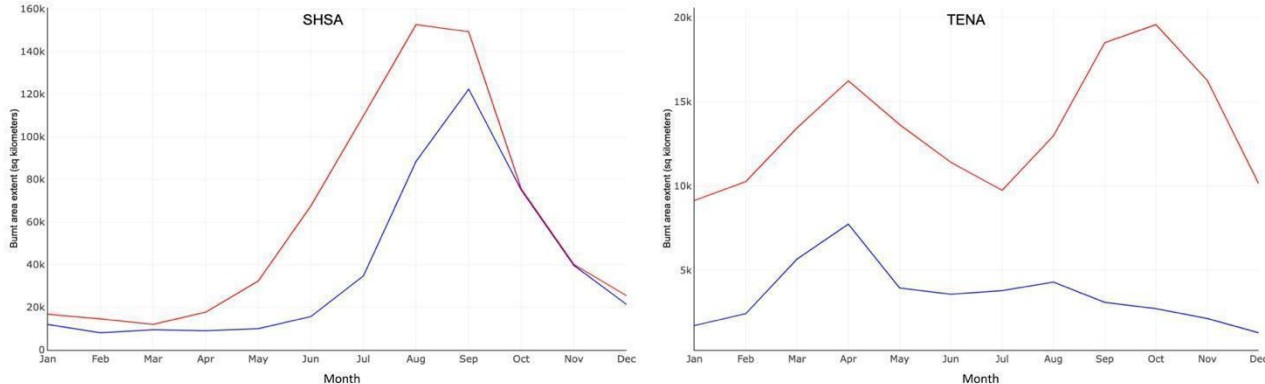

**Figure A2: Shows the observed (in red) and predicted (in blue) seasonal variability in burnt area fractions across different GFED regions.**





**Figure A3: Scatter plots illustrating single-factor relationships between burnt area fraction and various environmental and socio-economic variables: GPP Index, Fire Weather Index, Percentage Non-Tree Cover, Human Development Index, Percentage Tree Cover, Topographic Position Index, Percentage Dry Days, Road Density, Precipitation Seasonality and Annual Precipitation Index. The plots highlight distinct patterns, such as the negative correlation between percentage tree cover and burnt area fraction, and the positive correlation between number of dry days and burnt area fraction.**

| Region | Sen's slope | P-value |
|--------|-------------|---------|
| BONA | 558.354 | 0.1082 |
| TENA | 895.8292 | 0.4338 |



| | | |
|------|------------|--------|
| CEAM | -1963.035 | 0.1494 |
| NHSA | -1601.363 | 0.387 |
| SHSA | -9119.019 | 0.0529 |
| EURO | 189.2956 | 0.387 |
| MIDE | 202.3893 | 0.9016 |
| **NHAF** | **-22329.83** | **0.0026** |
| **SHAF** | **-28205.43** | **0.0001** |
| BOAS | -1560.25 | 0.1494 |
| **CEAS** | **-8342.713** | **0.0011** |
| **SEAS** | **-9671.238** | **0.0034** |
| EQAS | 69.04606 | 0.9671 |
| AUST | 1141.46 | 0.3434 |

**Table A1: Mann-Kendall test results for trend analysis across GFED regions from 2002 to 2018. Regions with significant trends are in bold (NHAF, SHAF, CEAS, SEAS); the remaining ten regions show insignificant trends.**

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

and control attributions based on the ensemble machine learning and satellite observations. Sci. Remote Sens. 7, 100088.