# Peer review of "Development of a statistical model for global burned area simulation within a DGVM-compatible framework"

_EGUsphere, 2024_

## Referee Comment (RC2)

**General comment**

The manuscript entitled "A statistical global burned area model for seamless integration into Dynamic Global Vegetation Models" by Blessing Kavhu and colleagues develops a Generalized Linear Model (GLM) with 19 predictors. The authors designed and tested 26 models using burned area data from the Global Fire Emissions Database version 5 (GFED5) and combinations of the selected predictors. Model 25 was chosen as the best-performing model, with an explained deviance of 0.568 and a Normalized Mean Error (NME) of 0.718. The authors identified key predictors such as Fire Weather Index (FWI), and Percentage Non-Tree Cover (PNTC), which strongly influence fire occurrence, and Human Development Index (HDI), Gross Primary Productivity Index (GPPI), and Population Density (PPN), which are negatively associated with fire occurrence. While the model demonstrates limited accuracy in predicting global annual burned area variability (Figure 5), it performs well in capturing global seasonal variability (Figure 8). The authors also discussed the comparison between predicted and observed data in terms of spatio-temporal variability at the GFED regional level.

In general, I have concern regarding the alignment of the manuscript's title with its methods and objectives. The current title suggests that the authors developed statistical models (GLMs) seamlessly integrated into Dynamic Global Vegetation Models (DGVMs). However, upon reviewing the manuscript, it becomes clear that the GLM was built independently of any DGVM, and the integration is only theoretically explained. According to my understanding, true integration with DGVMs requires a lot and long modification processes, testing within specific DGVM frameworks, reparameterization, new input-output verification, module integration, and validation process. The integration process involves technical adjustments such as modifying input data formats, calibrating modules, and ensuring compatibility with existing model components (e.g., physical, physiological, vegetation, or disturbance, and biogeochemical modules). Without actual implementation and demonstrated results, the claim of "seamless integration" remains unsupported. I suggest revising the title to reflect the study's scope and contributions more accurately. For example, the title could emphasize the development of a GLM, its evaluation of wildfire drivers, and its ability to predict spatio-temporal variability in burned area data.

Additionally, the study workflow needs to be presented more systematically. I recommend referencing workflows from published manuscripts in this field and ensuring that critical methodological details, such as data sources, temporal coverage of input data, and prediction periods, are clearly outlined. The abstract section is also not structurally strong enough, it should be rearranged. A clear and detailed workflow will greatly aid readers in understanding the study's methodology. Furthermore, the term "prediction" should be adjusted to "historical prediction" to reflect the study's temporal scope (2002–2018).

Overall, I recommend major revision before this manuscript can be considered for publication in Biogeosciences. Addressing the points mentioned above, along with detailed reviewers comments, will significantly enhance the manuscript's clarity and alignment with its objectives. Please find detailed comments below.

**Detailed Comments**

**L1** "Statistical global burned area model"

Could you please specify what kind of statistical model that you used in this study?

The phrase is somewhat broad. Consider specifying the nature or methodology of the statistical model (e.g., linear, regression-based, machine learning, empirical, so on). This would make the title more precise and appealing to a specialized audience.

"Seamless integration into Dynamic Global Vegetation Models"

Could you please explain what does "seamless integration" means?

The term "seamless integration" is somewhat subjective and may overpromise ease of implementation. Consider replacing it with a more objective phrase or point out the advantages or novelty of this statistical model with existing burned area models that are widely used globally.

**L11** The abstract structure is still weak and requires rewriting. Generally, an abstract should include, in sequence: the main problems identified by the authors, the solutions proposed to address these problems, the methodology applied in the study, and the objectives of the research. The results should be summarized, supported by numerical findings and validation metrics, and addition of standard deviations in the result. Additionally, the conclusions and key findings should be highlighted, and the abstract should conclude with a statement on the contribution of the study to the scientific field, which you have partially addressed in the first and final sentence of your current abstract.

**L13** Is wildfire modeling challenging solely due to human behavior? What about natural dynamics, such as climate and other environmental variables, that also influence wildfires?

**L14** Is the main goal of this study to demonstrate the relationship between biophysical and socioeconomic factors and wildfire dynamics, including monthly burned area? The manuscript title should clearly reflect the primary objective of your research.

**L15** The sentence, 'We developed Generalised (Generalized?) Linear Models (GLMs) to capture the relationships between potential predictor variables that are simulated by DGVMs…' can be combined with the previous sentence, as both explain the objectives of this study. Combining them would improve the flow and cohesiveness of the text while reducing redundancy.

**L18** What does the "final model" mean?

**L23** To enhance the scientific rigor and clarity of the manuscript, I suggest including the r-values (correlation coefficients) for both positive and negative correlations between BA and the predictor variables. Reporting these values will provide a clearer understanding of the strength of these relationships. Additionally, where possible, p-values should be included alongside r-values to

indicate statistical significance (e.g., p < 0.05). To maintain the brevity of the abstract, p-values can be detailed in another section that elaborates on these findings.

**L24** Before discussing the model predictions, could you please provide a comparison of your statistical model's performance with the benchmark dataset (GFED5) for burned area? Typically, in modeling, it is essential to first validate the model's performance using historical observation data before applying it to future predictions. This will help contextualize the model's accuracy and allow for a better understanding of its strengths and limitations in comparison to the established dataset.

**L27** The use of 'R > 0.50' is vague. To strengthen this statement, please provide a range of correlation values or specific values for different regions. This will offer a clearer understanding of the model's performance across various areas and enhance the scientific rigor of the analysis.

**L36** Please specify the year the data is from when mentioning the record fires in Australia and Canada, each burning more than 15 million hectares (Barnes et al., 2023; Jain et al., 2024). Additionally, it would be useful to include a comparison with the wildfires in Australia and Canada from the previous year. This will help strengthen the statement that 'climate change has led to more severe fire weather' by providing a clearer context of how fire severity has evolved over time.

**L60** Please explain why the predictive skill of process-based models is often not yet satisfactory. If the explanation is similar to the next point, you may consider combining both sentences, as they both address the limitations of process-based models. This will improve the coherence and conciseness of the text.

**L63** Please explain why statistical approaches are often used to evaluate human impacts on wildfires. Highlight the advantages of statistical methods.

**L65** Please explain why the application of statistical models to ecosystems other than those used in their derivation is often not reliable. Highlight the limitations of statistical methods.

**L88** "However, the integration of these techniques into DGVM is yet to be realized". Please check this article: https://doi.org/10.1029/2023MS003710. Son, et al., 2024 integrated of a Deep-Learning-Based Fire Model Into a Global Land Surface Model. I suggest reviewing the article directly to confirm the specifics of the DGVM's integration approach.

**L90** If possible, please mention the DGVM name you are criticizing to provide clearer information. By specifying which DGVM you are referring to, readers will better understand the context of your critique and the gap in current modeling approaches.

**L93** Please check this article: https://doi.org/10.5194/bg-21-4195-2024. Section 2.4 Model application. Nurrohman, et al., 2024 has integrated SEIB-DGVM with the SPITFIRE fire model and modified DGVM to produce monthly outputs using statistical downscaling methods. The method is able to capture monthly wildfire dynamics with results very similar to GFED4s benchmark data (Figure 5 in that article).

**L94** Using up-to-date remote sensing datasets in the DGVM as input or validation dataset?

**L95** Please clarify the integration between DGVMs and statistical models, DGVMs and remote sensing datasets, or both. The previous sentence discussed using remote sensing datasets for fire modeling, so it would be helpful to understand how these approaches are integrated and how each component contributes to enhancing the model's fire prediction accuracy.

**L111** Figure 1: Please create a simple workflow legend that explains the definition of each shape. Additionally, clarify the meaning of the color differences (red and black). There are international standards for workflow/flowchart design, including how to select shapes, so please refer to them. You can also refer to published manuscript in Biogeoscience for example. To enhance readability, when mentioning external datasets or input data, include the dataset names along with the corresponding year ranges for the data you are using.

**L111** I believe the small black boxes below the red tube shape are meant to describe each process, correct? For example, you used Pearson correlation in the correlation matrix, a quasi-binomial GLM, and so on. If this is the case, I recommend avoiding the use of shapes for these descriptions, as shapes can represent processes, data, or other meanings depending on the one selected. Instead, it would be clearer to provide descriptions below each process or explain them in a paragraph beneath Figure 1. This will enhance clarity and avoid redundancy in the diagram.

**L115** Could you please clarify whether you used the entire burned area (BA) time range from the GFED5 dataset for your study, or just specific years such as 2002 and 2018?

**L115** Please ensure that the abbreviation for the Global Fire Emissions Database is written uniformly as either "GFEDv5" or "GFED5" throughout the manuscript. Consistency in terminology helps maintain clarity and professionalism in scientific writing.

**L124** At first, please directly explain the predictor variable you are using, then followed by the sentence "whilst" along with the reason why you use those predictors only. This structure will help clarify the rationale behind your choices and ensure a logical flow in your explanation.

**L135** For Table 1, in the source column, I suggest adopting a consistent citation style for all the predictor data used. This means including either the author's name and publication year or the specific time when you accessed the dataset. Be sure to check the "how to cite" instructions for each dataset to provide the correct format. This will ensure clarity and credibility in referencing the data sources. Please check Table 1. Summary of predictor and fire response variables (Haas et al., 2022) https://doi.org/10.1088/1748-9326/ac6a69

**L135** Ensure that the table header is repeated on all pages where the table appears, and apply this adjustment to other tables as well.

**L155** Please check section 2.3 and 2.4. both have a same name "vegetation-related predictors"

**L169** Could you please provide a more detailed explanation of the HDI data? This should include the data range, units (if applicable), and guidance on interpreting the HDI data used in your study. Since your results indicate that HDI predictors have a significant impact, this clarification is necessary to ensure that the discussion aligns with the proper interpretation of the HDI data.

**L178** I suggest changing the structure of sections 2.3 to 2.6, to become sub-sections under section 2.2. Because it explains about predictor variables specifically, to simplify the reading flow. Example: 2.2. Predictor variables, 2.2.1. Vegetation-related predictors, ... 2.2.4. Weather-related predictors. 2.3. Data processing.

**L190** You describe data processing in subsection 2.7, but this step is not included in Figure 1, the study workflow. I recommend adding data processing to Figure 1 for consistency. Additionally, to enhance clarity for readers, ensure that Section 2 provides a detailed and sequential explanation of the data and study processes, aligning with the workflow illustrated in Figure 1.

**L200** "GPPI was originally defined as the Monthly Ecosystem Productivity Index (MEPI) in the study by Forrest et al. (2024)". After reviewing their work, I see that MEPI evaluates ecosystem productivity, including vegetation health and phenological states. Could you please clarify why you renamed MEPI to GPPI while using the same equation, and why it is referred to as the 'novel GPPI' (as mentioned in lines 20 and 533)?

**L223** Figure 2. This correlation matrix is a part of your results, isn't it? According to Figure 1. Correlation matrix is classified as the first step of your study. If so, please move Figure 2 to the result section.

**L229** Is it Figure A3 or Figure 3? Additionally, how many plots did you create ten or eight?

**L229** In my opinion, it would be better not to mention the result image (Figure 3) in Section 2, as this section should focus solely on the data and methodology.

**L236** What data did you use for model calibration and testing? Was it GFED5? Please include the calibration and testing process in Figure 1 for clarity.

**L262** You can delete this sentence, as this information is described in the table caption. This paragraph could start with the sentence "The initial models ... so on"

**L274** Could you please explain more details about the reason why you chose Model 25 instead of Model 24 or another model with better deviance and NME? If it is for the reason of future projection of RD data, we can refer to the deviance and NME values in the second, third and so on, because based on the explanation, these two variables are a reference for whether the model is good or not. If it is for simpler reasons, you better explain how the model is said to be simple, whether because there is no multiplication between predictor variables or other reasons.

**L267** Could you please explain why you introduce polynomial terms for Percentage Tree Cover (PTC)?

**L277** Could you please explain how the predictor formula in each model is determined, the reasons for summing or multiplying the predictors?

**L287** Please change the exponent symbol (e) to $\times 10^x$

**L302** Could you please explain what Larson and McCleary 1972 means? I checked the References and there is no citation information.

**L303** (GPP) or (GPPI)?

**L308** Please search additional references related to similar GLM modeling that have similar or lower explained deviance values than yours, and providing explanations to strengthen your results that the values are accepted.

**L322** Please write the year of the observed burnt area (GFED5) and predicted burnt area datasets in Figure 4.

**L330** Could you please explain the reason for adjusting the HDI predictor to be included, excluded, or constant? I don't see any explanation of this HDI setting in the methods section or in sub-section 2.5. Anthropogenic Influence Predictor.

**L331** Please remove the title of Figure 5 "Internanual variability .." at the top of the graph, as it is already written in the figure caption.

**L331** Please reduce the burnt area unit (y-axis) to be 2 digits, so that the digit is not too long by applying $\times 10^x$

**L338** Please write the full name of the abbreviation SHAF, SHSA, NHAF, CEAS and so on (because it has not been explained before)

**L345** Please clarify the figure caption to be understandable by the reader. Is it a interannual variability comparison of burnt areas between model projections and GFEDv5?

**L345** Please clarify the figure caption to be understandable by the reader. Is it a interannual variability comparison of burnt areas between model projections and GFEDv5?

**L355** To make it briefer, you can combine figure 6 and figure 7.

**L367** Please remove image title above the graph "Global Seasonal Cycle" and adjust the y-axis not using "k", you can use $\times 10^x$

**L372** Please delete the paragraph between section 4. Discussion and sub section 4.1. It's better to discuss the research result directly.

**L378** I have not seen any explanation about what DGVM you used in this study and how you integrate your GLM with those DGVM, either in the methodology section or elsewhere. How can you state that this GLM is compatible in DGVM? To my current understanding, each DGVM has its own characteristics, starting from their programming language and the flow of how it reads specific input data (so it needs data handling / pre-processing to be integrated into each DGVM).

**L423** Please provide cross-reference form figure that support this statement, so that readers can easily refer to specific figure. In addition, please provide more explanation why your model exhibits stronger performance in those regions including in the northern hemisphere.

**L432** After this sentence, it is better to explain how this model contributes to novel insights into the factors that influence global fire trends, and after that you can compare with other studies.

**L442** "This highlights the significant influence of HDI in projecting the purported negative global ire trend". If so, you can further discuss how the spatio-temporal variability of projected burnt area of each region (in this sub-section or other sub-section), how HDI affects the burnt area in the region. If the HDI data is the same as this source: https://hdr.undp.org/data-center/human-development-index#/indicies/HDI, then you can associate regions with low, medium, high and very high HDI.

**L442** "The HDI is related to factors like advancements in fire control methods, surveillance, technology, and outreach strategies increasing awareness, particularly in response to the growing human technological developments". Can you add one or some references that supports this statement?

**L450** In this sub-section, you can discuss how the interannual variability of your model in each GFED region (as shown in Figure A1), how your model performs compared to the observational data from GFED5.

**L463** Please also discuss another DGVM that used SPITFIRE fire module, as SPITFIRE is an updated fire module from GlobFIRM. SPITFIRE has implemented full burned area calculations and considers natural ignition factors from lightning and ignition and fire suppression based on population density (Thonicke et al., 2010).

SPITFIRE effectively includes human fire suppression on other lands because human ignitions first increase and then decrease with increasing population density (Hantson et al., 2016).

Models that explicitly simulate the impact of human suppression on fire growth or burnt area (CLM, CLASS–CTEM, JSBACH–SPITFIRE, LPJ–GUESS–SIMFIRE–BLAZE) are better at representing the spatial pattern in burnt area compared to models which do not include this effect (0.85 and 0.93 respectively). (Hantson et al., 2020)

Please try to check the following papers that discuss FireMIP and DGVM used to simulate fire and burned biomass emissions resulting from forest fires.

Historical (1700–2012) global multi-model estimates of the fire emissions from the Fire Modeling Intercomparison Project (FireMIP) (Li et al., 2019)

The Fire Modeling Intercomparison Project (FireMIP), phase 1: experimental and analytical protocols with detailed model descriptions (Rabin et al., 2017)

The status and challenge of global fire modelling (Hantson et al., 2016)

**L468** "The findings of this study exhibit robustness in capturing seasonal cycles ($R^2$= 0.536)," Could you please include a cross-reference an image that states this? To make it easier for readers to refer to the results you are discussing. Please apply throughout the rest of the section, when you mention the results of the study, include a cross-reference with a supporting figure or table.

**L471** Could you please provide evidence to support this assertion that it is due to climatic conditions in those regions? You can compare seasonal fire patterns and climatic conditions in those regions and discuss the result in this sub-section.

**L478** In my opinion, I think this paragraph is better presented at the beginning of sub section 4.5.

**L478** Do you do future predictions of annual or seasonal burned area data globally?

**L490** I suggest renaming this sub-section to "model limitation and excluded drivers". Include the explanation and discussion of "model shortcomings" in sub-section 4.7 to this section.

**L500** "FAPAR is highly correlated with GPP." Please confirm, according to Figure 2. FAPAR correlation with GPP is 0.59, or do you mean FAPAR in general which includes FAPAR, FAPAR12 and FAPAR6?

**L512** I suggest this sub-section be changed to 4.7. Recommendations (after you separate the discussion of shortcomings, as I suggested in L490). Or you could also combine section 5. Conclusions and Recommendations, to harmonize after you discuss the Conclusions, you can suggest recommendations regarding further studies.

**L513** "The findings of this study offer valuable insights into the underlying drivers and patterns shaping global fire dynamics". The sentence does not explain the model shortcomings or recommendations. It seems better to put it in the Conclusion section. In addition, in this sub-section, please explain about recommendations only, you can discuss recommendations on how to solve the limitations or shortcomings of the current model, or further studies from this research.

**L525** This first paragraph doesn't fit in the Conclusion section, it's more like an introduction, I suggest deleting this first paragraph.

**L531** Make sure the Conclusion section explains the research objectives .The first two sentences have answered the first objective, but add how much the major predictors correlate with burnt area (BA).

**L534** Explain the performance of the model when predicting BA compared to the GFED5 observational data -> state the evaluation value index that you used for validating the model before predicting, how do you state the model is suitable to be used to predict BA.

**L536** Before the phrase "We hope", explain the third objective, explain how the model performance, including spatio-temporal, interannual and seasonal cycle of BA compares with GFED5 observational data.

---

## Author Response (AR1)

**A summary of our main changes in response to the reviewer's comments is:**

- **Title updated for clarity**: The phrase "seamless integration" was replaced with "tailored for integration" to avoid overstating the status of integration into DGVMs and better reflect the model's compatibility focus (Reviewer 2).
- **Abstract restructured and rewritten**: The abstract was thoroughly revised to align with standard scientific structure, incorporating objectives, methodology, key results (with validation metrics), and contributions to the field (Reviewer 2).
- New section on future integration into DGVMs: A dedicated section titled "Next steps for DGVM integration and future predictions" was added to clearly outline the steps required for implementing the model within DGVM frameworks, including discussion of technical integration challenges (Reviewers 1 and 2).
- Workflow and methodology clarified: The model workflow was revised with a cleaner, more informative figure, removing confusing shapes and colors. Supporting text was added to explain data sources, training and testing splits, and each modeling step including data processing, correlation analysis, and VIF checks (Reviewers 1 and 2).
- Expanded comparison of modeling approaches: The manuscript now contains a more balanced discussion of process-based versus statistical fire modeling, highlighting the rationale behind using GLMs and how mechanistic reasoning was preserved in variable selection (Reviewer 1).
- Model calibration and validation explained in more detail: The periods used for training (2002–2010) and testing (2011–2018) were clarified, and justification was provided for selecting the final model based on parsimony, collinearity, and applicability for DGVM use (Reviewers 1 and 2).
- **Terminology standardized and clarified**: Terms such as MEPI (previously referred to as GPPI) were made consistent throughout, and abbreviations (e.g., GFED regions like SHAF, SHSA) were introduced early and explained clearly (Reviewer 2).
- **Predictor selection and transformation justified**: Detailed rationale was added for including polynomial terms (e.g., for Percentage Tree Cover), applying interaction terms, and using stepwise model building based on prior literature and empirical testing (Reviewer 2).
- Model performance clearly quantified: The manuscript now reports performance across spatial, seasonal, and interannual variability with Normalized Mean Error (NME) and R2 values, and provides regional breakdowns to show where model performance is strongest (Reviewer 2).
- **Figures revised for clarity**: Figures were updated to ensure axis formatting (e.g., scientific notation), remove redundant titles, and clearly state the time periods represented. Figure captions were improved for interpretability (Reviewer 1 and 2).
- **Discussion of model limitations and excluded variables expanded**: A revised section addresses model limitations, including the omission of lightning as a predictor, lack of explicit burn severity representation, and the trade-offs involved in excluding variables like FAPAR or litter explicitly (Reviewer 1).

- Clarification of HDI's role: Additional explanation was added to clarify the rationale for including, excluding, or holding HDI constant in different scenarios, and how this affected observed trends in burned area (Reviewers 1 and 2).
- References updated and expanded: The manuscript now cites recent relevant literature (e.g., Son et al., 2024; Nurrohman et al., 2024) to position the work within the current state of DGVM fire modeling and to clarify methodological distinctions.

**Reviewer 1**

**General comments:**

The authors developed the statistical simulation model for fire burnt area which was supposed to be incorporated into DGVM. The proposed GLMs as the functions of multiple environmental factors, would give the potential for a very light simulation tool on global scale wildfire.

However, one of my concerns is that the wildfire would be ignited considerably by lighting and expand their burning severity depending on the dry litter amount as fuel, both of which are not considered this time. The predictability for the long-term future is not secured as long as you do not use an accumulated ecosystem carbon stock, which DGVM has the advantage to simulate, but you use the GPP related index which is not necessarily related to fire severity. Statistical models cannot guarantee accuracy when a regime-shift for fire occurrence has happened mechanistically.

Response: The reviewer raises several important points here. It is true that we do not explicitly include litter amount. Initially, we did try to do this with an above ground biomass dataset as a proxy (AGB, Santoro et al. 2021, data not shown). However, we found that this data was not sufficiently sensitive to variations at the low end of AGB that we considered to be important for capturing this dynamic in fuel limited ecosystems. Instead, we do include proxies of both litter quantity and litter type through percentage tree cover, percentage non-tree cover and number of dry days (here dry days are functioning as an inversely related proxy for biomass accumulation in arid areas). We believe that this captures the core dynamic that more fuel produces more burned areas - at least at an appropriate level of complexity for a global model. Regarding burning severity, the model does not explicitly consider burn severity, although we appreciate that higher severity will, all else equal, give higher burned area. However, by using a statistical model with appropriate predictors, our intent is to implicitly include this effect on burnt area through the predictors. We note that when integrated into a DGVM, burn severity (or more likely fire line intensity) can be calculated using the DGVM's carbon pools, thus leveraging the strength of DGVM to give severity/intensity-dependent combustion and mortality.

The use of GPP index (now MEPI in the revised draft) here is not intended directly as fuel accumulation/fire severity proxy but rather is a seasonality predictor to which captures periods of low GPP (relative to that location's maximum).

Regarding lightning, we appreciate that lightning ignitions are regionally important. However, globally, and in most regions, most ignitions are human caused (Balch et al. 2017, Jannsen et al. 2023). We also consider that, most likely, ignitions are not globally a limiting factor (initially based on results in Haas et al. 2022 but also see the more recent Haas et al 2024). Furthermore, there are practical concerns with using lightning. Lightning projections from atmospheric models are not commonly available and highly uncertain, and there are even issues with present day observed lightning data - with the LIS/OTD product only providing climatology for 1995-2014 and the WGLC product having detection inefficiency issues. We therefore deliberately excluded lightning from this global analysis but fully agree that lightning should be considered in regional and future studies.

Finally, clearly there are pros and cons with process-based versus empirical models, and often both types are mixed. Here, we chose a statistical model for burned area, but only using input variables that also make sense from a mechanistic point of view, which also can be simulated by DGVMs or easily be derived from climate model output, to ensure the applicability of the fire model here in DGVMs for future predictions. When choosing such an approach, we think one also must base the final decision on driver variables on the statistical results, even if these are not fully in line with mechanistic hypotheses. However, we acknowledge the relevant point of the reviewer and have changed the text accordingly towards the end of the second paragraph in the introduction. In the second-last paragraph of the introduction (See line 96 – 112), we added the following text: "GLMs are easier to implement into DGVMs, and their partial residual plots show the relationships of each predictor with the response variable when all other drivers are kept constant, which facilitates a discussion of potential underlying mechanisms. The risk of overfitting can be minimized by only choosing potential driver variables that are mechanistically expected to play an important role and by choosing only a limited number of uncorrelated driver variables. Accordingly, Haas et al. (2022) developed a GLM for global burned areas with good model skill but without accounting for seasonal dynamics and without a focus on driver variables that can be predicted with DGVMs. (See line 96 – 104)

Considering the fewer predictors selected for the general representation of wildfire, the linear regression is shown not to be the best model for wildfire though still easy to use compared to machine learning or process-based models. Correlation coefficients for each term should be rearranged when this model was introduced to DGVM to use their simulated GPP and other ecosystem information which must be more or less different from MODIS or satellite products.

**Response:** Yes, we agree that when integrating in a DGVM, the model should be refitted based on model output as opposed to remotely sensed data. This is a necessary next step for DGVM integration, and it remains to be seen how much the relationships change by using DGVM outputs. We have added a new paragraph entitled "..." to the Discussion explaining which next steps are necessary for DGVM integration (See line 555 – 581). Furthermore, we replaced the formulation "for seamless integration" in the title with "tailored for integration" to avoid the impression that the integration into a DGVM is a trivial task. This point was raised by the second reviewer.

CEAM, BONA, MIDE, and TENA showed a 2-fold difference in interannual variations, which suggests that this model could be parameterized or separately made for vegetation types. A single statistical model cannot estimate the average BA for specific areas.

**Response:** Yes, good point. We tried this; however, we found very small improvements in our predictions when fitting the model regionally.

**Minor Comments:**

Page 3, Lines 90-91, you have to identify the typical model name that you criticize here. What you propose here is still a simple GLM based module, so I feel this sentence as contradiction.

**Response -Page 3, Lines 90-91:** We specified the model being criticized (LPJ-LMfire(v1) and clarified that this model like most other DGVMs was not formally parameterized to predict seasonal fire cycles. We highlight how our approach differs in methodology and performance particularly with regards to statistically predictions of seasonal cycles (See line 105-107).

Page 3, Line 94-95, Use of remote sensing data will reduce the advantage of DGVM which enables the long-term vegetation shift simulation in future. also wanna now the direction from start to end

**Response -Page 3, Lines 94-95:** We recognize the trade-off between using remote sensing data and DGVM-driven predictions. We clarified that our model is designed outside of the DGVM framework, with the goal of being later integrated using DGVM-simulated predictors (See line 110 - 111).

Page 17, Line 330 does this mean that HDI is going down in these years? you would better show the number in the decadal trend of averaged globally.

**Response -Page 17, Line 330:** HDI has increased over time, but its negative correlation with burned area leads to an apparent decreasing trend in burned area. We have clarified this by now stating "Excluding and holding HDI constant in the model made the projected trend remain steady, suggesting the role of anthropogenic developments (increasing HDI over time) driving a downward trend in wildfire distribution." (increasing HDI over time) was added (See line 350 – 351).

"Page 22, Lines 425-427: sounds repeated. you merge these two sentences into one.

**Response -Page 22, Lines 425-427:** Great point, we removed repeating phrases the sentences now read as follows: "Our findings revealed that HDI, MEPI and PPN are negatively associated with trends in global fire extent. For HDI, our findings imply that technological advancements, improved surveillance systems, and effective mitigation efforts play a significant role....." (See line 417 – 418).

Page 23, Lines 463: you also have to mention about the SPITFIRE-based model performance. GlobFIRM is an old version, and we know this is not accurate.

**Response -Page 23, Line 463:** Great point, however our statistical model is DGVM-agnostic, hence we removed discussions of results from DGVMs (including GlobFIRM) to avoid confusing our readers. As a result, we did not discuss SPITFIRE too

Figure 1: you should add more explanation on the shape of frames, rectangles, rounds, diamonds. what is the data and what is the process

**Response -Figure 1:** Great point, we revised the workflow diagram by removing colours and shapes that could confuse readers. Additionally, we added more details about the training and testing data that went into the model, including the other processes that were missing such as data processing, correlation analysis and variance inflation factor. (See line 122 -127)

**Table 2: explain the condition for color**

Response -Table 2: That's true, we explained the legend in the legend caption and the caption now reads "Results of modeling attempts using different combinations of predictor variables using a progressive inclusion of covariates approach. For Normalized Mean Error (NME), higher values are represented by warmer colours (with red indicating the highest error), while lower values appear in cooler colours (green indicating the lowest error). In contrast, for Deviance Explained, higher values are shown in cooler colours (green indicating better performance), and lower values in warmer colours (red indicating poorer performance). An optimal model is indicated by a combination of cooler colours for both metrics, whereas a combination of warmer colours suggests poor model performance (See line 624 – 632). Also, it's important to note that we moved the table to the appendices section as suggested by the editor.

**Figure 4: Specify the years for the average**

**Response -Figure 4:** Agreed, we included the years for the average in the revised map. Specifically, we used the average observed of the period 2011 to 2018 and the average predicted for the same period.

Figure 5: The y-axis should be in 106 to reduce the number of digits.

**Response -Figure 5:** That's true! We revised the y-axis and used the 106 notation to improve readability. (See line 351 – 355).

Citation: https://doi.org/10.5194/egusphere-2024-3595-RC1

**Reviewer 2**

**General comment**

The manuscript entitled "A statistical global burned area model for seamless integration into Dynamic Global Vegetation Models" by Blessing Kavhu and colleagues develops a Generalized Linear Model (GLM) with 19 predictors. The authors designed and tested 26 models using burned

area data from the Global Fire Emissions Database version 5 (GFED5) and combinations of the selected predictors. Model 25 was chosen as the best-performing model, with an explained deviance of 0.568 and a Normalized Mean Error (NME) of 0.718. The authors identified key predictors such as Fire Weather Index (FWI), and Percentage Non-Tree Cover (PNTC), which strongly influence fire occurrence, and Human Development Index (HDI), Gross Primary Productivity Index (GPPI), and Population Density (PPN), which are negatively associated with fire occurrence. While the model demonstrates limited accuracy in predicting global annual burned area variability (Figure 5), it performs well in capturing global seasonal variability (Figure 8). The authors also discussed the comparison between predicted and observed data in terms of spatio temporal variability at the GFED regional level.

In general, I have concern regarding the alignment of the manuscript's title with its methods and objectives. The current title suggests that the authors developed statistical models (GLMs) seamlessly integrated into Dynamic Global Vegetation Models (DGVMs). However, upon reviewing the manuscript, it becomes clear that the GLM was built independently of any DGVM, and the integration is only theoretically explained. According to my understanding, true integration with DGVMs requires a lot and long modification processes, testing within specific DGVM frameworks, reparameterization, new input-output verification, module integration, and validation process. The integration process involves technical adjustments such as modifying input data formats, calibrating modules, and ensuring compatibility with existing model components (e.g., physical, physiological, vegetation, or disturbance, and biogeochemical modules). Without actual implementation and demonstrated results, the claim of "seamless integration" remains unsupported. I suggest revising the title to reflect the study's scope and contributions more accurately. For example, the title could emphasize the development of a GLM, its evaluation of wildfire drivers, and its ability to predict spatio-temporal variability in burned area data. Additionally, the study workflow needs to be presented more systematically. I recommend referencing workflows from published manuscripts in this field and ensuring that critical methodological details, such as data sources, temporal coverage of input data, and prediction periods, are clearly outlined. The abstract section is also not structurally strong enough, it should be rearranged. A clear and detailed workflow will greatly aid readers in understanding the study's methodology. Furthermore, the term "prediction" should be adjusted to "historical prediction" to reflect the study's temporal scope (2002–2018). Overall, I recommend major revision before this manuscript can be considered for publication in Biogeosciences. Addressing the points mentioned above, along with detailed reviewers comments, will significantly enhance the manuscript's clarity and alignment with its objectives. Please find detailed comments below.

**Response -Title accuracy :**We agree that the phrase "seamless integration" may overstate the current status of DGVM incorporation. Therefore, we have replaced "seamless" from the title. We kept the rest because the choice of predictor variables was strongly influenced by the longer-term aim of integration within DGVMs for future predictions. In other studies, the choice of predictor variables was less constrained (with more focus on the best possible match with the data), but this hampers the integration of the results into DGVMs. We want to make this special focus here already clear in the title. (See line 1-2)

We have added text to the discussion to make clearer which additional steps are necessary for integration into DGVMs. (See line 555 – 581)

**Response -Workflow and abstract:** Great point on the workflow, this was also raised by Reviewer 1. Hence, we revised the workflow by removing colours and shapes that could confuse readers. Additionally, we added more details about the training and testing data that went into the model, including the other processes that were missing such as data processing, correlation analysis and variance inflation factor. The workflow description will be revised for clarity, ensuring that data sources, temporal coverage, and prediction periods are explicitly outlined. (See line 122-124).

We also revised the abstract following the journal structure to include phrases on problem identification, methodology, key results with numerical values, and contributions to the field. (See line 11 -32).

**Detailed Comments**

L1"Statistical global burned area model" Could you please specify what kind of statistical model that you used in this study? The phrase is somewhat broad. Consider specifying the nature or methodology of the statistical model (e.g., linear, regression-based, machine learning, empirical, so on). This would make the title more precise and appealing to a specialized audience. "Seamless integration into Dynamic Global Vegetation Models"

Could you please explain what does "seamless integration" means?

The term "seamless integration" is somewhat subjective and may overpromise ease of implementation. Consider replacing it with a more objective phrase or point out the advantages or novelty of this statistical model with existing burned area models that are widely used globally.

**Response -L1 (Title):**Great points, we revised the title and specified the linear statistical model and replaced the term "seamless integration" with "tailored for integration" to avoid misinterpretation. Our intent with "seamless" was to emphasize that the model uses predictors which are either already available within a typical DGVM or can be easily computed, or for which global gridded data is available including future projections. We suggest the word "tailored" better describes this and so suggest the title:

"A linear statistical global burned area model tailored for integration into Dynamic Global Vegetation Models" (See line 1-2)

**L11** The abstract structure is still weak and requires rewriting. Generally, an abstract should include, in sequence: the main problems identified by the authors, the solutions proposed to address these problems, the methodology applied in the study, and the objectives of the research. The results should be summarized, supported by numerical findings and validation metrics, and addition of standard deviations in the result. Additionally, the conclusions and key findings should be highlighted, and the abstract should conclude with a statement on the contribution of the study to the scientific field, which you have partially addressed in the first and final sentence of your current abstract.

**Response -L11 (Abstract):** Good point! We've completely restructured the abstract to align with standard scientific formatting. We have now included the relevant accuracies, strengths of relationships, and numerical validation metrics for spatial distribution, interannual variability and seasonal variability. (See line 11-32).

**L13** Is wildfire modeling challenging solely due to human behavior? What about natural dynamics, such as climate and other environmental variables, that also influence wildfires?

Response -L13: We acknowledge that wildfire modeling is influenced by both human activity and natural drivers. Our intent was to highlight that the early development of global fire models paid much more attention to climate and their environmental drivers. This sentence has been revised as part of the reformulated abstract. It now reads as follows "Fire-enabled Dynamic Global Vegetation Models (DGVMs) play an essential role in predicting vegetation dynamics and biogeochemical cycles amid climate change, but modelling wildfires has been challenging in process-based biophysics-oriented DGVMs, in particular regarding the role of socioeconomic drivers." (See line 11-13).

**L14** Is the main goal of this study to demonstrate the relationship between biophysical and socioeconomic factors and wildfire dynamics, including monthly burned area? The manuscript title should clearly reflect the primary objective of your research.

**Response -L14:** Great point, The main goal of this study is to build a linear statistical model based on the relationship between biophysical and socioeconomic factors and monthly burned area data, tailored for integration with DGVMs. We revised the manuscript title, and it now reflects the primary objective of our research. (See line 1-2).

**L15** The sentence, 'We developed Generalised (Generalized?) Linear Models (GLMs) to capture the relationships between potential predictor variables that are simulated by DGVMs...' can be combined with the previous sentence, as both explain the objectives of this study. Combining them would improve the flow and cohesiveness of the text while reducing redundancy.

**Response -L15:** Agreed! We combined the sentences, and it now reads as follows, "Using monthly burnt area (BA) data from the latest global burned area product from GFED5 as our response variable, we developed Generalised Linear Models (GLMs) to capture the relationships between potential predictor variables (biophysical and socio-economic) that are simulated by DGVMs and/or available in future scenarios." (See line 15–17).

**L18** What does the "final model" mean?

**Response -L18:** Good point, we clarified that the "final model" refers to the best-performing GLM, selected based on minimum collinearity and maximum model performance in terms of reproducing observations. (See line 19-20)

**L23** To enhance the scientific rigor and clarity of the manuscript, I suggest including the r-values (correlation coefficients) for both positive and negative correlations between BA and the predictor variables. Reporting these values will provide a clearer understanding of the strength of these relationships. Additionally, where possible, p-values should be included alongside r-values to indicate statistical significance (e.g., p < 0.05). To maintain the brevity of the abstract, p-values can be detailed in another section that elaborates on these findings.

**Response -L23 (Correlation Values):** We prefer not to include univariate correlation coefficients (r-values) and statistical significance (p-values) for our predictors. Given that a multivariate approach is essential for predicting fire, we believe that reporting univariate correlations does not add useful information. We also deliberately choose not to report t-values from the GLM. This is because there are high levels of spatial autocorrelation in our predictor variables which, although it does not affect the central estimates coefficients, will artificially reduce the estimated

uncertainties and thus make the t-values unreliable. However, some measure of the strengths of relationships can be gleaned from the partial dependence plots (Figure 3). (See line 318-320).

L24 Before discussing the model predictions, could you please provide a comparison of your statistical model's performance with the benchmark dataset (GFED5) for burned area? Typically, in modeling, it is essential to first validate the model's performance using historical observation data before applying it to future predictions. This will help contextualize the model's accuracy and allow for a better understanding of its strengths and limitations in comparison to the established dataset.

**Response -L24 (Model Comparison to GFED5):** We provided details of a direct comparison between our model's predictions and GFED5 observational data for spatial, seasonal and interannual variability using the Normalised Mean Square Error (NME) and R2 respectively. We explicitly specified the performance metrics for some of the predictors in the abstract to avoid confusion. (See line 26-28).

**L27** The use of 'R > 0.50' is vague. To strengthen this statement, please provide a range of correlation values or specific values for different regions. This will offer a clearer understanding of the model's performance across various areas and enhance the scientific rigor of the analysis.

**Response -L27 (R>0.50 Clarification):** Good point, we included specific correlation values for differents regions which performed better, and it now reads as follows " ... its standout performance lay in capturing the seasonal variability, especially in regions often characterized by distinct wet and dry seasons, notably southern Africa( $R^2 = 0.72$  to 0.99), Australia( $R^2 = 0.75$  to 0.90)". (See line 28).

L36 Please specify the year the data is from when mentioning the record fires in Australia and Canada, each burning more than 15 million hectares (Barnes et al., 2023; Jain et al., 2024). Additionally, it would be useful to include a comparison with the wildfires in Australia and Canada from the previous year. This will help strengthen the statement that 'climate change has led to more severe fire weather' by providing a clearer context of how fire severity has evolved over time.

Response -L36 (Fire Events in Australia & Canada): Good point! We provided the year of the latest record-breaking fires (2023) in the manuscript and the included phrase reads "climate change has led to more severe fire weather in large parts of the world and record fires have recently occurred in Australia and Canada, burning more than 15 million (Kirchmeier-Young et al 2024) and 7.8 million ha in 2023 (MacCarthy et al., 2024) respectively". (See line 38-39).

**L60** Please explain why the predictive skill of process-based models is often not yet satisfactory. If the explanation is similar to the next point, you may consider combining both sentences, as they both address the limitations of process-based models. This will improve the coherence and conciseness of the text.

**Response -L60:** The predictive skill of process-based models is often limited due to incomplete representation of fire drivers, uncertainty in parameterization, and difficulties in accurately simulating human-fire interactions. This explanation was integrated in the manuscript for coherence. (See line 62-65).

**L63** Please explain why statistical approaches are often used to evaluate human impacts on wildfires. Highlight the advantages of statistical methods.

**Response -L63:** We clarified why statistical approaches are commonly used to evaluate human impacts on wildfires in the manuscript. Our justification includes that statistical approaches can effectively quantify correlations between fire occurrence and socioeconomic drivers. These methods allow for empirical validation, provide flexibility in handling diverse datasets, and facilitate the inclusion of multiple spatial and temporal scales that can be integrated in DGVMs. This will be clarified in the manuscript.(See line 66-68).

**L65** Please explain why the application of statistical models to ecosystems other than those used in their derivation is often not reliable. Highlight the limitations of statistical methods.

**Response -L60-L65 (Process-Based vs. Statistical Models):** We clarified the limitations behind transferability of statistical methods to be associated with the assumptions behind how they analyse relationships. This has been clarified as follows in the manuscript: "This is mainly because statistical models assume that the relationship between predictors and responses is stationery and context dependent, which is not typical of fires that are stochastic in nature." (See line 69-72).

**L88** "However, the integration of these techniques into DGVM is yet to be realized". Please check this article: https://doi.org/10.1029/2023MS003710. Son, et al., 2024 integrated of a Deep-Learning-Based Fire Model Into a Global Land Surface Model. I suggest reviewing the article directly to confirm the specifics of the DGVM's integration approach.

Response -L88: Son et al. did not fully integrate their new fire model into the land surface model JSBACH for future prediction because some of the land surface input variables were remote sensing-based and climate reanalysis, e.g. MODIS LAI and ERA5 soil water. For a full integration and future application that accounts for dynamic vegetation changes one needs to refit the model with simulated land surface characteristics, or, at least, explore how much the relationships change if simulated land surface characteristics are used instead of satellite-derived ones. However, we have now included a new section entitled "Next steps for DGVM integration and future predictions" that also discusses the Son approach, and make mention of it in the Introduction. (See line 97-98)

**L90** If possible, please mention the DGVM name you are criticizing to provide clearer information. By specifying which DGVM you are referring to, readers will better understand the context of your critique and the gap in current modeling approaches.

**Response -L90:** We think that discussing DGVMs will confuse our readers, especially since we don't plan to design this model for specific DGVM. Hence, we removed these phrases and included a new section which details DGVM integration and future predictions.

**L93** Please check this article: https://doi.org/10.5194/bg-21-4195-2024. Section 2.4 Model application. Nurrohman, et al., 2024 has integrated SEIB-DGVM with the SPITFIRE fire model and modified DGVM to produce monthly outputs using statistical downscaling methods. The method is able to capture monthly wildfire dynamics with results very similar to GFED4s benchmark data (Figure 5 in that article).

**Response -L93:** We appreciate the suggestion. The study by Nurrohman et al. (2024) was cited and discussed. Their study is different from our study in that it generates monthly outputs from

downscaling, yet we build a model based on monthly data, and hence our study has strength. (See line 107).

**L94** Using up-to-date remote sensing datasets in the DGVM as input or validation dataset?

**Response -L94:** Good catch, we noticed that this sentence was a bit confusing, hence we rephrased it. We were initially referring to training our algorithm with some recent remotes sensing datasets that better capture fire dynamics.

**L95** Please clarify the integration between DGVMs and statistical models, DGVMs and remote sensing datasets, or both. The previous sentence discussed using remote sensing datasets for fire modeling, so it would be helpful to understand how these approaches are integrated and how each component contributes to enhancing the model's fire prediction accuracy.

**Response -L95:** We did not integrate our statistical model into a DGVM in this paper, however, the integration process of DGVMs with statistical models was explicitly explained in the discussion section (See line 557-583) to highlight their respective contributions to improving fire prediction accuracy.

L111 Figure 1: Please create a simple workflow legend that explains the definition of each shape. Additionally, clarify the meaning of the color differences (red and black). There are international standards for workflow/flowchart design, including how to select shapes, so please refer to them. You can also refer to published manuscript in Biogeoscience for example. To enhance readability, when mentioning external datasets or input data, include the dataset names along with the corresponding year ranges for the data you are using.

**Response -L111 (Workflow Diagram):** This is an important point which was raised by Reviewer 1 as well. We created a simpler and more detailed workflow this time. To remove the confusion and complexity to our readers, we removed the undescribed shapes and colours that were previously included in the workflow.(See line 124-128)

L111 I believe the small black boxes below the red tube shape are meant to describe each process, correct? For example, you used Pearson correlation in the correlation matrix, a quasi-binomial GLM, and so on. If this is the case, I recommend avoiding the use of shapes for these descriptions, as shapes can represent processes, data, or other meanings depending on the one selected. Instead, it would be clearer to provide descriptions below each process or explain them in a paragraph beneath Figure 1. This will enhance clarity and avoid redundancy in the diagram.

**Response -L111:** We created a simpler and more detailed workflow as alluded to in the previous response. Processes such as correlation analysis, VIF and data processing were incorporated in a much simpler way. We also removed the different shapes and colours that were previously included in the workflow. (See line 124-128)

L115 Could you please clarify whether you used the entire burned area (BA) time range from the GFED5 dataset for your study, or just specific years such as 2002 and 2018?

**Response -L115:** We clarified in the manuscript that we used data for the period 2002-2010 for model training and data for 2011 to 2018 for model testing. (See line 135 -136).

L115 Please ensure that the abbreviation for the Global Fire Emissions Database is written uniformly as either "GFEDv5" or "GFED5" throughout the manuscript. Consistency in terminology helps maintain clarity and professionalism in scientific writing.

Response -L115: Good point, we used GFED5 consistently across the entire manuscript.

L124 At first, please directly explain the predictor variable you are using, then followed by the sentence "whilst" along with the reason why you use those predictors only. This structure will help clarify the rationale behind your choices and ensure a logical flow in your explanation.

Response -L124: That's true. We revised how we explain our predictor variables by first introducing the predictors and then justifying their selection. Here is an example of how we framed it in the manuscript: "We used eight vegetation predictor variables to comprehensively evaluate their role on global fire distribution. These variables encompass Percentage Grass Cover (PGC), Percentage Non-Tree Cover (PNTC), Percentage Crop Cover (PCC), Percentage Graze Cover (PGZC), Percentage Rangeland Cover (PRC), and Percentage Tree Cover (PTC). Previous work emphasizes the important role of vegetation on burnt area dynamics. For example, Thonicke et al. (2010), discussed the crucial role of vegetation structure in shaping fire occurrence, spread and intensity." (See lines 158-159, 167-168, 171-172, 175-176, 186-187, 204-205).

L135 For Table 1, in the source column, I suggest adopting a consistent citation style for all the predictor data used. This means including either the author's name and publication year or the specific time when you accessed the dataset. Be sure to check the "how to cite" instructions for each dataset to provide the correct format. This will ensure clarity and credibility in referencing the data sources. Please check Table 1. Summary of predictor and fire response variables (Haas et al., 2022) https://doi.org/10.1088/1748-9326/ac6a69.

**L135** Ensure that the table header is repeated on all pages where the table appears, and apply this adjustment to other tables as well.

**Response -L135:** We standardized the citation format for predictor data sources in Table 1 according to dataset-specific citation guidelines as done in (Haas et al 2022). We also repeated the table header on all pages for consistency. (See lines 152-156)

L155 Please check section 2.3 and 2.4. both have a same name "vegetation-related predictors"

**Response -L155:** Good point, we addressed the duplicate section titles for "Vegetation-related predictors" in Sections 2.3 and 2.4. We revised section 2.4 to topographic predictors. (See lines 157-158, 174-175)

L169 Could you please provide a more detailed explanation of the HDI data? This should include the data range, units (if applicable), and guidance on interpreting the HDI data used in your study. Since your results indicate that HDI predictors have a significant impact, this clarification is necessary to ensure that the discussion aligns with the proper interpretation of the HDI data.

**Response -L169 (HDI Data):** We provided a detailed explanation of HDI data, its range, and interpretation in the manuscript. Our explanation in the manuscript reads as follows: "HDI is a composite index developed by the United Nations Development Program (UNDP) to assess longterm progress in three basic dimensions of human development, including health (life expectancy at birth), education (mean years of schooling and expected years of schooling), and standard of living (gross national income per capita) (UNDP, 2023). HDI values range from 0 to 1, with higher values indicating higher levels of human development." (See line 188-192).

**L178** I suggest changing the structure of sections 2.3 to 2.6, to become sub-sections under section 2.2. Because it explains about predictor variables specifically, to simplify the reading flow. Example: 2.2. Predictor variables, 2.2.1. Vegetation-related predictors, ... 2.2.4. Weather-related predictors. 2.3. Data processing.

**Response -L178:** We restructured Sections 2.3 to 2.6 into sub-sections under "2.2 Predictor variables" to enhance readability. They now fall between sections 2.2.1 and 2.2.4 (See lines 157-203)

**L190** You describe data processing in subsection 2.7, but this step is not included in Figure 1, the study workflow. I recommend adding data processing to Figure 1 for consistency. Additionally, to enhance clarity for readers, ensure that Section 2 provides a detailed and sequential explanation of the data and study processes, aligning with the workflow illustrated in Figure 1.

**Response -L190 (Data Processing in Workflow):** We incorporated data processing steps into Figure 1 for consistency. (See line 124-128).

**L200** "GPPI was originally defined as the Monthly Ecosystem Productivity Index (MEPI) in the study by Forrest et al. (2024)". After reviewing their work, I see that MEPI evaluates ecosystem productivity, including vegetation health and phenological states. Could you please clarify why you renamed MEPI to GPPI while using the same equation, and why it is referred to as the 'novel GPPI' (as mentioned in lines 20 and 533)?

**Response -L200:** We adopted MEPI as in Forrest et al. (2024) to avoid confusing our readers. We also revised the phrase that says it's a novel index to a refined index. (See lines 224-226).

**L223** Figure 2. This correlation matrix is a part of your results, isn't it? According to Figure 1. Correlation matrix is classified as the first step of your study. If so, please move Figure 2 to the result section.

**Response -L223:** Figure 2,True! We moved the correlation matrix to the results section since it represents part of the study findings. We also briefly described the results from the correlation matrix. (See lines 286-289).

**L229** Is it Figure A3 or Figure 3? Additionally, how many plots did you create ten or eight?

**L229** In my opinion, it would be better not to mention the result image (Figure 3) in Section 2, as this section should focus solely on the data and methodology.

**Response -L229:** We removed the sentence that cites Figure 3 in the methods section and corrected the number of plots created to 8 instead of 10. (See lines 323-326).

**L236** What data did you use for model calibration and testing? Was it GFED5? Please include the calibration and testing process in Figure 1 for clarity.

**Response -L236:** We clarified that we used GFED5 data at different time periods for model calibration (2002 to 2010) and testing (2011 to 2018) of our model. The calibration/testing process was also included in Figure 1 for clarity. (See line 124-128).

**L262** You can delete this sentence, as this information is described in the table caption. This paragraph could start with the sentence "The initial models ... so on"

**Response -L262:** We deleted the redundant sentence and restructured the paragraph for conciseness. (See 149-150).

L274 Could you please explain more details about the reason why you chose Model 25 instead of Model 24 or another model with better deviance and NME? If it is for the reason of future projection of RD data, we can refer to the deviance and NME values in the second, third and so on, because based on the explanation, these two variables are a reference for whether the model is good or not. If it is for simpler reasons, you better explain how the model is said to be simple, whether because there is no multiplication between predictor variables or other reasons.

**Response -L274 (Model Selection Rationale):** Good point, We clarified that we picked Model 25 as opposed to Model 24 due to its parsimony. By simple model we were referring to model parsimony and that has been clarified in the manuscript. (See lines 301-302).

**L267** Could you please explain why you introduce polynomial terms for Percentage Tree Cover (PTC)?

**Response -L267:** . Great point! We did this following finding by Forrest et al 2024 who found that introducing a polynomial term for PTC improves fire predictions for Europe, hence we tried this at a global scale. (See lines 267-269).

**L277** Could you please explain how the predictor formula in each model is determined, the reasons for summing or multiplying the predictors?

**Response -L277**: Sure, we clarified that our approach was motivated by work by Forrest et all who used this approach in Europe. Our justification reads as follows in the manuscript: "We followed the stepwise approach of variable inclusion, exclusion, interaction terms, log transformations, and polynomial transformations as described by Forrest et al. (2024). While their analysis focused on Europe, our objective was to replicate and apply the method at a global scale." . (See lines 267-269).

**L287** Please change the exponent symbol (e) to  $\times$  10x

**Response -L287:** We changed the exponent symbol (e) will be replaced with " $\times$  10 $^{x}$ " for consistency in notation. . (See lines 307-309).

**L302** Could you please explain what Larson and 1972 means? I checked the References and there is no citation information.

**Response -L302**: Good point! Larson and McCleary (1972) provided a description of the use of partial residual plots in regression analysis in detail. We included the reference of this citation in the list of references (See 781-782).

**L303** (GPP) or (GPPI)?

**Response -L303:** We revised GPPI to MEPI consistently across the manuscript to avoid this confusion.

L308 Please search additional references related to similar GLM modeling that have similar or lower explained deviance values than yours, and provide explanations to strengthen your results that the values are accepted.

**Response -L308:** Great suggestion. Not many studies have used GLM to model fires at this scale, which is one of the strengths of our study. We cited the relevant studies that used GLM for global fire modelling in our discussions.

**L322** Please write the year of the observed burnt area (GFED5) and predicted burnt area datasets in Figure 4.

**Response -L322:** We explicitly indicated the years of the observed burned area (GFED5) and predicted burned area datasets in Figure 4. Specifically, we used the testing period 2011 - 2018 for validation of our predictions using observed data. (See lines 344-347)

**L330** Could you please explain the reason for adjusting the HDI predictor to be included, excluded, or constant? I don't see any explanation of this HDI setting in the methods section or in sub-section 2.5. Anthropogenic Influence Predictor.

**Response -L330:** Good suggestion! We provided the rationale for including, excluding, or keeping HDI constant in the models to be added to the methodology section. The rationale reads "To evaluate model sensitivity to inclusion of HDI, we trained our model based on the three settings: including, excluding and holding HDI constant" (See lines 199-201).

**L331** Please remove the title of Figure 5 "Internanual variability .." at the top of the graph, as it is already written in the figure caption.

**L331** Please reduce the burnt area unit (y-axis) to be 2 digits, so that the digit is not too long by applying  $\times$  10x

**Response -L331:** We removed the title in Figure 5 to avoid redundancy. We also formatted the burnt area unit (y-axis) to two digits using "× 10^x" notation. (See lines 355-359).

**L338** Please write the full name of the abbreviation SHAF, SHSA, NHAF, CEAS and so on (because it has not been explained before)

Response -L338: Good point! We introduced the full names of abbreviations for all the 14 GFED regions in the methods section. The introduction reads "The 14 GFED regions include, Boreal North America (BONA), Temperate North America (TENA), Central America (CEAM), Northern Africa (NHAF), Southern Africa (SHAF), Europe (EURO), the Middle East (MIDE), Equatorial Asia (EQAS), Southern Asia (SEAS), Boreal Asia (BOAS), Temperate Asia (TEAS), Australia and New Zealand (AUST), and Northern Hemisphere South America (NHSA), and Southern Hemisphere South America (SHSA)" (See lines 276-280).

L345 Please clarify the figure caption to be understandable by the reader. Is it a interannual variability comparison of burnt areas between model projections and GFEDv5?

**L345** Please clarify the figure caption to be understandable by the reader. Is it a interannual variability comparison of burnt areas between model projections and GFEDv5?

**Response -L345:** We improved the caption for Figure 5 to indicate that it presents an interannual variability comparison between model projections and GFED5 burned area estimates. The caption now reads "Figure 5: Interannual variability in burnt area extent showing the observed trend (based

on GFED5 burnt estimates detection and model projections under different HDI treatments: when HDI was excluded. included and held constant from the value of the first year in the model". (See lines 359-363).

**L355** To make it briefer, you can combine figure 6 and figure 7.

**Response -L355:** Good suggestion! We merged Figures 6 and 7 to streamline the presentation of our results. (See lines 372-377).

**L367** Please remove image title above the graph "Global Seasonal Cycle" and adjust the y-axis not using "k", you can use  $\times$  10x

**Response -L367:** We removed the figure title "Global Seasonal Cycle" and adjusted the y-axis formatting for to using  $\times$  10x instead of k to ensure consistency across figures. (See lines 393-396).

**L372** Please delete the paragraph between section 4. Discussion and sub section 4.1. It's better to discuss the research result directly.

**Response -L372:** We deleted the paragraph between Section 4 and Subsection 4.1 to allow a direct transition into the discussion without confusing our readers. (See lines 400-401).

L378 I have not seen any explanation about what DGVM you used in this study and how you integrate your GLM with those DGVM, either in the methodology section or elsewhere. How can you state that this GLM is compatible in DGVM? To my current understanding, each DGVM has its own characteristics, starting from their programming language and the flow of how it reads specific input data (so it needs data handling / pre-processing to be integrated into each DGVM).

**Response -L378 (DGVM Integration Explanation):** We clarified that full DGVM integration requires further modifications and that our study focuses on developing a DGVM agnostic statistical model. We provided a dedicated subsection in the discussion on DGVM integration. (See lines 562-689).

L423 Please provide cross-reference form figure that support this statement, so that readers can easily refer to specific figure. In addition, please provide more explanation why your model exhibits stronger performance in those regions including in the northern hemisphere.

Response -L423: We included a cross-reference to a supporting figure. We also included the explanation for why the model performs better in certain regions. It now reads "....other world regions (See Fig. 4). The stronger performance in these areas is likely due to the well-defined and predictable fire regimes in these regions. Since fire activity here is strongly governed by distinct wet-dry seasonal cycles, which align closely with climate variables such as precipitation, temperature, and vegetation productivity, factors that our model captures effectively using linear functions (van der Werf et al., 2017; Archibald et al., 2013). These regions typically exhibit lower interannual variability in fire occurrence, facilitating better model generalization." (See lines 446-461).

**L432** After this sentence, it is better to explain how this model contributes to novel insights into the factors that influence global fire trends, and after that you can compare with other studies.

**Response -L432:** We provided a discussion on how the model contributes to novel insights into global fire trends before comparing it with previous studies. The revised phrase reads "Our model has contributed novel insights to the existing understanding of the factors influencing global fire

trends by revealing that excluding the HDI from our model and holding it constant to the value of the first year predicted a steady trend that deviates from the observed negative trend in global fire extent and including HDI follows a decreasing trend that aligns with the observed trend (Fig. 5)." (See lines 469-468).

**L442** "This highlights the significant influence of HDI in projecting the purported negative global ire trend". If so, you can further discuss how the spatio-temporal variability of projected burnt area of each region (in this sub-section or other sub-section), how HDI affects the burnt area in the region. If the HDI data is the same as this source: <a href="https://hdr.undp.org/data-center/human-">https://hdr.undp.org/data-center/human-</a> development-index#/indicies/HDI, then you can associate regions with low, medium, high and very high HDI.

Response -L442 (HDI Influence Discussion): Very valuable suggestion. While regional analysis of HDI contributions could offer interesting insights, we chose not to pursue this direction to preserve the explanatory power of HDI for the global decline in wildfires, as reported in previous studies such as Andela et al. (2017). Our primary objective is to develop a global statistical model that captures broad-scale trends suitable for integration with DGVMs, rather than focusing on regional patterns.

**L442** "The HDI is related to factors like advancements in fire control methods, surveillance, technology, and outreach strategies increasing awareness, particularly in response to the growing human technological developments". Can you add one or some references that supports this statement?

**Response** -L442: Good point, we included a citation from a paper by Teixeira et al (2023) which supports our discussion. However, we also softened the wording to reflect the HDI is a rather general proxy. The text now reads:

"The HDI is rather broad socioeconomic indicator, which we assume acts as a proxy for factor such as investments and advancements in fire control methods, surveillance, technology, and outreach strategies increasing awareness". (See lines 469-4710.

**L450** In this sub-section, you can discuss how the interannual variability of your model in each GFED region (as shown in Figure A1), how your model performs compared to the observational data from GFED5.

**Response -L450:** Thats a thoughtful suggestion. We deliberately kept the discussion of appendix results brief to maintain clarity and avoid potential confusion, given that the focus of the manuscript is on developing a simple global statistical model. The interannual variability of different regions was included in the appendix to provide additional context without detracting from the global perspective of the core analysis.

**L463** Please also discuss another DGVM that used SPITFIRE fire module, as SPITFIRE is an updated fire module from GlobFIRM. SPITFIRE has implemented full burned area calculations and considers natural ignition factors from lightning and ignition and fire suppression based on population density (Thonicke et al., 2010). SPITFIRE effectively includes human fire suppression on other lands because human ignitions first increase and then decrease with increasing population density (Hantson et al., 2016). Models that explicitly simulate the impact of human suppression on fire growth or burnt area (CLM, CLASS-CTEM, JSBACH-SPITFIRE, LPJ-GUESS-SIMFIRE-BLAZE) are

better at representing the spatial pattern in burnt area compared to models which do not include this effect (0.85 and 0.93 respectively). (Hantson et al., 2020) Please try to check the following papers that discuss FireMIP and DGVM used to simulate fire and burned biomass emissions resulting from forest fires. Historical (1700–2012) global multi-model estimates of the fire emissions from the Fire Modeling Intercomparison Project (FireMIP) (Li et al., 2019) The Fire Modeling Intercomparison Project (FireMIP), phase 1: experimental and analytical protocols with detailed model descriptions (Rabin et al., 2017) The status and challenge of global fire modelling (Hantson et al., 2016)

Response -L463 (SPITFIRE and FireMIP Discussion): We fully agree that making reference GlobFIRM only does not give a balanced view and that those papers give a wealth of information about fire modules and different DGVMs. However, we prefer not to focus on any specific DGVM - our reference to GlobFIRM and perhaps by inference LPJ-GUESS was a little misleading in this regard. The fire module here would be compatible with practically any DGVM, so we prefer to keep the discussion general. Therefore we have changed the text to:

"We note that in the recent comparison of fire-enabled DGVMs in the Fire Model Intercomparison Project (FireMIP) project (Hantson et al. 2020), all models did a poorer job of matching the interannual variability than the spatial patterns by a considerable margin. The seven acceptably-performing models achieved a mean spatial NME (across all data and model comparisons) of 0.84 with respect to spatial patterns, but an NME of 1.15 for interannual variability. (See lines 489-492).

**L468** "The findings of this study exhibit robustness in capturing seasonal cycles (R2= 0.536)," Could you please include a cross-reference an image that states this? To make it easier for readers to refer to the results you are discussing. Please apply throughout the rest of the section, when you mention the results of the study, include a cross-reference with a supporting figure or table.

**Response -L468:** A cross-reference to a relevant figure was provided to support the claim that the model captures seasonal cycles well ( $R^2 = 0.536$ ). This practice was applied throughout the section to improve the traceability of results. (See lines 509-510).

L471 Could you please provide evidence to support this assertion that it is due to climatic conditions in those regions? You can compare seasonal fire patterns and climatic conditions in those regions and discuss the result in this sub-section.

Response -L471: Great suggestion, we provided a reference to support our attribution of climate to this discussion. This is supported as follows in the manuscript: "This discrepancy could be attributed to the intricate climatic conditions inherent to these regions, which influence fires weather in a manner that eludes simple linear modeling. For instance, regions with clear-cut wet and dry seasons tend to exhibit more regular fire cycles, largely governed by seasonal shifts in precipitation, temperature, and vegetation growth. These predictable patterns make them well-suited to linear modeling approaches (Archibald et al., 2013; van der Werf et al., 2017). In contrast, areas in the northern hemisphere experience more irregular and less seasonally driven fire activity. Here, the interaction of drought events, land management, and socio-economic drivers introduces variability that weakens model performance (Forkel et al., 2019; Chuvieco et al., 2021). Additionally, varied ignition sources in temperate and boreal zones disrupt consistent seasonal fire patterns (Flannigan et al., 2009)" (See lines 512-518).

**L478** In my opinion, I think this paragraph is better presented at the beginning of sub section 4.5.

**Response -L478:** Good suggestion, the paragraph was moved to the beginning of the section for improved logical flow. (See lines 496-507).

**L478** Do you do future predictions of annual or seasonal burned area data globally?

**Response -L478:** Great question, we did not make any future predictions for seasonal and annual burned area in this study, however we splitted our study period into training period (2002 - 2010) and prediction period (2011 - 2018). This was clarified in the method section of the manuscript.

**L490** I suggest renaming this sub-section to "model limitation and excluded drivers". Include the explanation and discussion of "model shortcomings" in sub-section 4.7 to this section.

**Response -L490:** Great suggestion, we renamed the subsection to "model limitations and excluded drivers," and integrated discussions on model shortcomings into this section. (See lines 525-526).

**L500** "FAPAR is highly correlated with GPP." Please confirm, according to Figure 2. FAPAR correlation with GPP is 0.59, or do you mean FAPAR in general which includes FAPAR, FAPAR12 and FAPAR6?

**Response -L500 (FAPAR-GPP Correlation):** Great suggestion, We clarified that FAPAR in general is correlated with GPP to avoid confusion. (See lines 535-536).

**L512** I suggest this sub-section be changed to 4.7. Recommendations (after you separate the discussion of shortcomings, as I suggested in L490). Or you could also combine section 5. Conclusions and Recommendations, to harmonize after you discuss the Conclusions, you can suggest recommendations regarding further studies.

**Response -L512 (Recommendations Section):** Good suggestion, We renamed this section to "Next steps for DGVM integration, and future directions and model improvements" (See lines 559-560).

**L513** "The findings of this study offer valuable insights into the underlying drivers and patterns shaping global fire dynamics". The sentence does not explain the model shortcomings or recommendations. It seems better to put it in the Conclusion section. In addition, in this subsection, please explain about recommendations only, you can discuss recommendations on how to solve the limitations or shortcomings of the current model, or further studies from this research.

**Response -L513:** Good point. We removed this sentence and revised the subsection to focus solely on model limitations, including how to address current model limitations and potential directions for future research. (See lines 548-557)

**L525** This first paragraph doesn't fit in the Conclusion section, it's more like an introduction, I suggest deleting this first paragraph.

**Response -L525 (Conclusion Introductory Paragraph):** Great suggestion. We removed the introductory paragraph as it does not fit the conclusion structure. (See lines 586-587).

**L531** Make sure the Conclusion section explains the research objectives. The first two sentences have answered the first objective, but add how much the major predictors correlate with burnt area (B A).

**Response -L531:** We revised the conclusion to explicitly summarize research objectives and findings, including the correlation of major predictors with burned area (BA). (See lines 587-607).

L534 Explain the performance of the model when predicting BA compared to the GFED5 observational data -> state the evaluation value index that you used for validating the model before predicting, how do you state the model is suitable to be used to predict BA.

**Response -L534:** The model's performance relative to GFED5 observational data was decsribed, including validation metrics. It reads "While our parsimonious model exhibited limitations in predicting the interannual variability of global fires, it demonstrated commendable accuracy in forecasting the spatial (NME = 0.72). The strength of similarity between observed and predicted seasonal cycles varied according to the GFED region with  $R^2$  ranging between 0.06 to 0.99. Its standout performance laid in capturing the seasonal variability, especially in regions often characterized by distinct wet and dry seasons, notably southern Africa ( $R^2$  = 0.72 to 0.99), Australia ( $R^2$  68) and South America ( $R^2$  = 0.75 to 0.90)". (See lines 594-602).

L536 Before the phrase "We hope", explain the third objective, explain how the model performance, including spatio-temporal, interannual and seasonal cycle of BA compares with GFED5 observational data.

**Response -L536 (Model Performance Summary):** Great suggestion, we provided the NME that we used to evaluate our model and the corre;ation values indicating the strength of relationship between my observed and predicted seasonal cycles for outstanding GFED regions. We only provided over R2 for the interannual variability due to the poor performance of our model for interannual variability in different GFED regions. (See lines 599-602).

**List of additional supporting of references**

- Balch, J.K., Bradley, B.A., Abatzoglou, J.T., Nagy, R.C., Fusco, E.J., Mahood, A.L., 2017. Human-started wildfires expand the fire niche across the United States. Proc. Natl. Acad. Sci. 114, 2946–2951. https://doi.org/10.1073/pnas.1617394114
- Haas, O., Keeping, T., Gomez-Dans, J., Prentice, I.C., Harrison, S.P., 2024. The global drivers of wildfire. Front. Environ. Sci. 12, 1438262.
- Janssen, T.A., Jones, M.W., Finney, D., Van der Werf, G.R., van Wees, D., Xu, W., Veraverbeke, S., 2023. Extratropical forests increasingly at risk due to lightning fires. Nat. Geosci. 16, 1136–1144.

Santoro, M., Cartus, O., Carvalhais, N., Rozendaal, D. M. A., Avitabile, V., Araza, A., de Bruin, S., Herold, M., Quegan, S., Rodríguez-Veiga, P., Balzter, H., Carreiras, J., Schepaschenko, D., Korets, M., Shimada, M., Itoh, T., Moreno Martínez, Á., Cavlovic, J., Cazzolla Gatti, R., da Conceição Bispo, P., Dewnath, N., Labrière, N., Liang, J., Lindsell, J., Mitchard, E. T. A., Morel, A., Pacheco Pascagaza, A. M., Ryan, C. M., Slik, F., Vaglio Laurin, G., Verbeeck, H., Wijaya, A., and Willcock, S.: The global forest above-ground biomass pool for 2010 estimated from high-resolution satellite observations, Earth System Science Data, 13, 3927–3950, https://doi.org/10.5194/essd-13-3927-2021, 2021.

Teixeira, J.C.M., Burton, C., Kelly, D.I., Folberth, G.A., O'Connor, F.M., Betts, R.A., Voulgarakis, A., 2023. Representing socio-economic factors in the INFERNO global fire model using the Human Development Index. Biogeosciences Discuss. 2023, 1–27.

---

## Referee Report (RR1)

**General comment**

Overall, the revised manuscript shows noticeable improvements. However, there are several key points raised during the first-round review have not yet been adequately addressed.

I strongly recommend that the authors provide a clear and direct response to each of these previously raised points, indicating precisely where the revisions have been implemented in the manuscript (e.g., by referencing specific line numbers). This will greatly facilitate the cross-checking process and ensure transparency in the revision. Not just answering "Yes, I will adjust..."

Additionally, the authors should exercise greater attention to detail in presenting information to avoid inconsistencies or errors—such as the total number of models, the number and names of predictors, table headers, and the agreed-upon terminology. For example, the term Gross Primary Productivity Index (GPPI) should consistently be replaced with Monthly Ecosystem Productivity Index (MEPI) as previously agreed.

Lastly, the manuscript's structure should be refined. Specifically, the explanation of model validation using observational data should precede any presentation or discussion of model predictions. This adjustment will enhance the logical flow and clarity of the results section.

Please refer to the detailed comments provided below for point-by-point feedback.

**Detailed Comments**

L1 "A statistical global burnt area model tailored for integration into Dynamic Global Vegetation Models"

At this review stage, I personally find the current article title still not yet fully firm-particularly with regard to the mention of DGVM integration, which, based on your manuscript, has not yet been technically implemented (still an intention).

Referring to your response to the first-round review, particularly at points L14 and L378, I would suggest revising the title to better reflect the actual scope of the study. For instance:

"Development of a Biophysical and Socioeconomic-Based Generalized Linear Model for Global Burned Area Simulation", or

"Development of a Biophysical and Socioeconomic-Based Statistical Model for Global Burned Area Simulation within a DGVM-Compatible Framework"

I hope this suggestion is helpful for your consideration, as the title plays a crucial role in accurately representing the core contribution and scope of your research.

**L31** "The model presented should be compatible with most, if not all, DGVMs used to develop future scenarios."

Please delete this sentence.

This sentence could unintentionally weaken your argument, as it may suggest uncertainty regarding the model's compatibility with DGVMs.

Hope you'll consider to apply my suggestion on (L1) - indicating that it has been developed within a DGVM-compatible framework, without overstating its level of integration. This allows readers to understand that while integration with DGVMs is feasible, it is not necessarily straightforward. In practice, successful integration requires careful consideration of technical aspects, including differences in programming languages, data structures, and input-process—output mechanisms.

For instance, although it is quite complicated and challenging, SPITFIRE has been successfully implemented across multiple DGVMs within the FireMIP project, but such integration has required significant technical adaptation. Your model, if developed within a compatible framework, can follow a similar pathway.

**L37** "Notably, climate change has led to more severe fire weather in large parts of the world and record fires have recently occurred in Australia and Canada, burning more than 15 million and 7 million ha (Jain et al., 2024; MacCarthy et al., 2024)."

As previously suggested (first round review L36), I recommend including one historical burned area data for Australia and Canada for years prior to 2024 to provide a clearer basis for comparison.

Additionally, since the previous sentence states that "climate change led to more severe fire weather," it would strengthen the argument to include a temporal analysis of fire weather conditions in both countries. This would help substantiate the claim by demonstrating how fire weather patterns have evolved over time in response to climate change.

**L127** In Figure 1, the label currently reads "Monthly burnt area **GFEDv5**." To ensure consistency in the use of abbreviations throughout the manuscript, please standardize the format - either use GFED5 or GFEDv5 consistently across all figures, tables, and text.

**Additionally,** please clarify whether the prediction process is part of the Model Evaluation procedure or not?

If not, I suggest separating the Model Prediction process as a distinct subsection following the Model Evaluation. This will help to clearly distinguish between performance assessment and forward-looking application.

In the Model Prediction section, please specify the temporal coverage of the predictionfor example, from which year to which year the model was used to simulate burned area- and provide a clear rationale for this time frame.

L131 Monthly BA data for the periods 2002 and 2018 were derived from monthly mean fractional BA from the GFED5.

Please clarify whether this refers to the entire period from 2002 to 2018 (inclusive), rather than only the years 2002 and 2018. If so, I recommend revising the sentence to accurately reflect the time range, such as:

"Monthly BA data for the period 2002–2018 were derived from the monthly mean fractional BA provided by GFED5."

**L151** Please review the formatting and content of Table 1 carefully:**

- 1) The table caption should be placed above the table, in accordance with standard journal formatting conventions.
- 2) Please ensure that the "Repeat Header Row" option is enabled. Currently, the header on pages 7 and 8 differs from that on page 6. The column headings should consistently read "Temporal Resolution" and "Source", not "Temporal" and "Predictor".
- 3) Additionally, please include appropriate citations for the PGC and AGB data sources in the table to ensure transparency and proper attribution.
- 4) I recommend adding an additional column titled "Temporal Coverage". This will help clarify the specific years associated with each predictor dataset used in the analysis and enhance the transparency and reproducibility of the study.

**L157 "We used eight vegetation predictor variables..."**

On Line 158, only **six** vegetation predictors are listed, whereas Table 1 includes **nine** vegetation-related predictors.

Please review and clarify the exact number of vegetation predictors used in the analysis to ensure consistency across the manuscript.

L166 I recommend reviewing the opening phrases of each paragraph in this sub section or other section if any. The phrase "we used" is repeated frequently, which may reduce the overall readability and stylistic variation of the text.

Additionally, please consider improving the logical flow and continuity between paragraphs to ensure a more cohesive narrative. Enhancing transitions and reducing redundancy will help maintain reader engagement and strengthen the overall presentation of your work.

**L249 I would like to reiterate a key point from the first round of review: I recommend avoiding the use of the phrase "seamless integration into DGVMs."**

This is not a matter of opposition to the idea itself, but rather a concern about accuracy and scientific rigor. In the current study, you have developed a Generalized Linear Model (GLM) under the DGVM framework, with the intention that it may be integrated into DGVMs in the future. However, as the model has not yet been implemented or tested within an actual DGVM - nor validated across multiple DGVM platforms - the claim of "seamless integration" is not currently supported by sufficient evidence.

For this reason, I suggest using more cautious and evidence-based wording to reflect the current status of your model development and its potential for future integration.

**L251** The sentence "Calibration of the model utilized data from 2002 to 2010 while testing utilized data from 2011 to 2018" should be revised for clarity and consistency.

First, please explicitly mention that the data used are from GFED5.

Second, kindly verify the accuracy of this statement, as Figure 1 indicates that GFED5 data from 2002–2010 were used for model training, while data from 2011–2018 were used for model evaluation.

Lastly, I recommend using terminology consistently throughout the manuscript. Please choose a uniform set of terms - such as training, testing, calibration, evaluation or validation, and ensure that their usage aligns with standard definitions, as each term has a distinct meaning in modeling studies.

**L266** The sentence "A total of 25 model runs were conducted, each..." appears to be inconsistent with the information provided in Table A1, which lists 26 models.

Please clarify whether 25 or 26 model runs were actually performed and ensure consistency throughout the manuscript.

**L628** Could you please explain why you moved Table 2 (initial manuscript) to the appendix as Table A1 in the current revised version?

In my view, the information presented in Table A1 is essential to the core content of the manuscript, as it outlines the structure and configuration of Models 1 through 26 - an integral part of the model development process presented in this study. Including this table in the main text would improve clarity and allow readers to more easily follow the comparisons and results throughout the paper.

If the relocation was in response to a specific request from another reviewer, I might understand the consideration. Alternatively, if necessary, guidance from the editor may be helpful in determining the most appropriate placement for this table.

**L289** The sentence "The initial models (Model 1 to Model 3) progressively include more variables, however, a noticeable jump in deviance explained when PNTC is added (Model 3: 0.5298)." Should be improved for clarity and consistency.

While it is understandable that "Model 3: 0.5298" refers to the deviance explained by Model 3, the current phrasing may be unclear to some readers. I suggest rephrasing the sentence to explicitly state that Model 3 explains 52.98% of the deviance or adjusting the deviance writing style on the sentence.

Additionally, there is inconsistency in how models and their deviance values are presented in this sub-section. For example, "Model 15 (~0.5664789)" in Line 295 follows a different format.

Please revise the entire section to ensure a consistent and reader-friendly presentation of model identifiers and their corresponding deviance explained values.

**L300** The phrase "and consisted of other variables that we don't have future projections for (e.g., RD)" is not sufficiently formal for a scientific manuscript.

I suggest rephrasing it to something more appropriate, such as:

"and included variables for which future projections are currently unavailable (e.g., RD), due to the lack of established projection models or datasets."

This revision would provide a clearer and more scientifically sound justification. If possible, please support this statement with references or evidence to strengthen the rationale for excluding such variables from future scenario modeling.

L635 Could you please explain how the predictor formula in each model is determined, the reasons for summing or multiplying the predictors? This question was asked since the first review round (L277) but I couldn't find the explanation in the revised manuscript.

I believe it would be helpful - for both myself and the broader readership, to provide a clear rationale for the modeling approach used. Specifically, Models 1 to 12 apply only the sum of predictor variables, while subsequent models begin incorporating multiplicative interactions between predictors.

**Was this modeling structure formula based on established references, or was it determined subjectively defined by the authors?**

As a modeler, particularly from the perspective of DGVM applications, this distinction is crucial. In DGVMs, each variable or parameter is typically defined through well-established empirical or mechanistic relationships, reflecting biophysical, physiological, and ecological processes. The interactions between environmental drivers - such as soil, vegetation, disturbance, and atmospheric variables, lightning – as well as anthropogenic driver such as population density - are governed by interdependent equations grounded in process-based understanding.

Therefore, introducing mathematical operations such as summation or multiplication without a strong theoretical or empirical basis may present challenges for future model integration and validity. Arbitrary combinations may not align with the underlying mechanisms of DGVMs and could compromise the scientific robustness of future applications. I encourage the authors to explain and clarify the conceptual or empirical justification for the mathematical formulations applied in each or overall model structure.

I suggest adding this explanation in **Section 2.4**, positioned sequentially *before* explanation about model performance assessment.

L325 "Predictor variables were Gross Primary Production Index (GPP)..."

It appears that the term **Gross Primary Production Index (GPP?)** is still used in several parts of the manuscript, despite your agreement in the first-round review's response (see comment L200) to consistently use MEPI, as also shown in Figure 3.

For clarity and consistency, I recommend revising the manuscript to uniformly refer to this variable as MEPI throughout the text, figures, and tables.

L330 Please search additional references related to similar GLM modeling that have similar or lower explained deviance values than yours, and providing explanations to strengthen your results that the values are accepted. This point has been asked since the first review round (L308) but I couldn't find any change in this section.

The comparison with Haas et al. (2022), which reports a deviance explained value of 69%, may inadvertently weaken the presentation of your own model's performance (56.8%).

Since this comparison appears in Section 3 (Results), I recommend reconsidering its inclusion. The Results section should primarily focus on presenting your own findings, while comparative analysis with previous studies would be more appropriately placed in the Discussion section - if sufficiently supported.

If no additional references or contextual justification are available to frame this comparison constructively, it may be better to omit it altogether to maintain a focused and balanced presentation of your results.

**L345** Figure 4: As previously mentioned in the **first-round review (see comment L24)**, it is necessary to present a comparison between the observed and predicted burned area data before displaying the map titled "Predicted Burnt Area: 2011–2018."

In general modeling practice, model performance should be evaluated prior to making and presenting predictions. This includes comparing observed and predicted data using spatial analyses or summary statistics, and ideally supported by visualizations such as scatter plots.

If the model demonstrates satisfactory performance, the presentation of the spatial prediction for the 2011–2018 period will be better justified and more scientifically robust.

**L356** Figure 5: Could you please clarify whether the predicted burned area shown covers the period from 2002 to 2018, or only from 2011 to 2018 only?

Kindly ensure that your response is consistent with the point raised in comment L127. If the prediction only covers a subset of the years, please adjust the graph accordingly to reflect the correct prediction period.

Additionally, I recommend improving the x-axis labeling by clearly indicating both the start and end years. If space is limited, consider using italicized text or reduced font size so that all years (e.g., 2002–2018) can be displayed legibly. This will enhance clarity and make the graph more reader-friendly.

L361 Please write the full name of the abbreviation SHAF, SHSA, NHAF, CEAS and so on (because it has not been explained before). This point has been asked since the first review round (L338) but I couldn't find any additional information.

For improved clarity and accessibility, I recommend including the full definitions of all of these abbreviations used in Table A2.

**L370** Figure 6b presents the validation results of the selected model through a comparison between observed and predicted burned area data.

I suggest repositioning this figure before Figure 4, as it is more appropriate to present and discuss model validation prior to showing the prediction outputs. This will improve the logical flow of the results section.

Please also clarify the temporal coverage of the comparison: for example, confirm that both GFED5 and model data span 2011–2018, and specify whether the comparison is based on annual average data.

Additionally, please revise the figure caption to include this information and improve clarity.

This comment is closely related to your response to **point L345.** I recommend to show the scatter plot comparison between predicted and observed GFED5 data on the Supplementary file in accordance with this Figure 6 to increase the validation clarity.

L393 Figure 8. This figure presents the results of model validation using seasonal observation data.

In line with my previous comment, I recommend combining both the annual and seasonal validation results into a single figure to facilitate direct comparison and enhance clarity. Once the validation results are clearly presented and discussed, you may proceed with displaying and interpreting the prediction outputs.

This revised order will improve the logical flow of the results and allow readers to better assess model performance before evaluating its predictive capabilities.

**L395 Section 4. Discussion**

I recommend maintaining the subsection title 4.1. Main Drivers of Global Burned Area to help clearly categorize and structure the Discussion section.

As noted in my first-round review (comment L372), my suggestion was to delete only the intervening paragraph between the main section title (4. Discussion) and subsection 4.1, not the subsection itself.

Therefore, the structure should be:

- 4. Discussion
- 4.1. Main Drivers of Global Burned Area

**L396** I recommend removing the sentence "We found a DGVM-compatible parsimonious global statistical model made of FWI, PNTC, PTC, TPI, MEPI, HDI, VAT, and NDD."

Placing this statement at the beginning of the paragraph may be inappropriate, as the claim of "DGVM-compatible" is not yet strongly substantiated within the manuscript.

Instead, I suggest focusing the paragraph on discussing the main drivers of global burned area, in line with the theme of subsection 4.1. This will ensure better alignment with the section's objective and maintain a coherent flow of discussion based on the model results.

**L459** The sentence "Our model has contributed novel insights to the existing understanding of the factors influencing global fire trends (Joshi and Sukumar, 2021; Kraaij et al., 2018; Mukunga et al., 2023)" remains vague and insufficiently supported.

Could you please explain whether they say that your model contributed novel insights to the global fire trends?

As previously noted in my first-round review (comment L432), but authors didn't make any change on this point. This claim should be substantiated by explaining how your model offers novel insights. Simply stating the contribution without elaboration or contextual comparison weakens the impact of your findings.

I recommend strengthening this statement by clearly articulating what specific advancements or new perspectives your model provides - such as integrating novel predictor combinations, improved spatial resolution, or enhanced predictive accuracy - and then positioning these insights relative to previous studies.

A clearer structure could be:

"Our model has contributed novel insights to the existing understanding of the factors influencing global fire trends because "reason A" (Reference), "reason B" (Reference), so on if any. Previous studies, such as Andela ..."

**L468** "This highlights the significant influence of HDI in projecting the purported negative global ire trend".

The response provided by the authors to comment L442 in the first-round review does not appear to be fully implemented in the revised manuscript. While the authors agreed to include

a discussion on HDI regional classification to improve the logical flow between sentences, this has not been clearly reflected in the current version.

For example, the sentence "This highlights the significant influence of HDI in projecting the purported negative global fire trend." is immediately followed by "HDI is a rather broad socioeconomic indicator, which we assume acts as a proxy for factors..."

The connection between these two sentences is weak, and the paragraph lacks a transitional explanation or regional perspective that would justify the claim. I recommend adding a brief discussion on HDI variability across regions or classifications (e.g., low-, middle-, high-HDI countries), and how this influences fire trends. This addition would strengthen the coherence of the paragraph and enhance the reader's understanding of HDI's role in your model.

L496 "Globally, our model predicts a notable peak in burnt areas during February and August."

Could you please include a cross-reference to an image that states this?

L512 Could you please provide evidence to support this assertion that it is due to climatic conditions in those regions? You can compare seasonal fire patterns and climatic conditions in those regions and discuss the result in this sub-section. This point has been asked since the first review round (L471) but I couldn't find any additional information.

Author responded "Evidence supporting the assertion that climatic conditions influence fire dynamics in specific regions will be provided by comparing seasonal fire patterns with climate variables."

This can be addressed with a relatively straightforward approach. I suggest plotting the seasonal fire pattern alongside the relevant seasonal climate variables discussed in this section, to strengthen your argument.

You may include this plot as a supplementary figure, and provide an appropriate cross-reference to it within the main text.

L587 "We sought ..., both globally and regionally."

I recommend deleting the first paragraph of the Conclusion section, as it is not appropriate to restate the research objectives in this part of the manuscript. As mentioned in my first-round review (comment L531), the Conclusion should focus on explaining how the research objectives were achieved, based on the results and discussion presented.

I suggest starting with a concise summarizing statement such as:

Line 592 "We present a parsimonious statistical model specifically tailored for global burned area simulation."

Please remove the phrase "with the goal of integration into DGVMs" at this point, should emphasize demonstrated outcomes rather than intentions.

Then after Line 592, please discuss points 1) how the major drivers in the model you use, how the major drivers accommodate the fire incident or burnt area factors. 2) how the model can be integrated in DGVM and 3) how your model performs against interannual and seasonal observational data - global and regional.

**Note:** Please ignore the quotation 1), 2) and 3) -> my intention writing the quotation number to make ease the explanation of your research objective sequentially. Please write the content in a continuous, flowing narrative that cohesively summarizes how the model addresses and fulfills the research objectives.

**L604** "We hope that our research outcomes will stimulate a more rigorous implementation of global fire models within DGVM frameworks."

Please rewrite the sentence, perhaps something like this would be better

"The parsimonious statistical model developed in this study has demonstrated strong performance in simulating global burned area patterns. With further development, it holds potential for integration into DGVMs to enhance the representation of fire dynamics..."

---

## Referee Report (RR2)

**General comment**

I appreciate that the authors have adequately addressed all previous comments. From my perspective, no major issues remain, only a few minor corrections are needed to further improve the clarity and consistency of the manuscript.

I would also like to thank the Editor for managing the review process of this manuscript efficiently and professionally.

Please refer to the minor comments listed below for final adjustments.

**Minor Comments**

**L256-261** In my opinion, it is not necessary to include these sentences, as they describe basic and well-known technical procedures especially for scientific audience. Therefore, I recommend deleting the following lines "These time periods were chosen ..."

L710 The term "GPP Index" is still present in the manuscript. Please revise it to maintain consistency with the agreed terminology, "Monthly Ecosystem Productivity Index (MEPI)", throughout the entire text.

L719 (See L370, previous review round). Since the global burned area comparison does not exhibit a strong correlation, and Figure 4 already presents a comparison between predicted and observed burned area (GFED5) using GFED regional boundaries, I suggest maintaining consistency in the analysis.

Specifically, please revise Figure A4 by changing it from an interannual global comparison to an interannual comparison by GFED regional boundaries, to better complement and strengthen the results shown in Figure 4.

Accordingly, Figure A4 about "Interannual Comparison by GFED Regional Boundaries", and Figure B5 as "Seasonal Comparison by GFED5 Regional Boundaries" for clarity and coherence.

---

## Author Response (AR2)

**Reviewer response letter**

We appreciate the reviewers for their detailed and constructive feedback. In response, we have carefully revised the manuscript to address all points and improve clarity, transparency, and scientific rigor. Key changes include:

- 1. Introduction & contextualization: Historical burned area data for Australia and Canada have been added to provide temporal comparisons, strengthening the link between climate change and fire weather conditions.
- 2. A new title following the suggestion by the reviewer: "Development of a Model for Global Burned Area Simulation within a DGVM-Compatible Framework"
- 3. Modeling rationale & transparency: Section 2.4 now clearly explains the rationale behind additive (Models 1–12) and multiplicative (Models 13–26) predictor formulations. A new "Rationale" column in Table 2 summarizes conceptual and statistical justifications for each model configuration.
- 4. Terminology & consistency: All instances of GPPI have been replaced with MEPI, table headers standardized, and the number of models and predictors verified for accuracy.
- 5. Results & figures: The manuscript now presents model evaluation using observational data prior to predictions. Figures and captions have been updated for clarity, including temporal coverage and cross-references to supplementary figures. Annual and seasonal evaluation results have been combined for better readability.
- 6. Discussion & conclusions: The Discussion now clearly articulates the novel contributions of the model, including the role of major drivers and regional HDI variations. The conclusion has been streamlined to focus on demonstrated outcomes, model performance, and potential for DGVM integration.

These revisions collectively address all reviewer comments, improve the manuscript's readability and methodological transparency and strengthen the presentation of the study's findings.

**General comment**

Overall, the revised manuscript shows noticeable improvements. However, there are several key points raised during the first-round review that have not yet been adequately addressed.

I strongly recommend that the authors provide a clear and direct response to each of these previously raised points, indicating precisely where the revisions have been implemented in the manuscript (e.g., by referencing specific line numbers). This will greatly facilitate the crosschecking process and ensure transparency in the revision. Not just answering "Yes, I will adjust..."

Additionally, the authors should exercise greater attention to detail in presenting information to avoid inconsistencies or errors—such as the total number of models, the number and names of predictors, table headers, and the agreed-upon terminology. For example, the term Gross Primary Productivity Index (GPPI) should consistently be replaced with Monthly Ecosystem Productivity Index (MEPI) as previously agreed.

Lastly, the manuscript's structure should be refined. Specifically, the explanation of model validation using observational data should precede any presentation or discussion of model predictions. This adjustment will enhance the logical flow and clarity of the results section.

Please refer to the detailed comments provided below for point-by-point feedback.

Response: Thank you for your thoughtful and constructive feedback. We have carefully reviewed all previous comments and have provided detailed responses to each, clearly indicating where the revisions have been made in the manuscript (by referencing specific line numbers). We have ensured that all issues are addressed directly, rather than providing vague responses. Also, we have thoroughly revised the manuscript to ensure consistent use of terminology, particularly replacing "Gross Primary Productivity Index (GPPI)" with "Monthly Ecosystem Productivity Index (MEPI)" throughout the document. Additionally, we have corrected any inconsistencies regarding the number and names of models, predictors, and table headers (e.g., Table 2, lines 272-273). As suggested, we have restructured the results section to present the model evaluation using observational data before model predictions. This adjustment improves the logical flow and clarity of the manuscript (see lines 351-385 for evaluation and lines 386-430 for predictions).

**Detailed Comments**

**L1** "A statistical global burnt area model tailored for integration into Dynamic Global Vegetation Models"

At this review stage, I personally find the current article title still not yet fully firm- particularly with regard to the mention of DGVM integration, which, based on your manuscript, has not yet been technically implemented (still an intention).

Referring to your response to the first-round review, particularly at points **L14** and **L378**, I would suggest revising the title to better reflect the actual scope of the study. For instance:

"Development of a Biophysical and Socioeconomic-Based Generalized Linear Model for Global Burned Area Simulation", or

"Development of a Biophysical and Socioeconomic-Based Statistical Model for Global Burned Area Simulation within a DGVM-Compatible Framework"

I hope this suggestion is helpful for your consideration, as the title plays a crucial role in accurately representing the core contribution and scope of your research.

Response: Thanks for the suggestion. We have revised the title of the manuscript accordingly and made made it concise and it now reads "Development of a Statistical Model for Global Burned Area Simulation within a DGVM-Compatible Framework" [Line 1-3]

L31 "The model presented should be compatible with most, if not all, DGVMs used to develop future scenarios."

**Please delete this sentence.**

This sentence could unintentionally weaken your argument, as it may suggest uncertainty regarding the model's compatibility with DGVMs.

Hope you'll consider to apply my suggestion on (L1) - indicating that it has been developed within a DGVM-compatible framework, without overstating its level of integration. This allows readers to understand that while integration with DGVMs is feasible, it is not necessarily straightforward. In practice, successful integration requires careful consideration of technical aspects, including differences in programming languages, data structures, and input–process–output mechanisms.

For instance, although it is quite complicated and challenging, SPITFIRE has been successfully implemented across multiple DGVMs within the FireMIP project, but such integration has required significant technical adaptation. Your model, if developed within a compatible framework, can follow a similar pathway.

Response: The sentence stating, "The model presented should be compatible with most, if not all, DGVMs used to develop future scenarios," has been deleted as recommended. In its place, we emphasize that the model was developed within a DGVM-compatible framework, without overstating the level of integration (See lines 14-15). This revision clarifies that while integration with DGVMs is feasible, but not pursued in this study. However, we would like to point out, purely as a matter of historical accuracy and because some coauthors were involved in these activities, that SPITFIRE was implemented in many DGVMs independently from FireMIP. The role of FireMIP was to provide a systematic comparison of these implementations. Furthermore, the SPITFIRE model is far more complicated than the GLM presented here and integrating it into a DGVM requires more fundamental model development. Moreover, a European GLM-based model developed with the same approach but for Europe (Forrest et al. 2024), which also inspired the work here, has in the meantime been integrated within the LPJmL model, which was even for us surprisingly easy. For example, model skill was hardly reduced by replacing remote sensing-based vegetation input variables with those simulated by LPJmL. However, we are still finalizing the manuscript on this work and acknowledge that we cannot build upon these results for the manuscript here.

**L37** "Notably, climate change has led to more severe fire weather in large parts of the world and record fires have recently occurred in Australia and Canada, burning more than 15 million and 7 million ha (Jain et al., 2024; MacCarthy et al., 2024)."

As previously suggested (first round review L36), I recommend including one historical burned area data for Australia and Canada for years prior to 2024 to provide a clearer basis for comparison.

Additionally, since the previous sentence states that "climate change led to more severe fire weather," it would strengthen the argument to include a temporal analysis of fire weather conditions in both countries. This would help substantiate the claim by demonstrating how fire weather patterns have evolved over time in response to climate change.

Response: We appreciate the reviewer's suggestion and have revised the introduction to include historical burned area data for both Australia and Canada to provide a clear temporal comparison. The passage now reads "Globally, the impacts of climate change continue to manifest through extreme weather events and changes in weather patterns (Clarke et al., 2019). In Australia, for example, the mean annual burned area in forested regions was about 1.8 million ha per year between 1988–2001, increasing to 3.5 million ha per year between 2002–2018, before the 2019–20 "Black Summer" fires burned over 15 million ha nationally (Canadell et al., 2021; Australian Government, 2020). Similarly, in Canada, the 1986–2022 mean annual burned area was about 2.1 million ha, compared with the record-breaking 7.8 million ha burned in 2023 (Jain et al., 2024; Curasi et al., 2024; MacCarthy et al. 2024). These multi-decadal increases in burned area in both countries are consistent with evidence that climate change has intensified fire-conducive weather over time. "These temporal comparisons demonstrate a clear escalation in burned area over recent decades, consistent with published evidence that climate change has intensified fire-conducive weather conditions in both countries. [See lines 34 - 40]

**L127** In Figure 1, the label currently reads "Monthly burnt area **GFEDv5**." To ensure consistency in the use of abbreviations throughout the manuscript, please standardize the format - either use GFEDv5 or GFEDv5 consistently across all figures, tables, and text.

**Additionally,** please clarify whether the prediction process is part of the Model Evaluation procedure or not?

If not, I suggest separating the Model Prediction process as a distinct subsection following the Model Evaluation. This will help to clearly distinguish between performance assessment and forward-looking application.

In the Model Prediction section, please specify the temporal coverage of the predictionfor example, from which year to which year the model was used to simulate burned area- and provide a clear rationale for this time frame.

Response: Great suggestions. We've now adopted GFED5 as the standard name and revised all sections of the manuscript accordingly. Also we separated the model training, testing, and prediction and evaluation components on figure 1 and specified the time periods. Additionally we added a new section which clarifies model training and testing (Section 2.5) in which the training and testing periods are clarified as; "We used data from the period 2002–2010 for model training, the period 2011–2018 for model testing, and the full period 2002–2018 dataset for predictions and model evaluation. These time periods were chosen to ensure that the testing data remained independent from the training data while also allowing predictions to span a sufficiently long timeframe to enhance the robustness of the analysis and evaluation. The essence of splitting

training vs testing is to train the model on training data, and then check that the results are similarly good on the testing data (for example, no overfitting to the training data) before making predictions on the full dataset. During model testing we compared the performance of the model on training data vs training data to assess model robustness" [See lines 252-259].

**L131** Monthly BA data for the periods **2002 and 2018** were derived from monthly mean fractional BA from the GFED5.

Please clarify whether this refers to the entire period from 2002 to 2018 (inclusive), rather than only the years 2002 and 2018. If so, I recommend revising the sentence to accurately reflect the time range, such as:

"Monthly BA data for the period 2002–2018 were derived from the monthly mean fractional BA provided by GFED5."

Response: Good suggestion. As described above we added a new section on model training and testing(2.5) which clearly describes how data are split and the rationale behind the selected periods. In this section the revised passage reads "We used data from the period 2002–2010 for model training, the period 2011–2018 for model testing, and the full period 2002–2018 dataset for predictions and model evaluation. These time periods were chosen to ensure that the testing data remained independent from the training data while also allowing predictions to span a sufficiently long timeframe to enhance the robustness of the analysis and evaluation. The essence of splitting training vs testing is to train the model on training data, and then check that the results are similarly good on the testing data (for example, no overfitting to the training data) before making predictions on the full dataset. During model testing we compared the performance of the model on training data vs training data to assess model robustness" [See lines 252-259].

**L151** Please review the formatting and content of Table 1 carefully:**

- 1) The table caption should be placed above the table, in accordance with standard journal formatting conventions.
- 2) Please ensure that the "Repeat Header Row" option is enabled. Currently, the header on pages 7 and 8 differs from that on page 6. The column headings should consistently read "Temporal Resolution" and "Source", not "Temporal" and "Predictor".
- 3) Additionally, please include appropriate citations for the PGC and AGB data sources in the table to ensure transparency and proper attribution.
- 4) I recommend adding an additional column titled "Temporal Coverage". This will help clarify the specific years associated with each predictor dataset used in the analysis and enhance the transparency and reproducibility of the study.

Response: Great suggestions regarding Table 1. We have carefully revised the table to address all points:

- 1. The table caption has been moved above the table in accordance with journal formatting conventions.
- 2. The "Repeat Header Row" option has been enabled to ensure consistent headers across all pages.
- 3. The column headings have been updated to consistently read "*Temporal Resolution*" and "*Source*."
- 4. Appropriate citations for the PGC and AGB data sources have been added to enhance transparency and ensure proper attribution.
- 5. We have added a new column titled "*Temporal Coverage*" to clarify the specific years associated with each predictor dataset, thereby improving the reproducibility and clarity of the analysis. [See lines 151 152]

**L157 "We used eight vegetation predictor variables..."**

On Line 158, only **six** vegetation predictors are listed, whereas Table 1 includes **nine** vegetation-related predictors.

Please review and clarify the exact number of vegetation predictors used in the analysis to ensure consistency across the manuscript.

Response: We appreciate the reviewer for pointing out this discrepancy. Upon review, we have clarified that we used nine vegetation predictors in the analysis. This update has ensured consistency between the text and Table 1. [See line 156]

**L166** I recommend reviewing the opening phrases of each paragraph in this sub section or other section if any. The phrase "we used" is repeated frequently, which may reduce the overall readability and stylistic variation of the text.

Additionally, please consider improving the logical flow and continuity between paragraphs to ensure a more cohesive narrative. Enhancing transitions and reducing redundancy will help maintain reader engagement and strengthen the overall presentation of your work.

Response: Thanks for the feedback. We have revised the opening phrases throughout the subsection to reduce the repetitive use of "we used" and improve stylistic variation. For example: some of the passages were revised begin as as follows "PGZC, PRC, PTNC and PTC were used to evaluate the relationship between landcover and burnt area distribution" (See lines 166-167); "GPP, AGB, and FAPAR were proxies for vegetation health and productivity and type, and fuel load" (See lines 168); "To evaluate how topography can influence the occurrence and spread of fires, we incorporated topographic positioning index (TPI)" (See lines 189-190)

Additionally, we have enhanced the logical flow and continuity between paragraphs by refining transitions and minimizing redundancy. [See lines 156-187]

**L249** I would like to reiterate a key point from the first round of review: **I recommend avoiding** the use of the phrase "seamless integration into DGVMs."

This is not a matter of opposition to the idea itself, but rather a concern about accuracy and scientific rigor. In the current study, you have developed a Generalized Linear Model (GLM) under the DGVM framework, with the intention that it may be integrated into DGVMs in the future. However, as the model has not yet been implemented or tested within an actual DGVM - nor validated across multiple DGVM platforms - the claim of "seamless integration" is not currently supported by sufficient evidence.

For this reason, I suggest using more cautious and evidence-based wording to reflect the current status of your model development and its potential for future integration.

Response: We agree that "seamless integration" was potentially misleading. We have revised the manuscript to replace the phrase "seamless integration into DGVMs" with more cautious and evidence-based wording that reflects the current status of our model. Specifically, we emphasize that the GLM was developed to be "DGVM-compatible" or "for ease of transference to other modelling frameworks" (See lines 251-252). While this particular GLM-based model has not yet been implemented or tested within an actual DGVM, framing it as DGVM-compatible highlights its design alignment and flexibility, without implying untested integration.

**L251** The sentence "Calibration of the model utilized data from 2002 to 2010 while testing utilized data from 2011 to 2018" should be revised for clarity and consistency.

First, please explicitly mention that the data used are from GFED5.

Second, kindly verify the accuracy of this statement, as Figure 1 indicates that GFED5 data from 2002–2010 were used for model training, while data from 2011–2018 were used for model evaluation.

Lastly, I recommend using terminology consistently throughout the manuscript. Please choose a uniform set of terms - such as training, testing, calibration, evaluation or validation, and ensure that their usage aligns with standard definitions, as each term has a distinct meaning in modeling studies.

Response: We have revised the sentence for clarity and consistency. Specifically, we now explicitly state that the data are from GFED5 in Figure 1. We have also clarified the periods for training, testing, precision and evaluation using consistent terminology in the new section we added (Section 2.5). There is passage in this section which clarifies these periods as follows: "We used data from the period 2002–2010 for model training, the period 2011–2018 for model testing, and the full period 2002–2018 dataset for predictions and model evaluation. These time periods were chosen to ensure that the testing data remained independent from the training data while also allowing predictions to span a sufficiently long timeframe to enhance

the robustness of the analysis and evaluation. The essence of splitting training vs testing is to train the model on training data, and then check that the results are similarly good on the testing data (for example, no overfitting to the training data) before making predictions on the full dataset. During model testing we compared the performance of the model on training data vs training data to assess model robustness." [See lines 254-261]. The term "validation" is not used anymore as it might indicate that a model is generally valid and is often confused with evaluation, which we think is more appropriate for a model like here.

**L266** The sentence "A total of 25 model runs were conducted, each..." appears to be inconsistent with the information provided in Table A1, which lists 26 models.

Please clarify whether 25 or 26 model runs were actually performed and ensure consistency throughout the manuscript.

Response: We appreciate the reviewer for identifying this inconsistency. Upon review, we confirm that 26 model runs were conducted, as correctly listed in Table A1. The manuscript has been updated to reflect this number consistently throughout, ensuring alignment between the text and the table. Also, all other numbers have been carefully checked. [See line 290]

**L628** Could you please explain why you moved Table 2 (initial manuscript) to the appendix as Table A1 in the current revised version?

In my view, the information presented in Table A1 is essential to the core content of the manuscript, as it outlines the structure and configuration of Models 1 through 26 - an integral part of the model development process presented in this study. Including this table in the main text would improve clarity and allow readers to more easily follow the comparisons and results throughout the paper.

If the relocation was in response to a specific request from another reviewer, I might understand the consideration. Alternatively, if necessary, guidance from the editor may be helpful in determining the most appropriate placement for this table.

Response: The relocation of Table 2 (initial manuscript) to the appendix as Table A1 was made following a suggestion from the editor during the revision process. This change was intended to streamline the main text while preserving all essential methodological details for transparency and reproducibility. We agree that Table A1 provides important information on the structure and configuration of Models 1 through 26. We have ensured that the main text clearly references Table A1 at relevant points, so readers can easily follow the comparisons and results presented in the study. Besides we have now included an abridged version of Table 2 which summarises models (M1–M26) with corresponding formulas, performance metrics, and rationale for predictor inclusion or interaction terms [274-277]

**L289** The sentence "The initial models (Model 1 to Model 3) progressively include more variables, however, a noticeable jump in deviance explained when PNTC is added (Model 3: 0.5298)." Should be improved for clarity and consistency.

While it is understandable that "Model 3: 0.5298" refers to the deviance explained by Model 3, the current phrasing may be unclear to some readers. I suggest rephrasing the sentence to explicitly state that Model 3 explains 52.98% of the deviance or adjusting the deviance writing style on the sentence.

Additionally, there is inconsistency in how models and their deviance values are presented in this sub-section. For example, "Model 15 (~0.5664789)" in Line 295 follows a different format.

Please revise the entire section to ensure a consistent and reader-friendly presentation of model identifiers and their corresponding deviance explained values.

Response: Thanks for these constructive suggestions. The first sentence has been revised for clarity and now reads "The initial models (model 1 to model 3) had many variables and a significant improvement is observed in model 3 which explained 52.98% following the inclusion of PNTC. Models 4 to 8 involve adding vegetation (FAPAR) and various land use types (PCC, PPS, PRC, PGC). This is accompanied by marginal improvement in deviance explained, indicating these factors provide some additional predictive power but are not as impactful as existing vegetation covariates (such as GPP). Models 10 to 12 introduce polynomial terms for PTC. This results in an increase in performance, explaining 55.88% in Model 12. Models 13 to 16 incorporate interactions between HDI and land use types (e.g., PCC and PRC), resulting in marginal improvement in performance with the highest recorded in Model 15 which explained 56.65%. Models 19 to 26 fine-tune the overall performance by incorporating various variables and their interactions. Model 24, which includes a comprehensive set of climatic, vegetation, human, and topographic variables along with their interactions, achieves the highest performance as it explained 57.20%. The marginal improvements observed in subsequent models indicate that, while additional variables contribute to the model performance, the primary influencing factors were already identified by Model 19, however it was not the simplest model (~ parsimonious) and included variables for which future projections are currently unavailable (e.g., RD), due to the lack of established projection models or datasets. Therefore, model ... was chosen as the final model." [See lines 321-328]

**L300** The phrase "and consisted of other variables that we don't have future projections for (e.g., RD)" is not sufficiently formal for a scientific manuscript.

I suggest rephrasing it to something more appropriate, such as:

"and included variables for which future projections are currently unavailable (e.g., RD), due to the lack of established projection models or datasets."

This revision would provide a clearer and more scientifically sound justification. If possible, please support this statement with references or evidence to strengthen the rationale for excluding such variables from future scenario modeling.

Response: Good suggestion. The sentence has been revised to a more formal and scientifically precise wording: "and included variables for which future projections are currently unavailable (e.g., RD), due to the lack of established projection models or datasets." A supporting statement providing a clear rationale for excluding these variables modeling was provided and it reads "Since the main objective of the study was to produce a DGVM compatible model, availability of future projections for these datasets was indispensable to model building." [See line 323 -326]

**L635** Could you please explain how the predictor formula in each model is determined, the reasons for summing or multiplying the predictors? **This question was asked since the first review round** (L277) but I couldn't find the explanation in the revised manuscript.

I believe it would be helpful - for both myself and the broader readership, to provide a clear rationale for the modeling approach used. Specifically, Models 1 to 12 apply only the sum of predictor variables, while subsequent models begin incorporating multiplicative interactions between predictors.

**Was this modeling structure formula based on established references, or was it determined subjectively defined by the authors?**

As a modeler, particularly from the perspective of DGVM applications, this distinction is crucial. In DGVMs, each variable or parameter is typically defined through well-established empirical or mechanistic relationships, reflecting biophysical, physiological, and ecological processes. The interactions between environmental drivers - such as soil, vegetation, disturbance, and atmospheric variables, lightning – as well as anthropogenic driver such as population density - are governed by interdependent equations grounded in process-based understanding.

Therefore, introducing mathematical operations such as summation or multiplication without a strong theoretical or empirical basis may present challenges for future model integration and validity. Arbitrary combinations may not align with the underlying mechanisms of DGVMs and could compromise the scientific robustness of future applications. I encourage the authors to explain and clarify the conceptual or empirical justification for the mathematical formulations applied in each or overall model structure.

I suggest adding this explanation in **Section 2.4**, positioned sequentially *before* explanation about model performance assessment.

Response: We thank the reviewer for raising this point again and apologise that our previous revision did not sufficiently clarify the rationale for the model formulations. In response, we have substantially revised Section 2.4 and we now have a new section (2.6) that is dedicated

to describe the model selection procedure. A clarifying passage in this section reads "We employed a sequential model-building approach, beginning with additive structures ((M1–M12) to estimate the independent contribution of climate, vegetation, and human variables on burned area (Table 2). This approach aligns with established fire risk modeling practices (e.g., Forrest et al 2024). Additional predictors were introduced if they represented ecologically meaningful processes (e.g., drought severity, vegetation productivity) and improved model fit (deviance explained and Normalised Mean Error). Multiplicative interaction terms (M13 onward) were added only when fire ecology theory suggested synergistic effects (e.g., human ignitions under extreme weather, vegetation dryness and temperature) and retained if deviance explained improved. This stepwise approach ensures both statistical rigor and ecological interpretability rather than ad hoc formula selection." To improve transparency, we have added a Rationale column in Table 2 summarizing the conceptual and statistical justification for each model formulation. This ensures our modeling approach is systematic, theory-informed and empirically grounded, addressing the reviewer's concern about potential arbitrariness. [See lines 264 - 277]

**L325 "Predictor variables were Gross Primary Production Index (GPP)..."**

It appears that the term **Gross Primary Production Index (GPP?)** is still used in several parts of the manuscript, despite your agreement in the first-round review's response (see comment L200) to consistently use MEPI, as also shown in Figure 3.

For clarity and consistency, I recommend revising the manuscript to uniformly refer to this variable as MEPI throughout the text, figures, and tables.

Response: We thank the reviewer for highlighting this issue. We have carefully revised the manuscript to ensure that the variable is referred to as MEPI on figure 3 caption [see line 349] and throughout the text, and tables, in accordance with our previous agreement. Where GPP still exists, it's representing the inputs for determining MEPI not GPPI (as it was formerly addressed)

L330 Please search additional references related to similar GLM modeling that have similar or lower explained deviance values than yours, and providing explanations to strengthen your results that the values are accepted. This point has been asked since the first review round (L308) but I couldn't find any change in this section.

The comparison with Haas et al. (2022), which reports a deviance explained value of 69%, may inadvertently weaken the presentation of your own model's performance (56.8%).

Since this comparison appears in Section 3 (Results), I recommend reconsidering its inclusion. The Results section should primarily focus on presenting your own findings, while comparative analysis with previous studies would be more appropriately placed in the Discussion section - if sufficiently supported.

If no additional references or contextual justification are available to frame this comparison constructively, it may be better to omit it altogether to maintain a focused and balanced presentation of your results.

Response: We appreciate the recommendation. We have removed the comparison with Haas et al. (2022) from the Results section. We however note also that Haas et al. fitted a model for annual burned area, not capturing seasonal variation like here, and the number of used predictors was more and not constrained by DGVM applications in the discussion section [Lines 105-106]. Thus, a comparison of model performance might not be very useful. However, in the discussion we still mention that the performance here is comparable to other fire-enabled DGVMs on ability to capture interannual variability ".We note that in the recent comparison of fire-enabled DGVMs in the Fire Model Intercomparison Project (FireMIP) project (Hantson et al., 2020), all models did a poorer job of matching the interannual variability than the spatial patterns by a considerable margin. The seven acceptably-performing models achieved a mean spatial NME (across all data and model comparisons) of 0.84 with respect to spatial patterns, but an NME of 1.15 for interannual variability.." (See lines 541-544...).

**L345** Figure 4: As previously mentioned in the **first-round review** (see comment **L24**), it is necessary to present a comparison between the observed and predicted burned area data before displaying the map titled "Predicted Burnt Area: 2011–2018."

In general modeling practice, model performance should be evaluated prior to making and presenting predictions. This includes comparing observed and predicted data using spatial analyses or summary statistics, and ideally supported by visualizations such as scatter plots.

If the model demonstrates satisfactory performance, the presentation of the spatial prediction for the 2011–2018 period will be better justified and more scientifically robust.

Response: We agree and have added a new section (3.2) dedicated to model performance evaluation preceding the presentation of predictions. In this section, we present results demonstrating the strength of the relationship between observed and predicted burned area across GFED regions. These results are now visualized spatially in Figure 4, allowing a clear assessment of model performance prior to the presentation of the 2011–2018 predicted burned area map. [See line 353 - 392]

**L356** Figure 5: Could you please clarify whether the predicted burned area shown covers the period from 2002 to 2018, or only from 2011 to 2018 only?

Kindly ensure that your response is consistent with the point raised in comment **L127**. If the prediction only covers a subset of the years, please adjust the graph accordingly to reflect the correct prediction period.

Additionally, I recommend improving the x-axis labeling by clearly indicating both the start and end years. If space is limited, consider using italicized text or reduced font size so that all years (e.g., 2002–2018) can be displayed legibly. This will enhance clarity and make the graph more reader-friendly.

Response: The data used in Figure 5 cover the full period from 2002 to 2018 showing a comparison between observed and predicted data based on the full dataset. We have updated the x-axis labeling to clearly indicate both the start and end years (2002–2018), adjusting the font size to ensure legibility. [See lines 409-413]

L361 Please write the full name of the abbreviation SHAF, SHSA, NHAF, CEAS and so on (because it has not been explained before). This point has been asked since the first review round (L338) but I couldn't find any additional information.

For improved clarity and accessibility, I recommend including the full definitions of all of these abbreviations used in Table A2.

Response: We thank the reviewer for pointing this out. We have included the full names of all abbreviations (e.g., SHAF, SHSA, NHAF, CEAS) in lines 365-369 for improved clarity and accessibility, and we have included those in Table A2 caption as well.

**L370** Figure 6b presents the validation results of the selected model through a comparison between observed and predicted burned area data.

I suggest repositioning this figure before Figure 4, as it is more appropriate to present and discuss model validation prior to showing the prediction outputs. This will improve the logical flow of the results section.

Please also clarify the temporal coverage of the comparison: for example, confirm that both GFED5 and model data span 2011–2018, and specify whether the comparison is based on annual average data.

Additionally, please revise the figure caption to include this information and improve clarity.

This comment is closely related to your response to **point L345.** I recommend to show the scatter plot comparison between predicted and observed GFED5 data on the Supplementary file in accordance with this Figure 6 to increase the validation clarity.

Response: Thank you for your valuable feedback regarding Figure 6b and its placement within the manuscript. We have carefully considered your suggestions and made the following revisions:

1. In response to your suggestion, we have moved Figure 6b to precede Figure 4, in order to present the model evaluation results prior to the model prediction outputs. This adjustment enhances the logical flow of the results section and we believe it improves the

clarity and structure of the manuscript.

- 2. We have added the requested clarification regarding the temporal coverage of the comparison between the observed and predicted burned area data. Specifically, we confirm that both the GFED5 and model data cover the period 2011–2018. Additionally, we specify that the comparison is based on annual average data, as per your recommendation. This clarification has been incorporated into the revised manuscript (see lines 290-291).
- 3. The figure caption for Figure 6b has been revised to include the temporal coverage and further details on the comparison methodology to improve clarity. The updated caption now reads: "Figure 4. Evaluation of the selected model using observed burned area data from GFED5 predicted data (2011-2018). The map shows r-square values highlighting the model's performance for interannual (a) and seasonal variability (b) per GFED region."
- 4. In line with your suggestion, we have included the scatter plot comparison between predicted and observed GFED5 data in the Supplementary Material (Figure A4), which complements Figure 4 in the main manuscript. [See lines 716-721]

**L393 Figure 8. This figure presents the results of model validation using seasonal observation data.**

In line with my previous comment, I recommend combining both the annual and seasonal validation results into a single figure to facilitate direct comparison and enhance clarity. Once the validation results are clearly presented and discussed, you may proceed with displaying and interpreting the prediction outputs.

This revised order will improve the logical flow of the results and allow readers to better assess model performance before evaluating its predictive capabilities.

Response: We appreciate the suggestion. We have now combined the annual and seasonal evaluation results into a single figure (now Figure 4) to facilitate direct comparison and enhance clarity. We revised the presentation to allow readers to assess model performance more effectively before interpreting the prediction outputs. [See lines 382-386]

**L395 Section 4. Discussion**

I recommend maintaining the subsection title 4.1. Main Drivers of Global Burned Area to help clearly categorize and structure the Discussion section.

As noted in my first-round review (comment L372), my suggestion was to delete only the intervening paragraph between the main section title (4. Discussion) and subsection 4.1, not the subsection itself.

Therefore, the structure should be:

- 4. Discussion
- 4.1. Main Drivers of Global Burned Area

Response: Thanks for the clarification. We have retained the subsection title 4.1. Main Drivers of Global Burned Area and removed only the intervening paragraph between the main Discussion heading and this subsection, as suggested. This revision restores the intended structure and improves the clarity and organization of the Discussion section. [See line 434-441].

**L396** I recommend removing the sentence "We found a DGVM-compatible parsimonious global statistical model made of FWI, PNTC, PTC, TPI, MEPI, HDI, VAT, and NDD."

Placing this statement at the beginning of the paragraph may be inappropriate, as the claim of "DGVM-compatible" is not yet strongly substantiated within the manuscript.

Instead, I suggest focusing the paragraph on discussing the main drivers of global burned area, in line with the theme of subsection 4.1. This will ensure better alignment with the section's objective and maintain a coherent flow of discussion based on the model results.

Response: We agree. In the revised manuscript, we have removed the original sentence referring to a "DGVM-compatible" model from the beginning of the paragraph. The opening statement has been replaced with: "We found that the candidate variables, namely, FWI, PNTC, PTC, TPI, MEPI, HDI, PPN and NDD, had strong influence on burnt areas." This revision focuses the paragraph on the main drivers of global burned area, aligning with the theme of subsection 4.1 and maintaining a coherent flow of discussion based on the model results, without making claims that are not yet substantiated. [See lines 435-437].

**L459** The sentence "Our model has contributed novel insights to the existing understanding of the factors influencing global fire trends (Joshi and Sukumar, 2021; Kraaij et al., 2018; Mukunga et al., 2023)" remains vague and insufficiently supported.

Could you please explain whether they say that your model contributed novel insights to the global fire trends?

As previously noted in my first-round review (**comment L432**), but authors didn't make any change on this point. This claim should be substantiated by explaining how your model offers novel insights. Simply stating the contribution without elaboration or contextual comparison weakens the impact of your findings.

I recommend strengthening this statement by clearly articulating what specific advancements or new perspectives your model provides - such as integrating novel predictor combinations, improved spatial resolution, or enhanced predictive accuracy - and then positioning these insights relative to previous studies.

**A clearer structure could be:**

"Our model has contributed novel insights to the existing understanding of the factors influencing global fire trends because "reason A" (Reference), "reason B" (Reference), so on if any. Previous studies, such as Andela ..."

Response: We appreciate the suggestion. We have strengthened the statement to clearly articulate the novel contributions of our model. The revised text now specifies what previous studies explored and the strengths of our model compared to earlier studies, for example its integration of novel combinations of predictor variables and improved temporal resolution. Additionally, we contextualize these contributions relative to existing literature, highlighting how our findings extend the understanding of factors influencing global fire trends (e.g., Joshi and Sukumar, 2021; Kraaij et al., 2018; Mukunga et al., 2023; Andela et al., 2017). The revised section reads as follows: "Previous studies have improved our understanding of drivers of fire but differ in approach and attributional focus for fire trends. For instance, Joshi and Sukumar (2021) employed region-specific multilayer neural networks to reveal spatially varying sensitivities between fire and socio-environmental drivers, providing strong spatial diagnostics but limited transparency on attributions of burnt area trends. Kraaij et al. (2018) provided detailed biome-level attribution of destructive fires by linking drought, fuel state and vegetation context in case studies (e.g., fynbos/plantation complexes), emphasizing vegetation and weather controls at local scales. Mukunga et al. (2023) used random-forest analyses to quantify the added value of human predictors for ignition probability, focusing on anthropogenic controls of ignitions rather than burnt area extent. Building on these approaches, our study contributes novel attributional insight because it explicitly integrates a compact set of DGVM compatible fire-weather and fuel indices (FWI, PTC, TPI, PNTC) with a socio-economic indicator (HDI) within a parsimonious statistical framework for burnt area trends. This allows direct attribution of directional effects (for example, the negative association between HDI and burnt area) across regions" [See lines 496-506].

**L468** "This highlights the significant influence of HDI in projecting the purported negative global ire trend".

The response provided by the authors to comment **L442** in the first-round review does not appear to be fully implemented in the revised manuscript. While the authors agreed to include a discussion on HDI regional classification to improve the logical flow between sentences, this has not been clearly reflected in the current version.

For example, the sentence "This highlights the significant influence of HDI in projecting the purported negative global fire trend." is immediately followed by "HDI is a rather broad socioeconomic indicator, which we assume acts as a proxy for factors..."

The connection between these two sentences is weak, and the paragraph lacks a transitional explanation or regional perspective that would justify the claim. I recommend adding a brief discussion on HDI variability across regions or classifications (e.g., low-, middle-, high-HDI

countries), and how this influences fire trends. This addition would strengthen the coherence of the paragraph and enhance the reader's understanding of HDI's role in your model.

Response: Thanks for the suggestion. In the revised manuscript, we have strengthened the discussion of HDI by explicitly incorporating a regional perspective. We now briefly describe HDI variability across regions and classifications (e.g., low-, middle-, and high-HDI countries) and explain how these differences influence fire trends. The passage now reads "This highlights the significant influence of HDI in projecting the purported negative global fire trend. Importantly, HDI is not uniform worldwide but varies substantially across regions and levels of socioeconomic development. For instance, in high-HDI countries, greater financial resources, infrastructure, and institutional capacity often translate into stronger investments in fire control technologies, improved surveillance systems, and more effective prevention campaigns. By contrast, in low- and middle-HDI countries, limited resources and weaker institutional frameworks may constrain fire management capabilities, resulting in greater reliance on natural fire dynamics or less formalized suppression efforts. As many countries continue to develop, it translates improvements in HDI and fire management strategies. Although strategies are often implemented independently and on a smaller scale, their cumulative impact on global fire trends is substantial. Thus, HDI serves as a broad socioeconomic indicator that we assume acts as a proxy for the combined effects of investments, advancements in fire control methods, surveillance, technology, and outreach strategies that increase awareness (Teixeira et al., 2023)." This provided a clearer connection between the sentence on HDI's influence in projecting global fire trends and the subsequent discussion of HDI as a broad socioeconomic proxy. [See lines 513-523].

**L496** "Globally, our model predicts a notable peak in burnt areas during February and August."

Could you please include a cross-reference to an image that states this?

Response: A cross-reference to support this statement has been added. The sentence now directs readers to Figure 8, which visually illustrates the predicted peaks in burnt areas during February and August. This ensures that the textual description is clearly linked to the corresponding visual evidence. [See line 548].

**L512** Could you please provide evidence to support this assertion that it is due to climatic conditions in those regions? You can compare seasonal fire patterns and climatic conditions in those regions and discuss the result in this sub-section. **This point has been asked since the first review round (L471) but I couldn't find any additional information.**

Author responded "Evidence supporting the assertion that climatic conditions influence fire dynamics in specific regions will be provided by comparing seasonal fire patterns with climate variables."

This can be addressed with a relatively straightforward approach. I suggest plotting the seasonal fire pattern alongside the relevant seasonal climate variables discussed in this section, to strengthen your argument.

You may include this plot as a supplementary figure, and provide an appropriate crossreference to it within the main text.

Response: Thank you for your continued and insightful feedback. We apologize for not adequately addressing your previous request regarding the influence of climate on fire dynamics in specific regions. In response to your comment, we have made the following revisions:

- 1. We have revised the manuscript to provide a more robust explanation of how climate influences fire dynamics through the FWI to better represent the climatic conditions that drive seasonal fire patterns. We now explicitly show how the FWI is particularly effective in regions with distinct climatic patterns, and as a more comprehensive monthly climate variable. This revision is discussed in detail in Section 4.2 (see lines 480-483), where we highlight how regions with seasonal variability in climatic factors exhibit stronger relationships between the FWI and fire dynamics. The revised passage reads "The stronger performance in these areas is likely due to the well-defined and predictable fire regimes in these regions. Since fire activity here is strongly governed by distinct wet-dry seasonal cycles, which align closely with fire weather, enabling our model to capture these patterns effectively using linear functions (Fig. A5), hence better model generalization."
- 2. We have now included a plot (now Figure A5 in the Supplementary Material) that compares seasonal fire patterns (both observed and predicted) with the relevant seasonal climate variable in our model (FWI). This comparison strengthens our argument that climatic conditions are a primary driver of fire dynamics in certain regions. The plots in Figure A5 clearly illustrate the temporal alignment between seasonal fire patterns and climate variability, reinforcing the role of climate in influencing fire behavior. We have also cross-referenced this supplementary figure in Section 4.2 (see lines 483) to guide readers to the additional information.

**L587** "We sought ..., both globally and regionally."**

I recommend deleting the first paragraph of the Conclusion section, as it is not appropriate to restate the research objectives in this part of the manuscript. As mentioned in my **firstround review** (**comment L531**), the Conclusion should focus on explaining how the research objectives were achieved, based on the results and discussion presented.

I suggest starting with a concise summarizing statement such as:

Line 592 "We present a parsimonious statistical model specifically tailored for global burned area simulation."

Please remove the phrase "with the goal of integration into DGVMs" at this point, should emphasize demonstrated outcomes rather than intentions.

Then after Line 592, please discuss points 1) how the major drivers in the model you use, how the major drivers accommodate the fire incident or burnt area factors. 2) how the model can be integrated in DGVM and 3) how your model performs against interannual and seasonal observational data - global and regional.

**Note:** Please ignore the quotation 1), 2) and 3) -> my intention writing the quotation number to make ease the explanation of your research objective sequentially. Please write the content in a continuous, flowing narrative that cohesively summarizes how the model addresses and fulfills the research objectives.

Response: The first paragraph of the Conclusion section has been removed, as recommended, to avoid restating the research objectives. The revised section now begins with a concise summarizing statement: "We present a parsimonious statistical model specifically tailored for global burned area simulation." The subsequent discussion has been restructured into a continuous, cohesive narrative and now reads "We found the drivers FWI, TPI, and PNTC are positively associated with BA, whereas MEPI, HDI, PPN, and NDD exhibit negative relationships, and PTC showed a unimodal response with strongest effect at intermediate tree cover. The diversity of these drivers underscores the multifaceted influence of both climatic and socio-economic drivers on fire dynamics. Our model explicitly accommodates these drivers, capturing how variations in climate, vegetation productivity, and human development interact to modulate fire occurrence and extent. Notably, the use of HDI to represent societal development as a proxy for fire management capacity and the transition away from firedependent agricultural practices provides a coarse but global socioeconomic driver beyond GDP and population density. Including this in DGVMs can improve fire, vegetation and human feedbacks, particularly with respect to Shared Socioeconomic Pathways (SSPs, O'Neill et al., 2017) or other scenarios." (See lines 641-652].

**L604** "We hope that our research outcomes will stimulate a more rigorous implementation of global fire models within DGVM frameworks."

Please rewrite the sentence, perhaps something like this would be better

"The parsimonious statistical model developed in this study has demonstrated strong performance in simulating global burned area patterns. With further development, it holds potential for integration into DGVMs to enhance the representation of fire dynamics..."

Response: We appreciate the suggestion. The sentence has been revised as suggested to emphasize demonstrated outcomes rather than intentions. It now reads: "The parsimonious statistical model developed in this study has demonstrated strong performance in simulating global burned area patterns. It holds potential for integration into DGVMs to enhance the representation of fire dynamics, albeit it remains to be tested how well the model performs

when remote-sensing-derived vegetation and land cover variables are replaced with those simulated by a DGVM" [See lines 654-657]

---

## Author Response (AR3)

**Reviewer response letter**

We thank the editor and reviewer's for their valuable feedback and constructive suggestions that have helped us further improve the clarity, consistency, and accessibility of our manuscript. In this revised version, we have implemented the following major updates:

- Colour accessibility improvements (Figures 6 and 7):
   In response to the editor's comment, we revised the colour schemes in Figures 6 and 7 to ensure they are fully accessible for readers with colour vision deficiencies
- 2. Minor textual and figure adjustments:
  Following the reviewer's detailed feedback, we have refined the manuscript by
  ensuring terminology consistency, added new figures (Figure A4) and improving the
  coherence of figure captions. These revisions collectively enhance the clarity,
  precision, and presentation of our findings.

Below, we provide specific point-by-point responses outlining how each comment has been addressed.

**General comments**

**Editor's comment**

Please ensure that the colour schemes used in your maps and charts allow readers with colour vision deficiencies to correctly interpret your findings. Please check your figures using the Coblis – Color Blindness Simulator (https://www.color-blindness.com/coblis-color-blindness-simulator/) and revise the colour schemes accordingly with the next file upload request. -> Fig. 6, 7

Response: We thank the Editor for this important reminder. Figures 6 (Line 410-411) and 7(Line 425-427) have been fully revised using a color-blind friendly palette to ensure accessibility for readers with color vision deficiencies. Line styles and contrasts have also been adjusted to improve clarity.

**Reviewer 2**

**General comment**

I appreciate that the authors have adequately addressed all previous comments. From my perspective, no major issues remain, only a few minor corrections are needed to further improve the clarity and consistency of the manuscript. I would also like to thank the Editor for managing the review process of this manuscript efficiently and professionally. Please refer to the minor comments listed below for final adjustments.

Response: We appreciate the positive feedback and for acknowledging our revisions. We also appreciate the editor's management of the review process. We have carefully addressed all the minor comments below to further enhance the clarity and consistency of the manuscript.

**Minor Comments**

L256-261 In my opinion, it is not necessary to include these sentences, as they describe basic and well-known technical procedures especially for scientific audience. Therefore, I recommend deleting the following lines "These time periods were chosen ..."

Response: We thank the reviewer's suggestion. The indicated sentences have been removed from the revised manuscript to improve conciseness and avoid redundancy of well-known information. [Lines 256–260]

L710 The term "GPP Index" is still present in the manuscript. Please revise it to maintain consistency with the agreed terminology, "Monthly Ecosystem Productivity Index (MEPI)", throughout the entire text.

Response: Thank you for noticing this oversight. We have carefully reviewed the entire manuscript and replaced all remaining instances of "GPP Index" with "Monthly Ecosystem Productivity Index (MEPI)" to ensure consistent terminology throughout the text and figures. [Line 710]

L719 (See L370, previous review round). Since the global burned area comparison does not exhibit a strong correlation, and Figure 4 already presents a comparison between predicted and observed burned area (GFED5) using GFED regional boundaries, I suggest maintaining consistency in the analysis. Specifically, please revise Figure A4 by changing it from an interannual global comparison to an interannual comparison by GFED regional boundaries, to better complement and strengthen the results shown in Figure 4. Accordingly, Figure A4 about "Interannual comparison by GFED regional boundaries", and Figure B5 as "Seasonal Comparison by GFED5 Regional Boundaries" for clarity and coherence.

Response: We appreciate the constructive suggestion. Following the recommendation, Figure A4 caption has been revised to "Scatter plots illustrating interannual comparison by GFED regional boundaries between observed burnt area fraction (GFED5) and predicted burnt area fraction for the period between 2002 and 2018", and new figures for different GFED regions were added (see Line 720-728). Likewise, Figure B5 caption has been updated and renamed as "Shows the seasonal comparison by GFED5 regional boundaries between observed burnt area (in red), predicted burnt (in blue), fire weather index (in green)" to improve clarity. [Lines 723–725]